# Contextual bandits with surrogate losses:
# Margin bounds and efficient algorithms

**Dylan J. Foster**
Cornell University
djfoster@cs.cornell.edu

**Akshay Krishnamurthy**
Microsoft Research, NYC
akshay@cs.umass.edu

## Abstract

We use surrogate losses to obtain several new regret bounds and new algorithms for contextual bandit learning. Using the ramp loss, we derive new margin-based regret bounds in terms of standard sequential complexity measures of a benchmark class of real-valued regression functions. Using the hinge loss, we derive an efficient algorithm with a $\sqrt{dT}$-type mistake bound against benchmark policies induced by $d$-dimensional regressors. Under realizability assumptions, our results also yield classical regret bounds.

## 1  Introduction

We study sequential prediction problems with partial feedback, mathematically modeled as *contextual bandits* [29]. In this formalism, a learner repeatedly (a) observes a *context*, (b) selects an *action*, and (c) receives a *loss* for the chosen action. The objective is to learn a policy for selecting actions with low loss, formally measured via *regret* with respect to a class of benchmark policies. Contextual bandit algorithms have been successfully deployed in online recommendation systems [5], mobile health platforms [48], and elsewhere.

In this paper, we use surrogate loss functions to derive new *margin-based* algorithms and regret bounds for contextual bandits. Surrogate loss functions are ubiquitous in supervised learning (cf. [49, 8, 43]). Computationally, they are used to replace NP-hard optimization problems with tractable ones, e.g., the hinge loss makes binary classification amenable to convex programming techniques. Statistically, they also enable sharper generalization analysis for models including boosting, SVMs, and neural networks [43, 6], by replacing dependence on dimension in VC-type bounds with distribution-dependent quantities. For example, to agnostically learn $d$-dimensional halfspaces the optimal rates for excess risk are $\sqrt{d/n}$ for the 0/1 loss benchmark and $\frac{1}{\gamma} \cdot \sqrt{1/n}$ for the $\gamma$-margin loss benchmark [27], meaning *the margin bound removes explicit dependence on dimension*. Curiously, surrogate losses have seen limited use in partial information settings (some exceptions are discussed below). This paper demonstrates that these desirable computational and statistical properties indeed extend to contextual bandits.

In the first part of the paper we focus on statistical issues, namely whether *any algorithm* can achieve a generalization of the classical margin bound from statistical learning [10] in the adversarial contextual bandit setting. Our aim here is to introduce a theory of learnability for contextual bandits, in analogy with statistical and online learning, and our results provide an information-theoretic benchmark for future algorithm designers. We consider benchmark policies induced by a class of real-valued regression functions and obtain a regret bound in terms of the class' *sequential metric entropy*, a standard complexity measure in online learning [42]. As a consequence, we show that $\tilde{\mathcal{O}}(T^{\frac{d}{d+1}})$ regret is achievable for Lipschitz contextual bandits in $d$-dimensional metric spaces, improving on a recent result of Cesa-Bianchi et al. [14], and that an $\tilde{O}(T^{2/3})$ mistake bound is achievable for bandit multiclass prediction in smooth Banach spaces, extending Kakade et al. [26].

Technically, these results build on the non-constructive minimax analysis of Rakhlin et al. [42], which, for the online adversarial setting, prescribes a recipe for characterizing statistical behavior of arbitrary classes, and thus provides a counterpart to empirical risk minimization in statistical learning. Indeed, for full-information problems, this approach yields regret bounds in terms of sequential analogues of standard complexity measures including Rademacher complexity and metric entropy. However, since we work in the contextual bandit setting, we must extend these arguments to incorporate partial information. To do so, we leverage the adaptive minimax framework of Foster et al. [20] along with a careful "adaptive" chaining argument.

In the second part of the paper, we focus on computational issues and derive two new algorithms using the hinge loss as a convex surrogate. The first algorithm, HINGE-LMC, provably runs in polynomial time and achieves a $\sqrt{dT}$-mistake bound against $d$-dimensional benchmark regressors with convexity properties. HINGE-LMC is the first efficient algorithm with $\sqrt{dT}$-mistake bound for bandit multiclass prediction using a surrogate loss without curvature, and so it provides a new resolution to the open problem of Abernethy and Rakhlin [2]. This algorithm is based on the exponential weights update, along with Langevin Monte Carlo for efficient sampling and a careful action selection scheme. The second algorithm is much simpler: in the stochastic setting, Follow-The-Leader with appropriate smoothing matches our information-theoretic results for sufficiently large classes.

## 1.1 Preliminaries

Let $\mathcal{X}$ denote a context space and $\mathcal{A} = \{1, \ldots, K\}$ a discrete action space. In the adversarial contextual bandits problem, for each of $T$ rounds, an adversary chooses a pair $(x_t, \ell_t)$ where $x_t \in \mathcal{X}$ is the context and $\ell_t \in [0,1]^K \triangleq \mathcal{L}$ is a loss vector. The learner observes the context $x_t$, chooses an action $a_t$, and incurs loss $\ell_t(a_t) \in [0,1]$, which is also observed. The goal of the learner is to minimize the cumulative loss over the $T$ rounds, and, in particular, we would like to design learning algorithms that achieve low *regret* against a class $\Pi \subset (\mathcal{X} \to \mathcal{A})$ of benchmark policies:

$$\text{Regret}(T, \Pi) \triangleq \sum_{t=1}^{T} \mathbb{E}[\ell_t(a_t)] - \inf_{\pi \in \Pi} \sum_{t=1}^{T} \mathbb{E}[\ell_t(\pi(x_t))].$$

In this paper, we always identify $\Pi$ with a class of vector-valued regression functions $\mathcal{F} \subset (\mathcal{X} \to \mathbb{R}^K_{=0})$, where $\mathbb{R}^K_{=0} \triangleq \{s \in \mathbb{R}^K : \sum_a s_a = 0\}$. We use the notation $f(x) \in \mathbb{R}^K$ to denote the vector-valued output and $f(x)_a$ to denote the $a^{\text{th}}$ component. Note that we are assuming $\sum_a f(x)_a = 0$, which is a natural generalization of the standard formulation for binary classification [8] and appears in Pires et al. [34]. Define $B \triangleq \sup_{f \in \mathcal{F}} \sup_{x \in \mathcal{X}} \|f(x)\|_\infty$ to be the maximum value predicted by any regressor.

Our algorithms use *importance weighting* to form unbiased loss estimates. If at round $t$, the algorithm chooses action $a_t$ by sampling from a distribution $p_t \in \Delta(\mathcal{A})$, the loss estimate is defined as $\hat{\ell}_t(a) \triangleq \ell_t(a_t)\mathbf{1}\{a_t = a\}/p_t(a)$. Given $p_t$, we also define a smoothed distribution as $p_t^\mu \triangleq (1 - K\mu)p_t + \mu$ for some parameter $\mu \in [0, 1/K]$.

We introduce two surrogate loss functions, the *ramp loss* and the *hinge loss*, whose scalar versions are defined as $\phi^\gamma(s) \triangleq \min(\max(1 + s/\gamma, 0), 1)$ and $\psi^\gamma(s) \triangleq \max(1 + s/\gamma, 0)$ respectively, for $\gamma > 0$. For $s \in \mathbb{R}^K$, $\phi^\gamma(s)$ and $\psi^\gamma(s)$ are defined coordinate-wise. We start with a simple lemma, demonstrating how $\phi^\gamma, \psi^\gamma$ act as surrogates for cost-sensitive multiclass losses.

**Lemma 1** (Surrogate Loss Translation). *For $s \in \mathbb{R}^K_{=0}$, define $\pi_{ramp}(s), \pi_{hinge}(s) \in \Delta(\mathcal{A})$ by $\pi_{ramp}(s)_a \propto \phi^\gamma(s_a)$ and $\pi_{hinge}(s)_a \propto \psi^\gamma(s_a)$. For any vector $\ell \in \mathbb{R}^K_+$, we have*

$$\langle \pi_{ramp}(s), \ell \rangle \le \langle \ell, \phi^\gamma(s) \rangle \le \sum_{a \in \mathcal{A}} \ell(a)\mathbf{1}\{s_a \ge -\gamma\}, \qquad and \qquad \langle \pi_{hinge}(s), \ell \rangle \le K^{-1}\langle \ell, \psi^\gamma(s) \rangle.$$

Based on this lemma, it will be convenient to define $L_T^\gamma(f) \triangleq \sum_{t=1}^{T} \sum_{a \in \mathcal{A}} \ell_t(a)\mathbf{1}\{f(x_t)_a \ge -\gamma\}$, which is the *margin-based cumulative loss* for the regressor $f$. $L_T^\gamma$ should be seen as a cost-sensitive multiclass analogue of the classical margin loss in statistical learning [10]. We use the term "surrogate loss" here because these quantities upper bound the cost-sensitive loss: $\ell(\text{argmax}_a s_a) \le \langle \ell, \phi^\gamma(s) \rangle \le \langle \ell, \psi^\gamma(s) \rangle$.[1] In the sequel, $\pi_{ramp}$ and $\pi_{hinge}$ are used by our algorithms, but do not define the benchmark policy class, since we compare directly to $L_T^\gamma$ or the surrogate loss.

**Related work.** Contextual bandit learning has been the subject of intense investigation over the past decade. The most natural categorization of these works is between parametric, realizability-based, and agnostic approaches. Parametric methods (e.g., [1, 16]) assume a (generalized) linear relationship between the losses and the contexts/actions. Realizability-based methods generalize parametric ones by assuming the losses are predictable by some abstract regression class [3, 21]. Agnostic approaches (e.g., [7, 29, 4, 38, 46, 47]) avoid realizability assumptions and instead compete with VC-type policy classes for statistical tractability. Our work contributes to all of these directions, as our margin bounds apply to the agnostic adversarial setting and yield true regret bounds under realizability assumptions.

A special case of contextual bandits is *bandit multiclass prediction*, where the loss vector is zero for one action and one for all others [26]. Several recent papers obtain surrogate regret bounds and efficient algorithms for this setting when the benchmark regressor class $\mathcal{F}$ consists of linear functions [26, 24, 9, 22]. Our work contributes to this line in two ways: our bounds and algorithms extend beyond linear/parametric classes, and we consider the more general contextual bandit setting.

Our information-theoretic results on achievability are similar in spirit those of Daniely and Halbertal [18], who derive tight generic bounds for bandit multiclass prediction in terms of the Littlestone dimension. This result is incomparable to our own: their bounds are on the $0/1$ loss regret directly rather than surrogate regret, but the Littlestone dimension is not a tight complexity measure for real-valued function classes in agnostic settings, which is our focus.

At a technical level, our work builds on several recent results. To derive achievable regret bounds, we use the adaptive minimax framework of Foster et al. [20], along with a new adaptive chaining argument to control the supremum of a martingale process [42]. Our HINGE-LMC algorithm is based on log-concave sampling [12], and it uses randomized smoothing [19] and the geometric resampling trick of Neu and Bartók [33]. We also use several ideas from classification calibration [49, 8], and, in particular, the surrogate hinge loss we work with is studied by Pires et al. [34].

## 2 Achievable regret bounds

This section provides generic surrogate regret bounds for contextual bandits in terms of the sequential metric entropy [41] of the regressor class $\mathcal{F}$. Notably, our general techniques apply when the ramp loss is used as a surrogate, and so, via Lemma 1, they yield the main result of the section—-a margin-based regret guarantee—as a special case.

To motivate our approach, consider a well-known reduction from bandits to full information online learning: If a full information algorithm achieves a regret bound in terms of the so-called *local norms* $\sum_t \langle p_t, \ell_t^2 \rangle$, then running the full information algorithm on importance-weighted losses $\hat{\ell}_t(a)$ yields an expected regret bound for the bandit setting. For example, when $\Pi$ is finite, EXP4 [7] uses HEDGE [23] as the full information algorithm, and obtains a deterministic regret bound of

$$\text{Regret}(T, \Pi) \le \frac{\eta}{2} \sum_{t=1}^{T} \mathbb{E}_{\pi \sim p_t} \langle \pi(x_t), \hat{\ell}_t \rangle^2 + \frac{\log(|\Pi|)}{\eta}, \tag{1}$$

where $\eta > 0$ is the learning rate and $p_t$ is the distribution over policies in $\Pi$ (inducing an action distribution) for round $t$. Evaluating conditional expectations and optimizing $\eta$ yields a regret bound of $\mathcal{O}(\sqrt{KT \log(|\Pi|)})$, which is optimal for contextual bandits with a finite policy class.

To use this reduction beyond the finite class case and with surrogate losses we face two challenges:

1. **Infinite classes.** The natural approach of using a pointwise (or sup-norm) cover for $\mathcal{F}$ is insufficient—not only because there are classes that have infinite pointwise covers yet are online-learnable, but also because it yields sub-optimal rates even when a finite pointwise cover is available. Instead, we establish existence of a full-information algorithm for large nonparametric classes that has 1) strong adaptivity to loss scaling as in (1) and 2) regret scaling with the sequential covering number for $\mathcal{F}$, which is the correct generalization of the empirical covering number in statistical learning to the adversarial online setting. This is achieved via non-constructive methods.

2. **Variance control**. With surrogate losses, controlling the variance/local norm term $\mathbb{E}_\pi \langle \pi(x_t), \hat{\ell}_t \rangle^2$ in the reduction from bandit to full information is more challenging, since the surrogate loss of a policy depends on the scale of the underlying regressor, not just the action it selects. To address this, we develop a new sampling scheme tailored to scale-sensitive losses.

**Full-information regret bound.**  We consider the following full information protocol, which in the sequel will be instantiated via reduction from contextual bandits. Let the context space $\mathcal{X}$ and $\mathcal{A}$ be fixed as in Subsection 1.1, and consider a function class $\mathcal{G} \subset (\mathcal{X} \to \mathcal{S})$, where $\mathcal{S} \subseteq \mathbb{R}_+^K$. The reader may think of $\mathcal{G}$ as representing $\phi^\gamma \circ \mathcal{F}$ or $\psi^\gamma \circ \mathcal{F}$, i.e. the surrogate loss composed with the regressor class, so that $\mathcal{S}$ (which is not necessarily convex) represents the image of the surrogate loss over $\mathcal{F}$.

The online learning protocol is: For time $t = 1, \ldots, T$, (1) the learner observes $x_t$ and chooses a distribution $p_t \in \Delta(\mathcal{S})$, (2) the adversary picks a loss vector $\ell_t \in \mathcal{L} \subset \mathbb{R}_+^K$, (3) the learner samples outcome $s_t \sim p_t$ and experiences loss $\langle s_t, \ell_t \rangle$. Regret against the benchmark class $\mathcal{G}$ is given by

$$\sum_{t=1}^T \mathbb{E}_{s_t \sim p_t} \langle s_t, \ell_t \rangle - \inf_{g \in \mathcal{G}} \sum_{t=1}^T \langle g(x_t), \ell_t \rangle.$$

As our complexity measure, we use a multi-output generalization of *sequential covering numbers* introduced by Rakhlin et al. [41]. Define a $\mathcal{Z}$-*valued tree* $\boldsymbol{z}$ to be a sequence of mappings $\boldsymbol{z}_t : \{\pm 1\}^{t-1} \to \mathcal{Z}$. The tree $\boldsymbol{z}$ is a complete rooted binary tree with nodes labeled by elements of $\mathcal{Z}$, where for any "path" $\epsilon \in \{\pm 1\}^T$, $\boldsymbol{z}_t(\epsilon) \triangleq \boldsymbol{z}_t(\epsilon_{1:t-1})$ is the value of the node at level $t$ on the path $\epsilon$.

**Definition 1.** *For a function class $\mathcal{G} \subset (\mathcal{X} \to \mathbb{R}^K)$ and $\mathcal{X}$-valued tree $\boldsymbol{x}$ of length $T$, the $L_\infty/\ell_\infty$ sequential covering number[2] for $\mathcal{G}$ on $\boldsymbol{x}$ at scale $\varepsilon$, denoted by $\mathcal{N}_{\infty,\infty}(\varepsilon, \mathcal{G}, \boldsymbol{x})$, is the cardinality of the smallest set $V$ of $\mathbb{R}^K$-valued trees for which*

$$\forall g \in \mathcal{G} \ \forall \epsilon \in \{\pm 1\}^T \ \exists \boldsymbol{v} \in V \text{ s.t. } \max_{t \in [T]} \|g(\boldsymbol{x}_t(\epsilon)) - \boldsymbol{v}_t(\epsilon)\|_\infty \leq \varepsilon. \tag{2}$$

*Define $\mathcal{N}_{\infty,\infty}(\varepsilon, \mathcal{G}, T) \triangleq \sup_{\boldsymbol{x}: \text{length}(\boldsymbol{x})=T} \mathcal{N}_{\infty,\infty}(\varepsilon, \mathcal{G}, \boldsymbol{x})$.*

We refer to $\log \mathcal{N}_{\infty,\infty}$ as the *sequential metric entropy*. Note that in the binary case, for learning unit $\ell_2$ norm linear functions in $d$ dimensions, the pointwise metric entropy is $O(d \log(1/\varepsilon))$, whereas the sequential metric entropy is $O(d \log(1/\varepsilon) \wedge \varepsilon^{-2} \log(d))$, leading to improved rates in high dimension.

With this definition, we can now state our main theorem for full information.

**Theorem 2.** *Assume[3] $\sup_{\ell \in \mathcal{L}} \|\ell\|_1 \leq R$ and $\sup_{s \in \mathcal{S}} \|s\|_\infty \leq B$. Fix any constants $\eta \in (0,1]$, $\lambda > 0$, and $\beta > \alpha > 0$. Then there exists an algorithm with the following deterministic regret guarantee:*

$$\sum_{t=1}^T \mathbb{E}_{s_t \sim p_t} \langle s, \ell_t \rangle - \inf_{g \in \mathcal{G}} \sum_{t=1}^T \langle g(x_t), \ell_t \rangle \leq \frac{2\eta}{RB} \sum_{t=1}^T \mathbb{E}_{s_t \sim p_t} \langle s_t, \ell_t \rangle^2 + \frac{4RB}{\eta} \log \mathcal{N}_{\infty,\infty}(\beta/2, \mathcal{G}, T) + 3e^2 \alpha \sum_{t=1}^T \|\ell_t\|_1$$

$$+ 24e \left( \frac{\lambda}{4R} \sum_{t=1}^T \|\ell_t\|_1^2 + \frac{R}{\lambda} \right) \int_\alpha^\beta \sqrt{\log \mathcal{N}_{\infty,\infty}(\varepsilon, \mathcal{G}, T)} d\varepsilon.$$

Observe that the bound involves the variance/local norms $\mathbb{E}_{s_t \sim p_t} \langle s_t, \ell_t \rangle^2$, and has a very mild explicit dependence on the loss range $R$; this can be verified by optimizing over $\eta$ and $\lambda$. This adaptivity to the loss range is crucial for our bandit reduction. Further observe that the bound contains a Dudley-type entropy integral, which is essential for obtaining sharp rates for complex nonparametric classes.

**Bandit reduction and variance control.**  To lift Theorem 2 to contextual bandits we use the following reduction: First, initialize the full information algorithm from Theorem 2 with $\mathcal{G} = \phi^\gamma \circ \mathcal{F}$. For each round $t$, receive $x_t$, and define $P_t(a) \triangleq \mathbb{E}_{s_t \sim p_t} \frac{s_t(a)}{\sum_{a' \in [K]} s_t(a')}$ where $p_t$ is the full information algorithm's distribution. Then sample $a_t \sim P_t^\mu$, observe $\ell_t(a_t)$, and pass the importance-weighted loss $\hat{\ell}_t(a)$ back to the algorithm. For the hinge loss we use the same strategy, but with $\mathcal{G} = \psi^\gamma \circ \mathcal{F}$.

The following lemma shows that this strategy leads to sufficiently small variance in the loss estimates. The definition of the action distribution $P_t^\mu(a)$ in terms of the real-valued predictions is crucial here.

**Lemma 3.** *Define a filtration $\mathcal{J}_t = \sigma((x_1, \ell_1, a_1), \ldots, (x_{t-1}, \ell_{t-1}, a_{t-1}), x_t, \ell_t)$. Then for any $\mu \in [0, 1/K]$ the importance weighting strategy above guarantees*

$$\mathbb{E}_{a_t \sim P_t^\mu} \left[ \mathbb{E}_{s_t \sim p_t} \langle s_t, \hat{\ell}_t \rangle^2 \mid \mathcal{J}_t \right] \leq \begin{cases} K, & \text{for } \mathcal{S} \subset \Delta(\mathcal{A}). \\ K^2, & \text{for } \mathcal{S} = \phi^\gamma \circ \mathcal{F}. \\ \left(1 + \frac{B}{\gamma}\right)^2 K^2, & \text{for } \mathcal{S} = \psi^\gamma \circ \mathcal{F}. \end{cases}$$

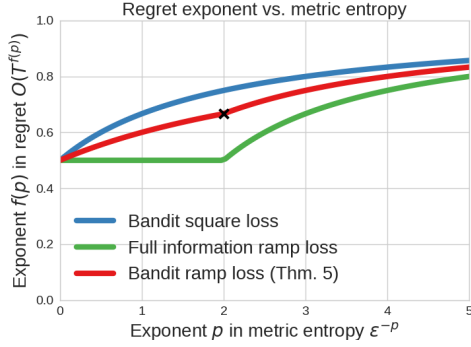

Figure 1: Regret bound exponent as a function of (sequential) metric entropy. The cross marks the point $p = 2$ where the exponent from Theorem 4 changes growth rate. "Full information" refers to the optimal rate of $T^{\frac{1}{2} \vee (\frac{p-1}{p})}$ for the same setting under full information feedback [41]. "Square loss" refers to the optimal rate of $T^{\frac{p+1}{p+2}}$ for Lipschitz contextual bandits over metric spaces of dimension $p$, which have sequential metric entropy $\varepsilon^{-p}$, under square loss realizability [44].

Theorem 2 and Lemma 3 together imply our central theorem: a chaining-based margin bound for contextual bandits, generalizing classical results in statistical learning (cf. [10]).

**Theorem 4** (Contextual bandit margin bound). *For any fixed constants $\beta > \alpha > 0$, smoothing parameter $\mu \in (0, 1)$ and margin loss parameter $\gamma > 0$ there exists an adversarial contextual bandit strategy with expected regret against the $\gamma$-margin benchmark bounded as*

$$\mathbb{E}\left[\sum_{t=1}^{T} \ell_t(a_t)\right] \leq \inf_{f \in \mathcal{F}} \mathbb{E}[L_T^\gamma(f)] + 4\sqrt{2K^2 T \log \mathcal{N}_{\infty,\infty}(\beta/2, \mathcal{F}, T)} + \mu K T \tag{3}$$

$$+ \frac{8}{\mu} \log \mathcal{N}_{\infty,\infty}(\beta/2, \mathcal{F}, T) + \frac{1}{\gamma}\left(3e^2 \alpha K T + 24e\sqrt{\frac{KT}{\mu}} \int_\alpha^\beta \sqrt{\log \mathcal{N}_{\infty,\infty}(\varepsilon, \mathcal{F}, T)} d\varepsilon\right).$$

We derive an analogous bound for the hinge loss in Appendix C. The hinge loss bound differs only through stronger dependence on scale parameters.

Before showing the implications of Theorem 4 for specific classes $\mathcal{F}$ we state a coarse upper bound in terms of the growth rate for the sequential metric entropy.

**Proposition 5.** *Suppose that $\mathcal{F}$ has sequential metric entropy growth $\log \mathcal{N}_{\infty,\infty}(\varepsilon, \mathcal{F}, T) \propto \varepsilon^{-p}$ for some $p > 0$ (nonparametric case), or that $\log \mathcal{N}_{\infty,\infty}(\varepsilon, \mathcal{F}, T) \propto d \log(1/\varepsilon)$ (parametric case). Then there exists a contextual bandit strategy with the following regret guarantee:*

$$\mathbb{E}\left[\sum_{t=1}^{T} \ell_t(a_t)\right] \leq \inf_{f \in \mathcal{F}} \mathbb{E}\left[L_T^\gamma(f)\right] + \begin{cases} O(K\sqrt{dT \log(KT/\gamma)}), & \text{parametric case.} \\ \tilde{O}((KT)^{\frac{p+2}{p+4}} \gamma^{-\frac{2p}{p+4}}), & \text{nonparametric w/ } p \leq 2. \\ \tilde{O}((KT)^{\frac{p}{p+1}} \gamma^{-\frac{p}{p+1}}), & \text{nonparametric w/ } p \geq 2. \end{cases} \tag{4}$$

Proposition 5 recovers the parametric rate of $\sqrt{dT}$ seen with e.g., LINUCB [16] but is most interesting for complex classes. The rate exhibits a phase change between the "moderate complexity" regime of $p \in (0, 2]$ and the "high complexity" regime of $p \geq 2$. This is visualized in Figure 1.

**Remark 1.** *Under i.i.d. losses and hinge/ramp loss realizability, the standard tools of classification calibration [8] can be used to deduce a proper policy regret bound from (3). However, these realizability assumptions are somewhat non-standard, and moreover if one imposes the stronger assumption of a hard margin it is possible to derive improved rates [18]. See also Appendix B.*

**Remark 2.** *Classical margin bounds typically hold for all values of $\gamma$ simultaneously, but Theorem 4 requires that $\gamma$ is chosen in advance. Learning the best value of $\gamma$ online appears challenging.*

**Rates for specific classes.** We now instantiate our results for concrete classes of interest.

**Example 1** (Finite classes). *In the finite class case there is an algorithm with $O\left(K\sqrt{T \log|\mathcal{F}|}\right)$ margin regret. When $\Pi \subset (\mathcal{X} \to \mathcal{A})$ is a finite policy class, directly reducing to Theorem 2 yields the optimal $O\left(\sqrt{KT \log|\Pi|}\right)$ policy regret, hinting at the optimality of our approach.*

**Example 2** (Lipschitz CB). *The class of Lipschitz functions over $[0, 1]^p$ admits a sequential cover with metric entropy $\tilde{O}(\varepsilon^{-p})$, so Proposition 5 implies an $\tilde{O}(T^{\frac{p+2}{p+4} \vee \frac{p}{p+1}})$ regret bound. Since our proof goes through Lemma 1, it also yields a policy regret bound against the $\pi_{ramp}(\cdot)$ class. Therefore, this result is directly comparable to the $\tilde{O}(T^{\frac{p+1}{p+2}})$ bound of Cesa-Bianchi et al. [14], applied to the $\pi_{ramp}$ policy class. Our bound achieves a smaller exponent for all values of $p$ (see Figure 1).*

Learnability in full information online learning is known to be characterized entirely by the *sequential Rademacher complexity* of the hypothesis class [41], and tight bounds on this quantity are known for standard classes including linear predictors, decision trees, and neural networks. The next example, a corollary of Theorem 4, bounds contextual bandit margin regret in terms of sequential Rademacher complexity, which is defined for any scalar-valued function class $\mathcal{G} \subseteq (\mathcal{X} \to \mathbb{R})$ as:

$$\mathfrak{R}(\mathcal{G}, T) \triangleq \sup_{\boldsymbol{x}} \mathbb{E}_{\epsilon} \sup_{g \in \mathcal{G}} \sum_{t=1}^{T} \epsilon_t g(\boldsymbol{x}_t(\epsilon)).$$

**Example 3.** *Let $\mathcal{F}|_a \triangleq \{x \mapsto f(x)_a \mid f \in \mathcal{F}\}$ be the scalar restriction of $\mathcal{F}$ to output coordinate $a$ and suppose that $\max_{a \in [K]} \mathfrak{R}(\mathcal{F}|_a, T) \geq 1$ and $B \leq 1$.[4] Then there exists an adversarial contextual bandit algorithm with margin regret bound $\tilde{O}\left(\max_a K(\mathfrak{R}(\mathcal{F}|_a, T)/\gamma)^{2/3} T^{1/3}\right).$*

Thus, for margin-based contextual bandits, full information learnability is equivalent to bandit learnability. Since the optimal regret in full information is $\Omega(\max_a \mathfrak{R}(\mathcal{F}|_a, T))$, it further shows that the price of bandit information is at most $\tilde{O}\left(\max_a K(T/\mathfrak{R}(\mathcal{F}|_a, T))^{1/3}\right)$. Note that while this bound is fairly user-friendly, it yields worse rates than Proposition 5 when translated to sequential metric entropy, except when $p = 2$ [40]. For comparison, Rakhlin et al. [41] obtain $\tilde{O}(\mathfrak{R}(\mathcal{F}, T)/\gamma)$ margin regret in full information for binary classification. For partial information, BISTRO [38] has an $O(\sqrt{KT\mathfrak{R}(\Pi, T)})$ policy regret bound, which involves the *policy* complexity and a worse $T$ dependence than our bound, but our bound (in terms of $\mathcal{F}$) applies only to the margin regret. A similar discussion applies to Theorem 4.4 of Lykouris et al. [31].

Instantiating Example 3 with linear classes generalizes the $O(T^{2/3})$ dimension-independent guarantee of BANDITRON [26] from Euclidean geometry to arbitrary uniformly convex Banach spaces, essentially the largest linear class for which online learning is possible [45]. The result also generalizes BANDITRON from multiclass to general contextual bandits and strengthens it from hinge loss to ramp loss. Note that many subsequent works [2, 9, 22] obtain dimension-dependent $O(\sqrt{dT})$ bounds for bandit multiclass prediction, as we will in the next section, but, none have explored dimension-independent $O(T^{2/3})$-type rates, which are more appropriate for high-dimensional settings.

**Example 4.** *Let $\mathcal{X}$ be the unit ball in a Banach space $(\mathfrak{B}, \|\cdot\|)$, and let $\mathcal{F}$ be induced by stacking $K-1$ linear predictors[5] each in the unit ball of the dual space $(\mathfrak{B}^\star, \|\cdot\|_\star)$. Suppose that $\|\cdot\|$ has martingale type 2 [35], which means there exists $\Psi : \mathfrak{B} \to \mathbb{R}$ such that $\frac{1}{2}\|x\|^2 \leq \Psi(x)$ and $\Psi$ is $\beta$-smooth w.r.t. $\|\cdot\|$.[6] Then there exists a contextual bandit strategy with margin regret $O(K(T/\gamma)^{2/3}).$*

Beyond linear classes, we also obtain $\tilde{O}(K(T/\gamma)^{2/3})$ margin regret when each $\mathcal{F}_a$ is a class of neural networks with weights in each layer bounded in the $(1, \infty)$ group norm, or when each $\mathcal{F}_a$ is a class of bounded depth decision trees on finitely many decision functions. These results follow by appealing to the existing sequential Rademacher complexity bounds derived in Rakhlin et al. [41].

As our last example, we consider $\ell_p$ spaces for $p < 2$. These spaces fail to satisfy martingale type 2 in a dimension-independent fashion, but they do satisfy martingale type $p$ without dimension dependence, and so have sequential metric entropy of order $\varepsilon^{-\frac{p}{p-1}}$ [39]. Moreover, in $\mathbb{R}^d$ the $\ell_p$ spaces admit a pointwise cover with metric entropy $O(d \log(1/\varepsilon))$, leading to the following dichotomy.

**Example 5.** *Consider the setting of Example 4, with $(\mathfrak{B}, \|\cdot\|) = (\mathbb{R}^d, \|\cdot\|_p)$ for $p \leq 2$. Then there exists a contextual bandit strategy with margin regret $\tilde{O}(K(T/\gamma)^{\frac{p}{2p-1}} \wedge K\sqrt{dT \log(KT/\gamma)}).$*

## 3 Efficient Algorithms

We derive two new algorithms for contextual bandits using the hinge loss $\psi^\gamma$. The first algorithm, HINGE-LMC, focuses on the parametric setting; it is based on a continuous version of exponential

| **Algorithm 1** HINGE-LMC | **Algorithm 2** Langevin Monte Carlo (LMC) |
|---|---|
| Input: Class $\Theta$, learning rate $\eta$, margin parameter $\gamma$. | Input: $F(\cdot)$, parameters $m, u, \lambda, N, \alpha$. |
| Define $w_0(\theta) \triangleq 1$ for all $\theta \in \Theta$. | // Parameter choices are in Appendix D. |
| **for** $t = 1, \ldots, T$ **do** | Set $\tilde{\theta}_0 \leftarrow 0 \in \mathbb{R}^d$ |
| $\quad$ Receive $x_t$, set $\theta_t \leftarrow \text{LMC}(\eta w_{t-1})$. | **for** $k = 1, \ldots, N$ **do** |
| $\quad$ Set $p_t(\cdot; \theta_t) \propto \psi^\gamma(f(x_t; \theta_t))$. | $\quad$ Draw $z_1, \ldots, z_m \overset{iid}{\sim} \mathcal{N}(0, u^2 I_d)$. Define |
| $\quad$ Set $p_t^\mu(\cdot; \theta_t) \triangleq (1 - K\mu)p_t + \mu$. | $\quad \tilde{F}_k(\theta) \triangleq \frac{1}{m} \sum_{i=1}^m F(\theta + z_i) + \frac{\lambda}{2}\|\theta\|_2^2$ |
| $\quad$ Play $a_t \sim p_t^\mu(\cdot; \theta_t)$, observe $\ell_t(a_t)$. | |
| $\quad$ // Geometric resampling. | $\quad$ Draw $\xi_k \sim \mathcal{N}(0, I_d)$ and update |
| $\quad$ **for** $m = 1, \ldots, M$ **do** | |
| $\quad\quad \tilde{\theta}_t \leftarrow \text{LMC}(\eta w_{t-1})$. | $\quad \tilde{\theta}_k \leftarrow \mathcal{P}_\Theta\left(\tilde{\theta}_{k-1} - \frac{\alpha}{2}\nabla \tilde{F}_k(\tilde{\theta}_{k-1}) + \sqrt{\alpha}\xi_k\right).$ |
| $\quad\quad$ Sample $\tilde{a}_t \sim p_t^\mu(\cdot; \tilde{\theta}_t)$, if $\tilde{a}_t = a_t$, break. | |
| $\quad$ **end for** | **end for** |
| $\quad$ Set $m_t = m$, and $\tilde{\ell}_t(a) \triangleq \ell_t(a_t) \cdot m_t \mathbf{1}\{a_t = a\}$. | Return $\tilde{\theta}_N$. |
| $\quad$ Update $w_t(\theta) \leftarrow w_{t-1}(\theta) + \langle \tilde{\ell}_t, \psi^\gamma(f(x_t; \theta))\rangle$. | |
| **end for** | |

weights using a log-concave sampler. The second, SMOOTHFTL, is simply Follow-The-Leader with uniform smoothing. SMOOTHFTL applies to the stochastic setting with classes that have "high complexity" in the sense of Proposition 5.

## 3.1 Hinge-LMC

For this section, we identify $\mathcal{F}$ with a compact convex set $\Theta \subset \mathbb{R}^d$, using the notation $f(x; \theta) \in \mathbb{R}_{=0}^K$ to describe the parametrized function. We assume that $\psi^\gamma(f(x; \theta)_a)$ is convex in $\theta$ for each $(x, a)$ pair, $\sup_{x,\theta}\|f(x; \theta)\|_\infty \leq B$, $f(x; \cdot)_a$ is $L$-Lipschitz in $\theta$ with respect to the $\ell_2$ norm, and that $\Theta$ contains the centered Euclidean ball of radius $1$ and is contained within a Euclidean ball of radius $R$. These assumptions are satisfied when $\mathcal{F}$ is a linear class, under appropriate boundedness conditions.

The pseudocode for HINGE-LMC is displayed in Algorithm 1, and all parameters settings are given in Appendix D. The algorithm is a continuous variant of exponential weights [7], where at round $t$, we define the exponential weights distribution via its density (w.r.t. the Lebesgue measure over $\Theta$):

$$P_t(\theta) \propto \exp(-\eta w_{t-1}(\theta)), \qquad w_{t-1}(\theta) \triangleq \sum_{s=1}^{t-1}\langle \tilde{\ell}_s, \psi^\gamma(f(x_s; \theta))\rangle,$$

where $\eta$ is a learning rate and $\tilde{\ell}_s$ is a loss vector estimate. At a high level, at each iteration the algorithm samples $\theta_t \sim P_t$, then samples the action $a_t$ from the induced policy distribution $p_t(\cdot; \theta) = \pi_{\text{hinge}}(f(x_t; \theta_t))$, appropriately smoothed. The algorithm plays $a_t$ and constructs a loss estimate $\tilde{\ell}_t \triangleq m_t \cdot \ell_t(a)\mathbf{1}\{a = a_t\}$, where $m_t$ is an approximate importance weight computed by repeatedly sampling from $P_t$. This vector $\tilde{\ell}_t$ is passed to exponential weights to define the distribution at the next round. To sample from $P_t$ we use Projected Langevin Monte Carlo (LMC), displayed in Algorithm 2.

The algorithm has many important subtleties. Apart from passing to the hinge surrogate loss to obtain a tractable log-concave sampling problem, by using the induced policy distribution $\pi_{\text{hinge}}(\cdot)$, we are also able to control the local norm term in the exponential weights regret bound.[7] Then, the analysis for Projected LMC [12] requires a smooth potential function, which we obtain by convolving with the gaussian density, also known as randomized smoothing [19]. We also use $\ell_2$ regularization for strong convexity and to overcome sampling errors introduced by randomized smoothing. Finally, we use the geometric resampling technique [33] to approximate the importance weight by repeated sampling.

Here, we state the main guarantee and its consequences. A more complete theorem statement, with exact parameter specifications and the precise running time is provided in Appendix D as Theorem 18.

**Theorem 6** (Informal). *Under the assumptions of Subsection 3.1,* HINGE-LMC *with appropriate parameter settings runs in time* $poly(T, d, B, K, \frac{1}{\gamma}, R, L)$ *and guarantees*

$$\mathbb{E}\sum_{t=1}^{T}\ell_t(a_t) \le \inf_{\theta \in \Theta}\frac{1}{K}\mathbb{E}\sum_{t=1}^{T}\langle \ell_t, \psi^{\gamma}(f(x_t;\theta))\rangle + \tilde{\mathcal{O}}\left(\frac{B}{\gamma}\sqrt{dT}\right).$$

Since bandit multiclass prediction is a special case of contextual bandits, Theorem 6 immediately implies a $\sqrt{dT}$-mistake bound for this setting. See Appendix B for more discussion.

**Corollary 7** (Bandit multiclass). *In the bandit multiclass setting, Algorithm 1 enjoys a mistake bound of* $\tilde{\mathcal{O}}((B/\gamma)\sqrt{dT})$ *against the cost-sensitive $\gamma$-hinge loss and runs in polynomial time.*

Additionally, under a realizability condition for the hinge loss, we obtain a standard regret bound. For simplicity in defining the condition, assume that for every $(x, \ell)$ pair, $\ell$ is a random variable with conditional mean $\bar{\ell}$ (chosen by the adversary) and $\bar{\ell}$ has a unique action with minimal loss.

**Corollary 8** (Realizable bound). *In addition to the conditions above, assume that there exists $\theta^{\star} \in \Theta$ such that for every $(x, \ell)$ pair and for all $a \in \mathcal{A}$, we have $f(x; \theta^{\star})_a \triangleq K\gamma \mathbf{1}\{\bar{\ell}(a) \le \min_{a'} \bar{\ell}(a')\} - \gamma$. Then* HINGE-LMC *runs in polynomial time and guarantees*

$$\sum_{t=1}^{T}\mathbb{E}\bar{\ell}_t(a_t) \le \sum_{t=1}^{T}\mathbb{E}\min_{a}\bar{\ell}(a) + \tilde{\mathcal{O}}\left(\frac{B}{\gamma}\sqrt{dT}\right).$$

A few comments are in order:

1.  The use of LMC for sampling is not strictly necessary. Other log-concave samplers do exist for non-smooth potentials [30], which will remove the parameters $m, u, \lambda$, significantly simplify the algorithm, and even lead to a better run-time guarantee using current theory. However, we prefer to use LMC due to its success in Bayesian inference and deep learning, and its connections to incremental optimization methods. Note that more recent results in slightly different settings [36, 17, 15] suggest that it may be possible to substantially improve upon the LMC analysis that we use and even extend it to non-convex settings. We are hopeful that the LMC approach will lead to a practically useful contextual bandit algorithm and plan to explore this direction further.

2.  Corollary 7 provides a new solution to the open problem of Abernethy and Rakhlin [2]. In fact, it is the first efficient $\sqrt{dT}$-type regret bound against a hinge loss benchmark, although our loss is slightly different from the multiclass hinge loss used by Kakade et al. [26] in their $T^{2/3}$-regret BANDITRON algorithm (which motivated the open problem). All prior $\sqrt{dT}$-regret algorithms [24, 9, 22] use losses with curvature such as the multiclass logistic loss or the squared hinge loss. See Appendix B for a comparison between cost-sensitive and multiclass hinge losses.

3.  In Corollary 8, regret is measured relative to the policy that chooses the best action (in expectation) on *every round*. As in prior results [1, 3], this is possible because the realizability condition ensures that this policy is in our class. Note that here, a requirement for realizability is that $B \ge K\gamma$, and hence the dependence on $K$ is implicit and in fact slightly worse than the optimal rate [16].

4.  For Corollary 8, the best points of comparisons are methods based on square-loss realizability [3, 21], although our condition is different. Compared with LINUCB and variants [16, 1] specialized to $\ell_2/\ell_2$ geometry, our assumptions are somewhat weaker but these methods have slightly better guarantees for linear classes.[8] Compared with Foster et al. [21], which is the only other efficient approach at a comparable level of generality, our assumptions on the regressor class are stronger, but we obtain better guarantees, in particular removing distribution-dependent parameters.

To summarize, HINGE-LMC is the first efficient $\sqrt{dT}$-regret algorithm for bandit multiclass prediction using the hinge loss. It also represents a new approach to adversarial contextual bandits, yielding $\sqrt{dT}$ policy regret under hinge-based realizability. Finally, while we lose the theoretical guarantees, the algorithm easily extends to non-convex classes, which we expect to be practically effective.

## 3.2 SMOOTHFTL

A drawback of HINGE-LMC is that it only applies in the parametric regime. We now introduce an efficient (in terms of queries to a hinge loss minimization oracle) algorithm with a regret bound similar to Theorem 4, but in the stochastic setting, where $\{(x_t, \ell_t)\}_{t=1}^{T}$ are drawn i.i.d. from some joint distribution $\mathcal{D}$ over $\mathcal{X} \times \mathbb{R}_{+}^{K}$. Here we return to the abstract setting with regression class $\mathcal{F}$, and for simplicity, we assume $B = 1$.

The algorithm we analyze is simply Follow-The-Leader with uniform smoothing and epoching, which we refer to as SMOOTHFTL. We use an epoch schedule where the $m^{\text{th}}$ epoch lasts for $n_m \triangleq 2^m$ rounds (starting with $m = 0$). At the beginning of the $m^{\text{th}}$ epoch, we compute the empirical importance weighted hinge-loss minimizer $\hat{f}_{m-1}$ using *only* the data from the previous epoch. That is, we set

$$\hat{f}_{m-1} \triangleq \underset{f \in \mathcal{F}}{\operatorname{argmin}} \sum_{\tau = n_{m-1}}^{n_m - 1} \langle \hat{\ell}_\tau, \psi^\gamma(f(x_\tau)) \rangle.$$

Then, for each round $t$ in the $m^{\text{th}}$ epoch, we sample $a_t$ from $p_t \triangleq (1 - K\mu)\pi_{\text{hinge}}(\hat{f}_{m-1}(x_t)) + \mu$. The parameter $\mu \in (0, 1/K]$ controls the smoothing. At time $t = 1$ we simply take $p_1$ to be uniform.

**Theorem 9.** *Suppose that $\mathcal{F}$ satisfies* $\log \mathcal{N}_{\infty,\infty}(\varepsilon, \mathcal{F}, T) \propto \varepsilon^{-p}$ *for some $p > 2$. Then in the stochastic setting, with $\mu = K^{-1} T^{\frac{-1}{p+1}}$,* SMOOTHFTL *enjoys the following expected regret guarantee*[9]

$$\sum_{t=1}^{T} \mathbb{E}\ell_t(a_t) \le \inf_{f \in \mathcal{F}} \frac{T}{K} \mathbb{E}\langle \ell, \psi^\gamma(f(x)) \rangle + \tilde{O}\left( (T/\gamma)^{\frac{p}{p+1}} \right).$$

This provides an algorithmic counterpart to Proposition 5 in the $p \ge 2$ regime. The algorithm is quite similar to EPOCH-GREEDY [29], and the main contribution here is to provide a careful analysis for large function classes. We leave obtaining an oracle-efficient algorithm that matches Proposition 5 in the regime $p \in (0, 2)$ as an open problem.

A similar bound can be obtained for the ramp loss by simply replacing the hinge loss ERM. We analyze the hinge loss version because standard (e.g. linear) classes admit efficient hinge loss minimization oracles. Interestingly, the bound in Theorem 9 actually improves on Proposition 5, in that it is independent of $K$. This is due to the scaling of the hinge loss in Lemma 1.

In Appendix F, we extend the analysis to the stochastic Lipschitz contextual bandit setting. Here, instead of measuring regret against the benchmark $\psi^\gamma \circ \mathcal{F}$ we compare to the class of all 1-Lipschitz functions from $\mathcal{X}$ to $\Delta(\mathcal{A})$, where $\mathcal{X}$ is a metric space of bounded covering dimension. We show that SMOOTHFTL achieves $T^{\frac{p}{p+1}}$ regret with a $p$-dimensional context space and finite action space. This improves on the $T^{\frac{p+1}{p+2}}$ bound of Cesa-Bianchi et al. [14], as in Example 2, yet the best available lower bound is $T^{\frac{p-1}{p}}$ [25]. Closing this gap remains an intriguing open problem.

## 4 Discussion

This paper initiates a study of the utility of surrogate losses in contextual bandit learning. We obtain new margin-based regret bounds in terms of sequential complexity notions on the benchmark class, improving on the best known rates for Lipschitz contextual bandits and providing dimension-independent bounds for linear classes. On the algorithmic side, we provide the first solution to the open problem of Abernethy and Rakhlin [2] with a non-curved loss and we also show that Follow-the-Leader with uniform smoothing performs well in nonparametric settings.

Yet, several open problems remain. First, our bounds in Section 2 are likely suboptimal in the dependence on $K$, and improving this is a natural direction. Other questions involve deriving stronger lower bounds (e.g., for the non-parametric setting) and adapting to the margin parameter. We also hope to experiment with HINGE-LMC, and develop a better understanding of computational-statistical tradeoffs with surrogate losses. We look forward to studying these questions in future work.

**Acknowledgements.** We thank Haipeng Luo, Karthik Sridharan, Chen-Yu Wei, and Chicheng Zhang for several helpful discussions. D.F. acknowledges the support of the NDSEG PhD fellowship and Facebook PhD fellowship.

## Footnotes

[1] On a related note, the information-theoretic results we present are also compatible with the surrogate function $\theta^\gamma(s)_a := \max\{1 + (s_a - \max_{a'} s_{a'})/\gamma, 0\}$, which also satisfies $\ell(\text{argmax}_a s_a) \le \langle \ell, \theta^\gamma(s) \rangle$. This leads to a perhaps more standard notion of multiclass margin bound but does not lead to efficient algorithms.

[2]Sequential coverings for $L_p/\ell_q$ can be defined similarly, but do not appear in the present paper.

[3]Measuring loss in $\ell_1$ may seem restrictive, but it is natural when working with the 1-sparse importance-weighted losses, and it enables us to cover the output space in $\ell_\infty$ norm.

[4]This restriction serves only to simplify calculations and can be relaxed.

[5]Only $K-1$ predictors are needed due to the sum-to-zero constraint of $\mathbb{R}^K_{=0}$.

[6]Norms that satisfy this property with dimension-independent or logarithmic constants include $\ell_p$ for all $p \geq 2$, Schatten $S_p$ norms for $p \geq 2$ (including the spectral norm), and $(2, p)$ group norms for $p \geq 2$ [27, 28].

[7]This seems specialized to surrogates that can be expressed as an inner product between the loss vector and (a transformation of) the prediction, so it does not apply to standard loss functions in bandit multiclass prediction.

[8]In the abstract linear setting we take $\mathcal{F}$ to be the set of linear functions in the ball for some norm $\|\cdot\|$ and contexts to be bounded in the dual norm $\|\cdot\|_{\star}$. The runtime of HINGE-LMC will degrade (polynomially) with the ratio $\|\theta\|/\|\theta\|_2$, but the regret bound is the same for any such norm pair.

[9]This result is stated in terms of the sequential cover $\mathcal{N}_{\infty,\infty}$ to avoid additional definitions, but can easily be improved to depend on the classical (worst-case) covering number seen in statistical learning.

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
