[Supplementary Material]

## A    Calibration Lemmas

**Proof of Lemma 1.**  We start with the ramp loss. First since $s \in \mathbb{R}_{=0}^K$, we know that the normalization term in $\pi_{\mathrm{ramp}}(s)$ is

$$\sum_{a \in \mathcal{A}} \phi^\gamma(s_a) \geq 1,$$

from which the first inequality follows. The second inequality follows from the fact that $s_a \leq -\gamma$ implies that $\pi_{\mathrm{ramp}}(s)_a = 0$, along with the trivial fact that $\pi_{\mathrm{ramp}}(s)_a \leq 1$.

The hinge loss claim is also straightforward, since here the normalization is

$$\sum_{a \in \mathcal{A}} \psi^\gamma(s_a) = \sum_{a \in \mathcal{A}} \max\{1 + s_a/\gamma, 0\} \geq \sum_a 1 + \frac{s_a}{\gamma} \geq K. \qquad \square$$

**Lemma 10** (Hinge loss realizability). *Let $\ell \in \mathbb{R}_+^K$ and let $a^\star = \operatorname{argmin}_{a \in \mathcal{A}} \ell_a$. Define $s \in \mathbb{R}_{=0}^K$ via $s_a \triangleq K\gamma \mathbf{1}\{a = a^\star\} - \gamma$. Then we have*

$$\langle \ell, \psi^\gamma(s) \rangle = K \langle \ell, \pi_{hinge}(s) \rangle = K\ell_{a^\star}.$$

**Proof.**  For this particular $s$, the normalizing constant in the definition of $\pi_{\mathrm{hinge}}$ is

$$\sum_{a \in \mathcal{A}} \max\left(1 + \frac{K\gamma \mathbf{1}\{a = a^\star\} - \gamma}{\gamma}, 0\right) = K,$$

and so the first equality follows. The second equality is also straightforward since the score for every action except $a^\star$ is clamped to zero. $\qquad \square$

**Proof of Lemma 3.**
For the case when $\mathcal{S} \subset \Delta(\mathcal{A})$, this claim is a well-known property of importance weighting:

$$\mathbb{E}\left[\mathbb{E}_{s_t \sim p_t}\langle s_t, \hat{\ell}_t \rangle^2 \mid \mathcal{J}_t\right] = \sum_{a \in [K]} P_t^\mu(a) \frac{\mathbb{E}_{s_t \sim p_t} \ell_t^2(a) s_t^2(a)}{(P_t^\mu(a))^2} \leq \sum_{a \in [K]} \frac{\mathbb{E}_{s_t \sim p_t} s_t^2(a)}{P_t^\mu(a)}$$

$$\leq \sum_{a \in [K]} \frac{\mathbb{E}_{s_t \sim p_t} s_t(a)}{P_t^\mu(a)} = \sum_{a \in [K]} \frac{P_t(a)}{(1 - K\mu)P_t(a) + \mu}.$$

Here we use Hölder's inequality twice, using that $\|\ell\|_\infty \leq 1$ and $s \in \Delta(\mathcal{A})$. Now, since the function $x \mapsto 1/(1 - K\mu + \mu/x)$ is concave in $x$, it follows that

$$\sum_{a \in [K]} \frac{P_t(a)}{(1 - K\mu)P_t(a) + \mu} = \sum_{a \in [K]} \frac{1}{(1 - K\mu) + \mu/P_t(a)} \leq K \frac{1}{(1 - K\mu) + K\mu/\sum_{a \in [K]} P_t(a)} = K,$$

which proves the claim for $\mathcal{S} \subset \Delta(\mathcal{A})$.

We proceed in the same fashion for both the ramp and hinge loss. Recall the definition $P_t^\mu(a) = (1 - K\mu)\mathbb{E}_{s_t \sim p_t} \frac{s_t(a)}{\sum_{a' \in [K]} s_t(a')} + \mu$. We have

$$\mathbb{E}\left[\mathbb{E}_{s_t \sim p_t}\langle s_t, \hat{\ell}_t \rangle^2 \mid \mathcal{J}_t\right] = \sum_{a \in [K]} P_t^\mu(a) \frac{\mathbb{E}_{s_t \sim p_t} \ell_t^2(a) s_t^2(a)}{(P_t^\mu(a))^2} = \sum_{a \in [K]} \frac{\mathbb{E}_{s_t \sim p_t} \ell_t^2(a) s_t^2(a)}{P_t^\mu(a)}$$

$$\leq \sum_{a \in [K]} \frac{\mathbb{E}_{s_t \sim p_t} s_t^2(a)}{P_t^\mu(a)} \leq \max_{a \in [K]} \max_{s \in \mathcal{S}} s(a) \cdot \sum_{a \in [K]} \frac{\mathbb{E}_{s_t \sim p_t} s_t(a)}{P_t^\mu(a)}$$

$$= \max_{a \in [K]} \max_{s \in \mathcal{S}} s(a) \cdot \sum_{a \in [K]} \frac{\mathbb{E}_{s_t \sim p_t} s_t(a)}{(1 - \mu K)\mathbb{E}_{s_t \sim p_t} \frac{s_t(a)}{\sum_{a' \in [K]} s_t(a')} + \mu}$$

$$\leq K \cdot \left(\max_{a \in [K]} \max_{s \in \mathcal{S}} s(a)\right)^2 \cdot \sum_{a \in [K]} \frac{\mathbb{E}_{s_t \sim p_t} \frac{s_t(a)}{\sum_{a' \in [K]} s_t(a')}}{(1 - K\mu)\mathbb{E}_{s_t \sim p_t} \frac{s_t(a)}{\sum_{a' \in [K]} s_t(a')} + \mu}.$$

Here we first apply the definition of $\hat{\ell}_t$ and cancel out one factor of $P_t^\mu$ in the denomator. Then we apply Hölder's inequality, using that $s_t(a) \geq 0$. Expanding the definition $P_t^\mu$ and using the upper bound $\sum_{a' \in [K]} s_t(a') \leq K \max_a \max_s s_t(a)$, yields the final expression.

Now, let $q_a \triangleq \mathbb{E}_{s_t \sim p_t} \frac{s_t(a)}{\sum_{a' \in [K]} s_t(a')}$, and apply the concavity argument above. This yields

$$K^2 \cdot \left( \max_{a \in [K]} \max_{s \in \mathcal{S}} s(a) \right)^2.$$

For the set $\mathcal{S}$ induced by the ramp loss we have $\max_{a \in [K]} \max_{s \in \mathcal{S}} s(a) \leq 1$, and for the set $\mathcal{S}$ induced by the hinge loss we have $\max_{a \in [K]} \max_{s \in \mathcal{S}} s(a) \leq (1 + \frac{B}{\gamma})$. □

# B  Comparing Multiclass Loss Functions and Notions of Realizability

While our surrogate loss functions apply to general cost-sensitive classification, when specialized to the multiclass zero-one feedback, as in bandit multiclass prediction, they are somewhat non-standard. In this appendix we provide a discussion of the differences, focusing on the hinge loss.

Let us detail the multiclass setting: On each round, the adversary chooses a pair $(x, y^\star)$ where $x \in \mathcal{X}$, $y^\star \in \mathcal{A}$ and shows $x$ to the learner. The learner then makes a prediction $\hat{y}$. The 0/1-loss for the learner is $\mathbf{1}\{\hat{y} \neq y\}$. Using a class of regression functions $\mathcal{G} \subset (\mathcal{X} \to \mathbb{R}^K)$, the standard multiclass hinge loss for a regressor $g \in \mathcal{G}$ is:

$$\ell^\gamma_{\text{MC-hinge}}(g, (x, y^\star)) \triangleq \max\{1 - \gamma^{-1}(g(x)_{y^\star} - \max_{y \neq y^\star} g(x)_y), 0\}.$$

On the other hand, for our results we assume that the regressor class $\mathcal{F} \subset (\mathcal{X} \to \mathbb{R}^K_{=0})$, and the cost-sensitive hinge loss that we use here is:

$$\ell^\gamma_{\text{CC-hinge}}(f, (x, y^\star)) \triangleq \sum_{y \neq y^\star} \psi^\gamma(f(x)_y) = \sum_{y \neq y^\star} \max\{1 + \gamma^{-1} f(x)_y, 0\}.$$

More precisely in Corollary 7, we are measuring the benchmark using $\ell_{\text{CC-hinge}}$ and our bound is

$$\mathbb{E} \sum_{t=1}^T \ell_t(a_t) \leq \inf_{\theta \in \Theta} \frac{1}{K} \sum_{t=1}^T \ell^\gamma_{\text{CC-hinge}}(f(\cdot; \theta), (x_t, y_t^\star)) + \tilde{O}\left( \frac{B}{\gamma} \sqrt{dT} \right).$$

On the other hand, the open problem of Abernethy and Rakhlin [2] asks for a $\sqrt{dT}$ bound when the benchmark is measured using $\ell_{\text{MC-hinge}}$. As we will see, the two loss functions are somewhat different.

Let us first standardize the function classes. By rebinding $f_g(x)_y \triangleq g(x)_y - K^{-1} \sum_{y'} g(x)_{y'}$ we can easily construct a "sum-to-zero" class from an unconstrained class $\mathcal{G} \subset (\mathcal{X} \to \mathbb{R}^K)$, and with this definition, the cost-sensitive hinge loss for any function $g \in \mathcal{G}$ is:

$$\ell^\gamma_{\text{CC-hinge}}(g, (x, y^\star)) \triangleq \sum_{y \neq y^\star} \psi^\gamma(f_g(x)_y) = \sum_{y \neq y^\star} \max\{1 - \gamma^{-1}(K^{-1} \sum_{y'} g(x)_{y'} - g(x)_y), 0\}.$$

The main proposition in this appendix is that if the cost-sensitive hinge loss is zero, then so is the multiclass hinge loss, while the converse is not true.

**Proposition 11.** *We have the following implication*

$$\ell^\gamma_{\text{CC-hinge}}(g, (x, y^\star)) = 0 \Rightarrow \ell^{K\gamma}_{\text{MC-hinge}}(g, (x, y^\star)) = 0.$$

*The converse does not hold: For any $\gamma, \tilde{\gamma} > 0$ and $K \geq 0$ there exists a function $g$ and an $(x, y^\star)$ pair such that $\ell^\gamma_{\text{MC-hinge}}(g, (x, y^\star)) = 0$ but $\ell^{\tilde{\gamma}}_{\text{CC-hinge}}(g, (x, y^\star)) \geq 1$.*

Note that $\ell^{\gamma_1}_{\text{MC-hinge}} \geq \ell^{\gamma_2}_{\text{MC-hinge}}$ whenever $\gamma_1 \geq \gamma_2$, so the first implication also holds when the RHS is replaced with $\ell^\gamma_{\text{MC-hinge}}$.

The proposition implies that cost-sensitive hinge realizability — that there exists a predictor $g^\star$ such that $\ell_{\text{CC-hinge}}(g^\star, (x, y^\star)) = 0$ for all rounds — is a *strictly stronger* condition than multiclass hinge realizability. Under multiclass separability assumptions [26], the bandit multiclass surrogate

benchmark is zero, while our cost-sensitive benchmark may still be large, and so the cost-sensitive translation approach cannot be used to obtain a sublinear upper bound on the number of mistakes made by the learner. In this vein, HINGE-LMC does not completely resolve the open problem of Abernethy and Rakhlin [2], since the loss function we use can result in a weaker bound than desired. Note that the cost-sensitive surrogate loss may also prevent us from exploiting *small-loss* structure in the multiclass surrogate to obtain fast rates.

On the other hand, our cost-sensitive surrogate losses are applicable in a much wider range of problems, and cost-sensitive structure is common in contextual bandit settings. As such, we believe that designing algorithms for this more general setting is valuable.

**Proof of Proposition 11.** If $\ell_{\text{CC-hinge}}^{\gamma}(g,(x,y^{\star})) = 0$ then we know that for all $y \neq y^{\star}$, we must have

$$\forall y \neq y^{\star} : \frac{1}{K}\sum_{y'} g(x)_{y'} - g(x)_y \geq \gamma.$$

Adding these inequalities together for $y \neq y^{\star}$ and subtracting $g(x)_{y^{\star}}$ from both sides we get

$$\frac{K-1}{K}\sum_{y'} g(x)_{y'} - \sum_{y \neq y^{\star}} g(x)_y - g(x)_{y^{\star}} \geq (K-1)\gamma - g(x)_{y^{\star}}$$

$$\Rightarrow g(x)_{y^{\star}} \geq (K-1)\gamma + \frac{1}{K}\sum_y g(x)_y$$

$$\Rightarrow g(x)_{y^{\star}} \geq K\gamma + g(x)_y \ \forall y \neq y^{\star}.$$

The last line follows from re-using the original inequality and proves the desired implication.

On the other hand, if $\ell_{\text{MC-hinge}}^{K\gamma}(g,(x,y^{\star})) = 0$ it does not imply that $\ell_{\text{CC-hinge}}^{\gamma}(g,(x,y^{\star})) = 0$. To see why, suppose $K = 3$ and assume $y^{\star} = y_1$ with $y_2, y_3$ as the other labels. Set $g(x)_{y_1} = 3\gamma$, $g(x)_{y_2} = 0$ and $g(x)_{y_3} = -3\gamma$. With these predictions, we have $\sum_y g(x)_y = 0$ and also that $\ell_{\text{MC-hinge}}^{3\gamma}(g,(x,y_1)) = 0$. On the other hand since $g(x)_{y_2} = \sum_y g(x)_y = 0$, we get:

$$\ell_{\text{CC-hinge}}^{\tilde{\gamma}}(g,(x,y)) \geq \max\{1 - \tilde{\gamma}^{-1}(\sum_y g(x)_y - g(x)_{y_2}), 0\} = 1,$$

for any $\tilde{\gamma}$. This proves that the converse cannot be true. $\qquad\square$

## C  Proofs from Section 2

Let us start with an intermediate result, which will simplify the proof of Theorem 2.

**Theorem 12.** *Assume $\|\ell\|_1 \leq 1$ for all $\ell \in \mathcal{L}$[10] and $\sup_{s \in \mathcal{S}}\|s\|_{\infty} \leq 1$. Further assume that $\mathcal{S}$ and $\mathcal{L}$ are compact. Fix any constants $\eta \in (0,1]$, $\lambda > 0$, and $\beta > \alpha > 0$. Then there exists an algorithm with the following deterministic regret guarantee:*

$$\sum_{t=1}^{T}\mathbb{E}_{s_t \sim p_t}\langle s, \ell_t\rangle - \inf_{g \in \mathcal{G}}\sum_{t=1}^{T}\langle g(x_t), \ell_t\rangle \leq 2\eta\sum_{t=1}^{T}\mathbb{E}_{s_t \sim p_t}\langle s_t, \ell_t\rangle^2 + \frac{4}{\eta}\log\mathcal{N}_{\infty,\infty}(\beta/2,\mathcal{G},T) + 3e^2\alpha\sum_{t=1}^{T}\|\ell_t\|_1$$

$$+ 24e\left(\frac{\lambda}{4}\sum_{t=1}^{T}\|\ell_t\|_1^2 + \frac{1}{\lambda}\right)\int_{\alpha}^{\beta}\sqrt{\log\mathcal{N}_{\infty,\infty}(\varepsilon,\mathcal{G},T)}d\varepsilon.$$

The difference here is that have set $R, B = 1$. The first part of this section will be devoted to proving this theorem, and Theorem 2 will follow from this result via Corollary 16.

### C.1  Preliminaries

**Definition 2** (Cover for a collection of trees). *For a collection of $\mathbb{R}^K$-valued trees $U$ of length $T$, we let $\mathcal{N}_{\infty,\infty}(\varepsilon,U)$, denote the cardinality of the smallest set $V$ of $\mathbb{R}^K$ valued trees for which*

$$\forall \boldsymbol{u} \in U \ \forall \epsilon \in \{\pm 1\}^T \ \exists \boldsymbol{v} \in V \text{ s.t. } \max_{t \in [T]}\|\boldsymbol{u}_t(\epsilon) - \boldsymbol{v}_t(\epsilon)\|_{\infty} \leq \varepsilon.$$

**Definition 3** ($L_\infty/\ell_\infty$ radius). *For a function class $\mathcal{F}$, define*
$$\mathrm{rad}_{\infty,\infty}(\mathcal{F},T) = \min\{\varepsilon \mid \log \mathcal{N}_{\infty,\infty}(\varepsilon,\mathcal{F},T) = 0\}.$$
*For a collection $U$ of trees, define $\mathrm{rad}_{\infty,\infty}(U) = \min\{\varepsilon \mid \log \mathcal{N}_{\infty,\infty}(\varepsilon,U) = 0\}$.*

The following two lemmas are Freedman-type inequalities for Rademacher tree processes that we will use in the sequel. The first has an explicit dependence on the range, while the second does not.

**Lemma 13.** *For any collection of $[-R,+R]$-valued trees $V$ of length $T$, for any $\eta > 0$ and $\alpha > 0$,*
$$\mathbb{E}_\epsilon \sup_{\boldsymbol{v}\in V}\left[\sum_{t=1}^{T} \epsilon_t\big(\boldsymbol{v}_t(\epsilon) - \eta\boldsymbol{v}_t^2(\epsilon)\big) - \alpha\eta\boldsymbol{v}_t^2(\epsilon)\right] \le 2\log|V|\cdot\left(\frac{1}{\alpha\eta} \vee \frac{\eta R^2}{\alpha}\right).$$

**Proof of Lemma 13.** Take $V$ to be finite without loss of generality (otherwise the bound is vacuous). As a starting point, for any $\lambda > 0$ we have
$$\mathbb{E}_\epsilon \sup_{\boldsymbol{v}\in V}\left[\sum_{t=1}^{T} \epsilon_t\big(\boldsymbol{v}_t(\epsilon) - \eta\boldsymbol{v}_t^2(\epsilon)\big) - \alpha\eta\boldsymbol{v}_t^2(\epsilon)\right]$$
$$\le \frac{1}{\lambda}\log\left(\sum_{\boldsymbol{v}\in V} \mathbb{E}_\epsilon \exp\left(\sum_{t=1}^{T} \epsilon_t\lambda\big(\boldsymbol{v}_t(\epsilon) - \eta\boldsymbol{v}_t^2(\epsilon)\big) - \lambda\alpha\eta\boldsymbol{v}_t^2(\epsilon)\right)\right).$$

Applying the standard Rademacher mgf bound $\mathbb{E}_\epsilon\, e^{\lambda\epsilon} \le e^{\frac{1}{2}\lambda^2}$ conditionally at each time starting from $t = T$, this is upper bounded by
$$\le \frac{1}{\lambda}\log\left(\sum_{\boldsymbol{v}\in V} \max_\epsilon \exp\left(\sum_{t=1}^{T} \frac{1}{2}\lambda^2\big(\boldsymbol{v}_t(\epsilon) - \eta\boldsymbol{v}_t^2(\epsilon)\big)^2 - \lambda\alpha\eta\boldsymbol{v}_t^2(\epsilon)\right)\right).$$

Since $v$ takes values in $[-R,+R]$, the exponent at time $t$ can be upper bounded as
$$\frac{1}{2}\lambda^2\big(\boldsymbol{v}_t(\epsilon) - \eta\boldsymbol{v}_t^2(\epsilon)\big)^2 - \lambda\alpha\eta\boldsymbol{v}_t^2(\epsilon) \le \lambda^2\big(1 + \eta^2 R^2\big)\boldsymbol{v}_t^2(\epsilon) - \lambda\alpha\eta\boldsymbol{v}_t^2(\epsilon).$$

By setting $\lambda = \frac{1}{2}\min\{\alpha\eta, \alpha/(\eta R^2)\}$, this is bounded by zero, which leads to a final bound of $\log|V|/\lambda$. $\qquad\square$

**Lemma 14.** *For any collection of trees $V$ of length $T$, for any $\eta > 0$,*
$$\mathbb{E}_\epsilon \sup_{\boldsymbol{v}\in V}\left[\sum_{t=1}^{T} \epsilon_t\boldsymbol{v}_t(\epsilon) - \eta\boldsymbol{v}_t^2(\epsilon)\right] \le \frac{\log|V|}{2\eta}.$$

**Proof of Lemma 14.** Take $V$ to be finite without loss of generality. As in the proof of Lemma 13, using the standard Rademacher mgf bound and working backward from $T$, for any $\lambda > 0$ we have
$$\mathbb{E}_\epsilon \sup_{\boldsymbol{v}\in V}\left[\sum_{t=1}^{n} \epsilon_t\boldsymbol{v}_t(\epsilon) - \eta\boldsymbol{v}_t^2(\epsilon)\right] \le \frac{1}{\lambda}\log\left(\sum_{\boldsymbol{v}\in V} \mathbb{E}_\epsilon \exp\left(\sum_{t=1}^{n} \epsilon_t\lambda\boldsymbol{v}_t(\epsilon) - \eta\lambda\boldsymbol{v}_t^2(\epsilon)\right)\right)$$
$$\le \frac{1}{\lambda}\log\left(\sum_{\boldsymbol{v}\in V} \max_\epsilon \exp\left(\sum_{t=1}^{n} \frac{1}{2}\lambda^2\boldsymbol{v}_t(\epsilon)^2 - \eta\lambda\boldsymbol{v}_t^2(\epsilon)\right)\right).$$

The exponent at time $t$ is
$$\frac{1}{2}\lambda^2\boldsymbol{v}_t^2(\epsilon) - \eta\lambda\boldsymbol{v}_t^2(\epsilon).$$

By setting $\lambda = 2\eta$, this is exactly zero, which leads to a final bound of $\log|V|/\lambda$. $\qquad\square$

**Lemma 15.** *Let $\mathcal{Z}$, $\mathcal{W}$, and $\mathcal{G}$ be abstract sets and let functions $A_g : \mathcal{W} \times \mathcal{Z} \times \mathcal{Z} \to \mathbb{R}$ and $B_g : \mathcal{W} \times \mathcal{Z} \times \mathcal{Z} \to \mathbb{R}$ be given for each element $g \in \mathcal{G}$. Suppose that for any $z, z' \in \mathcal{Z}$ and $w \in \mathcal{W}$ it holds that $A(w,z,z') = -A(w,z',z)$ and $B(w,z,z') = B(w,z',z)$. Then*
$$\left\langle\!\!\left\langle \sup_{w_t\in\mathcal{W}} \sup_{q_t\in\Delta(\mathcal{Z})} \mathbb{E}_{z_t,z_t'\sim q_t} \right\rangle\!\!\right\rangle_{t=1}^{T} \sup_{g\in\mathcal{G}} \sum_{t=1}^{T} A_g(w_t,z_t,z_t') + B_g(w_t,z_t,z_t') \tag{5}$$
$$\le \left\langle\!\!\left\langle \sup_{w_t\in\mathcal{W}} \sup_{q_t\in\Delta(\mathcal{Z})} \mathbb{E}_{\epsilon_t} \mathbb{E}_{z_t,z_t'\sim q_t} \right\rangle\!\!\right\rangle_{t=1}^{T} \sup_{g\in\mathcal{G}} \sum_{t=1}^{T} \epsilon_t A_g(w_t,z_t,z_t') + B_g(w_t,z_t,z_t'), \tag{6}$$
*where $\epsilon$ is a sequence of independent Rademacher random variables.*

See Subsection C.2 for a discussion of the $\langle\!\langle \star \rangle\!\rangle$ notation used in the above lemma statement.

**Proof of Lemma 15.** See proof of Lemma 3 in [40]. $\square$

## C.2 Proof of Theorem 12

Before proceeding, we note that this proof uses a number of techniques which are now somewhat standard in minimax analysis of online learning, and the reader may wish to refer to, e.g., [41] for a comprehensive introduction to this type of analysis.

Let $\eta_1, \eta_2, \eta_3 > 0$ be fixed constants to be chosen later in the proof, and define

$$B(p_{1:T}, \ell_{1:T}) \triangleq \underbrace{\eta_1 \sum_{t=1}^{T} \|\ell_t\|_1 + \eta_2 \sum_{t=1}^{T} \|\ell_t\|_1^2}_{\triangleq B_1(\ell_{1:T})} + \underbrace{2\eta_3 \sum_{t=1}^{T} \mathbb{E}_{s \sim p_t} \langle s, \ell_t \rangle^2}_{\triangleq B_2(p_{1:T}, \ell_{1:T})}.$$

We consider a game where the goal of the learner is to achieve regret bounded by $B$, plus some additive constant that will depend on $\eta_1, \eta_2, \eta_3$, and the complexity of the class $\mathcal{F}$. The value of the game is given by:

$$\mathcal{V} \triangleq \left\langle\!\!\left\langle \sup_{x_t \in \mathcal{X}} \inf_{p_t \in \Delta(\mathcal{S})} \sup_{\ell_t \in \mathcal{L}} \mathbb{E}_{s \sim p_t} \right\rangle\!\!\right\rangle_{t=1}^{T} \left[ \sum_{t=1}^{T} \langle s, \ell_t \rangle - \inf_{g \in \mathcal{G}} \sum_{t=1}^{n} \langle g(x_t), \ell_t \rangle - B(p_{1:T}, \ell_{1:T}) \right].$$

Here we are using the notation $\langle\!\langle \star \rangle\!\rangle_{t=1}^{T}$ to denote sequential application of the operator $\star$ (indexed by $t$) from time $t = 1, \ldots, T$, following e.g. [41]. This notation means that first the adversary chooses $x_1$, then the learner chooses $p_1$, and then the adversary chooses $\ell_1$ while the learner samples $s_1$ and suffers the loss $\langle s_1, \ell_1 \rangle$. Then we proceed to round 2 and so on, so that the learner is trying to minimize the (offset) regret after $T$ rounds while the adversary is trying to maximize it. If we show that $\mathcal{V} \leq C$ for some constant $C$ then we have established existence of a randomized strategy that achieves an adaptive regret bound of $B(\cdot) + C$. See Foster et al. [20] for a more extensive discussion of this principle.

### C.2.1 Minimax swap

At time $t$ the value to go is given by

$$\sup_{x_t \in \mathcal{X}} \inf_{p_t \in \Delta(\mathcal{S})} \sup_{\ell_t \in \mathcal{L}} \Bigg[ \mathbb{E}_{s \sim p_t} \langle s, \ell_t \rangle - 2\eta_3 \mathbb{E}_{s \sim p_t} \langle s, \ell_t \rangle^2 - \eta_1 \|\ell_t\|_1 - \eta_2 \|\ell_t\|_1^2$$
$$+ \left\langle\!\!\left\langle \sup_{x_\tau \in \mathcal{X}} \inf_{p_\tau \in \Delta(\mathcal{S})} \sup_{\ell_\tau \in \mathcal{L}} \right\rangle\!\!\right\rangle_{\tau=t+1}^{T} \left[ \sum_{\tau=t+1}^{T} \mathbb{E}_{s \sim p_\tau} \langle s, \ell_\tau \rangle - \inf_{g \in \mathcal{G}} \sum_{\tau=1}^{T} \langle g(x_\tau), \ell_\tau \rangle - B(p_{\tau+1:T}, \ell_{\tau+1:T}) \right] \Bigg].$$

Note that the benchmark's loss is only evaluated at the end, while we are incorporating the adaptive term into the instantaneous value. Convexifying the $\ell_t$ player by allowing them to select a randomized strategy $q_t$, this is equal to

$$\sup_{x_t \in \mathcal{X}} \inf_{p_t \in \Delta(\mathcal{S})} \sup_{q_t \in \Delta(\mathcal{L})} \mathbb{E}_{\ell_t \sim q_t} \Bigg[ \mathbb{E}_{s \sim p_t} \langle s, \ell_t \rangle - 2\eta_3 \mathbb{E}_{s \sim p_t} \langle s, \ell_t \rangle^2 - \eta_1 \|\ell_t\|_1 - \eta_2 \|\ell_t\|_1^2$$
$$+ \left\langle\!\!\left\langle \sup_{x_\tau \in \mathcal{X}} \inf_{p_\tau \in \Delta(\mathcal{S})} \sup_{\ell_\tau \in \mathcal{L}} \right\rangle\!\!\right\rangle_{\tau=t+1}^{T} \left[ \sum_{\tau=t+1}^{T} \mathbb{E}_{s \sim p_\tau} \langle s, \ell_\tau \rangle - \inf_{g \in \mathcal{G}} \sum_{\tau=1}^{T} \langle g(x_\tau), \ell_\tau \rangle - B(p_{\tau+1:T}, \ell_{\tau+1:T}) \right] \Bigg].$$

This quantity is convex in $p_t$ and linear in $q_t$ so, under the compactness assumption on $\mathcal{S}$ and $\mathcal{L}$, the minimax theorem implies that this is equal to

$$\sup_{x_t \in \mathcal{X}} \sup_{q_t \in \Delta(\mathcal{L})} \inf_{p_t \in \Delta(\mathcal{S})} \mathbb{E}_{\ell_t \sim q_t} \Bigg[ \mathbb{E}_{s \sim p_t} \langle s, \ell_t \rangle - 2\eta_3 \mathbb{E}_{s \sim p_t} \langle s, \ell_t \rangle^2 - \eta_1 \|\ell_t\|_1 - \eta_2 \|\ell_t\|_1^2$$
$$+ \left\langle\!\!\left\langle \sup_{x_\tau \in \mathcal{X}} \inf_{p_\tau \in \Delta(\mathcal{S})} \sup_{\ell_\tau \in \mathcal{L}} \right\rangle\!\!\right\rangle_{\tau=t+1}^{T} \left[ \sum_{\tau=t+1}^{T} \mathbb{E}_{s \sim p_\tau} \langle s, \ell_\tau \rangle - \inf_{g \in \mathcal{G}} \sum_{\tau=1}^{T} \langle g(x_\tau), \ell_\tau \rangle - B(p_{\tau+1:T}, \ell_{\tau+1:T}) \right] \Bigg].$$

Repeating this analysis at each timestep and expanding the terms from $B_2$, we arrive at the expression

$$\mathcal{V} = \left\langle\!\!\left\langle \sup_{x_t \in \mathcal{X}} \sup_{q_t \in \Delta(\mathcal{L})} \inf_{p_t \in \Delta(\mathcal{S})} \mathbb{E}_{\ell_t \sim q_t} \right\rangle\!\!\right\rangle_{t=1}^{T} \left[ \sum_{t=1}^{T} \mathbb{E}_{s \sim p_t} \left[ \langle s, \ell_t \rangle - 2\eta_3 \langle s, \ell_t \rangle^2 \right] - \inf_{g \in \mathcal{G}} \sum_{t=1}^{T} \langle g(x_t), \ell_t \rangle - B_1(\ell_{1:T}) \right].$$

### C.2.2  Upper bound by martingale process

We now use a standard "rearrangement" trick (see [41], Theorem 1) to show that

$$\mathcal{V} = \left\langle\!\!\left\langle \sup_{x_t \in \mathcal{X}} \sup_{q_t \in \Delta(\mathcal{L})} \mathbb{E}_{\ell_t \sim q_t} \right\rangle\!\!\right\rangle_{t=1}^{T} \left[ \sup_{g \in \mathcal{G}}\left[ \sum_{t=1}^{T} \inf_{p_t \in \Delta(\mathcal{S})} \mathbb{E}_{s \sim p_t} \mathbb{E}_{\ell'_t \sim q_t}\left[ \langle s, \ell'_t \rangle - 2\eta_3 \langle s, \ell'_t \rangle^2 \right] - \sum_{t=1}^{T} \langle f(x_t), \ell_t \rangle \right] - B_1(\ell_{1:T}) \right],$$

where $\ell'_{1:T}$ is a sequence of "tangent" samples, where $\ell'_t$ is an independent copy of $\ell_t$ conditioned on $\ell_{1:t-1}$. This can be seen by working backwards from time $T$. Indeed, at time $T$, expanding the $\langle\!\langle \star \rangle\!\rangle_{t=1}^{T}$ operator, we have

$$\mathcal{V} = \langle\!\langle \cdots \rangle\!\rangle_{t=1}^{T-1} \sup_{x_T \in \mathcal{X}} \sup_{q_T \in \Delta(\mathcal{L})} \inf_{p_T \in \Delta(\mathcal{S})} \mathbb{E}_{\ell_T \sim q_T}\left[ \sum_{t=1}^{T-1} \mathbb{E}_{s \sim p_t}\left[ \langle s, \ell_t \rangle - 2\eta_3 \langle s, \ell_t \rangle^2 \right] + \mathbb{E}_{s \sim p_T}\left[ \langle s, \ell_T \rangle - 2\eta_3 \langle s, \ell_T \rangle^2 \right] \right.$$

$$\left. - \inf_{g \in \mathcal{G}} \sum_{t=1}^{T} \langle g(x_t), \ell_t \rangle - B_1(\ell_{1:T}) \right].$$

Using linearity of expectation:

$$= \langle\!\langle \cdots \rangle\!\rangle_{t=1}^{T-1} \sup_{x_T \in \mathcal{X}} \sup_{q_T \in \Delta(\mathcal{L})} \inf_{p_T \in \Delta(\mathcal{S})} \mathbb{E}_{\ell_T \sim q_T}\left[ \sum_{t=1}^{T-1} \mathbb{E}_{s \sim p_t}\left[ \langle s, \ell_t \rangle - 2\eta_3 \langle s, \ell_t \rangle^2 \right] + \mathbb{E}_{\ell'_T \sim q_T} \mathbb{E}_{s \sim p_T}\left[ \langle s, \ell'_T \rangle - 2\eta_3 \langle s, \ell'_T \rangle^2 \right] \right.$$

$$\left. - \inf_{g \in \mathcal{G}} \sum_{t=1}^{T} \langle g(x_t), \ell_t \rangle - B_1(\ell_{1:T}) \right].$$

Using that only a single term has functional dependence on $p_T$:

$$= \langle\!\langle \cdots \rangle\!\rangle_{t=1}^{T-1} \sup_{x_T \in \mathcal{X}} \sup_{q_T \in \Delta(\mathcal{L})} \mathbb{E}_{\ell_T \sim q_T}\left[ \sum_{t=1}^{T-1} \mathbb{E}_{s \sim p_t}\left[ \langle s, \ell_t \rangle - 2\eta_3 \langle s, \ell_t \rangle^2 \right] + \inf_{p_T \in \Delta(\mathcal{S})} \mathbb{E}_{\ell'_T \sim q_T} \mathbb{E}_{s \sim p_T}\left[ \langle s, \ell'_T \rangle - 2\eta_3 \langle s, \ell'_T \rangle^2 \right] \right.$$

$$\left. - \inf_{g \in \mathcal{G}} \sum_{t=1}^{T} \langle g(x_t), \ell_t \rangle - B_1(\ell_{1:T}) \right].$$

Expanding the infimum over $g \in \mathcal{G}$:

$$= \langle\!\langle \cdots \rangle\!\rangle_{t=1}^{T-1} \sup_{x_T \in \mathcal{X}} \sup_{q_T \in \Delta(\mathcal{L})} \mathbb{E}_{\ell_T \sim q_T} \sup_{g \in \mathcal{G}}\left[ \sum_{t=1}^{T-1} \mathbb{E}_{s \sim p_t}\left[ \langle s, \ell_t \rangle - 2\eta_3 \langle s, \ell_t \rangle^2 \right] + \inf_{p_T \in \Delta(\mathcal{S})} \mathbb{E}_{\ell'_T \sim q_T} \mathbb{E}_{s \sim p_T}\left[ \langle s, \ell'_T \rangle - 2\eta_3 \langle s, \ell'_T \rangle^2 \right] \right.$$

$$\left. - \sum_{t=1}^{T} \langle g(x_t), \ell_t \rangle - B_1(\ell_{1:T}) \right].$$

We handle time $T - 1$ in a similar fashion by first splitting the $\langle\!\langle \star \rangle\!\rangle_{t=1}^{T-1}$ operator:

$$= \langle\!\langle \cdots \rangle\!\rangle_{t=1}^{T-2} \sup_{x_{T-1} \in \mathcal{X}} \sup_{q_{T-1} \in \Delta(\mathcal{L})} \inf_{p_{T-1} \in \Delta(\mathcal{S})} \mathbb{E}_{\ell_{T-1} \sim q_{T-1}} \sup_{x_T \in \mathcal{X}} \sup_{q_T \in \Delta(\mathcal{L})} \mathbb{E}_{\ell_T \sim q_T}$$

$$\sup_{g \in \mathcal{G}}\left[ \sum_{t=1}^{T-2} \mathbb{E}_{s \sim p_t}\left[ \langle s, \ell_t \rangle - 2\eta_3 \langle s, \ell_t \rangle^2 \right] + \mathbb{E}_{s \sim p_{T-1}}\left[ \langle s, \ell_{T-1} \rangle - 2\eta_3 \langle s, \ell_{T-1} \rangle^2 \right] \right.$$

$$\left. + \inf_{p_T \in \Delta(\mathcal{S})} \mathbb{E}_{\ell'_T \sim q_T} \mathbb{E}_{s \sim p_T}\left[ \langle s, \ell'_T \rangle - 2\eta_3 \langle s, \ell'_T \rangle^2 \right] - \sum_{t=1}^{T} \langle g(x_t), \ell_t \rangle - B_1(\ell_{1:T}) \right].$$

Rearranging the supremums to make dependence on terms from time $T - 1$ clear:

$$= \langle\!\langle \cdots \rangle\!\rangle_{t=1}^{T-2} \sup_{x_{T-1} \in \mathcal{X}} \sup_{q_{T-1} \in \Delta(\mathcal{L})} \inf_{p_{T-1} \in \Delta(\mathcal{S})} \mathbb{E}_{\ell_{T-1} \sim q_{T-1}}$$

$$\left[ \sum_{t=1}^{T-2} \mathbb{E}_{s \sim p_t}\left[ \langle s, \ell_t \rangle - 2\eta_3 \langle s, \ell_t \rangle^2 \right] + \mathbb{E}_{s \sim p_{T-1}}\left[ \langle s, \ell_{T-1} \rangle - 2\eta_3 \langle s, \ell_{T-1} \rangle^2 \right] \right.$$

$$\left. + \sup_{x_T \in \mathcal{X}} \sup_{q_T \in \Delta(\mathcal{L})} \mathbb{E}_{\ell_T \sim q_T} \sup_{g \in \mathcal{G}}\left[ \inf_{p_T \in \Delta(\mathcal{S})} \mathbb{E}_{\ell'_T \sim q_T} \mathbb{E}_{s \sim p_T}\left[ \langle s, \ell'_T \rangle - 2\eta_3 \langle s, \ell'_T \rangle^2 \right] - \sum_{t=1}^{T} \langle g(x_t), \ell_t \rangle - B_1(\ell_{1:T}) \right] \right].$$

Using linearity of expectation and moving the infimum over $q_{T-1}$:

$$= \langle\!\langle \cdots \rangle\!\rangle_{t=1}^{T-2} \sup_{x_{T-1} \in \mathcal{X}} \sup_{q_{T-1} \in \Delta(\mathcal{L})} \mathbb{E}_{\ell_{T-1} \sim q_{T-1}}$$

$$\left[ \sum_{t=1}^{T-2} \mathbb{E}_{s \sim p_t}\left[ \langle s, \ell_t \rangle - 2\eta_3 \langle s, \ell_t \rangle^2 \right] + \inf_{p_{T-1} \in \Delta(\mathcal{S})} \mathbb{E}_{\ell'_{T-1} \sim q_{T-1}} \mathbb{E}_{s \sim p_{T-1}}\left[ \langle s, \ell'_{T-1} \rangle - 2\eta_3 \langle s, \ell'_{T-1} \rangle^2 \right] \right.$$

$$\left. + \sup_{x_T \in \mathcal{X}} \sup_{q_T \in \Delta(\mathcal{L})} \mathbb{E}_{\ell_T \sim q_T} \sup_{g \in \mathcal{G}}\left[ \inf_{p_T \in \Delta(\mathcal{S})} \mathbb{E}_{\ell'_T \sim q_T} \mathbb{E}_{s \sim p_T}\left[ \langle s, \ell'_T \rangle - 2\eta_3 \langle s, \ell'_T \rangle^2 \right] - \sum_{t=1}^{T} \langle g(x_t), \ell_t \rangle - B_1(\ell_{1:T}) \right] \right].$$

The last step is to move the supremums from time $t = T$ and the supremum over $g \in \mathcal{G}$ outside the entire expression, similar to what was done at time $t = T$.

$$= \langle\!\langle \cdots \rangle\!\rangle_{t=1}^{T-2} \sup_{x_{T-1} \in \mathcal{X}} \sup_{q_{T-1} \in \Delta(\mathcal{L})} \mathbb{E}_{\ell_{T-1} \sim q_{T-1}} \sup_{x_T \in \mathcal{X}} \sup_{q_T \in \Delta(\mathcal{L})} \mathbb{E}_{\ell_T \sim q_T} \sup_{g \in \mathcal{G}}$$

$$\left[ \sum_{t=1}^{T-2} \mathbb{E}_{s \sim p_t} \big[ \langle s, \ell_t \rangle - 2\eta_3 \langle s, \ell_t \rangle^2 \big] + \inf_{p_{T-1} \in \Delta(\mathcal{S})} \mathbb{E}_{\ell'_{T-1} \sim q_{T-1}} \mathbb{E}_{s \sim p_{T-1}} \big[ \langle s, \ell'_{T-1} \rangle - 2\eta_3 \langle s, \ell'_{T-1} \rangle^2 \big] \right.$$

$$\left. + \inf_{p_T \in \Delta(\mathcal{S})} \mathbb{E}_{\ell'_T \sim q_T} \mathbb{E}_{s \sim p_T} \big[ \langle s, \ell'_T \rangle - 2\eta_3 \langle s, \ell'_T \rangle^2 \big] - \sum_{t=1}^T \langle g(x_t), \ell_t \rangle - B_1(\ell_{1:T}) \right].$$

Repeating this argument down from time $t = T - 2$ to time $t = 1$ yields the result.

To conclude this portion of the proof, we move to an upper bound by choosing the infimum over $p_t$ at each timestep $t$ to match $g$, which is possible because each infimum now occurs inside the expression for which the supremum over $g \in \mathcal{G}$ is taken:

$$\mathcal{V} = \left\langle\!\!\left\langle \sup_{x_t \in \mathcal{X}} \sup_{q_t \in \Delta(\mathcal{L})} \mathbb{E}_{\ell_t \sim q_t} \right\rangle\!\!\right\rangle_{t=1}^T \left[ \sup_{g \in \mathcal{G}} \left[ \sum_{t=1}^T \inf_{p_t \in \Delta(\mathcal{S})} \mathbb{E}_{s \sim p_t} \mathbb{E}_{\ell'_t \sim q_t} \big[ \langle s, \ell'_t \rangle - 2\eta_3 \langle s, \ell'_t \rangle^2 \big] - \sum_{t=1}^T \langle f(x_t), \ell_t \rangle \right] - B_1(\ell_{1:T}) \right]$$

$$\leq \left\langle\!\!\left\langle \sup_{x_t \in \mathcal{X}} \sup_{q_t \in \Delta(\mathcal{L})} \mathbb{E}_{\ell_t \sim q_t} \right\rangle\!\!\right\rangle_{t=1}^T \left[ \sup_{g \in \mathcal{G}} \left[ \sum_{t=1}^T \mathbb{E}_{\ell'_t \sim q_t} \big[ \langle g(x_t), \ell'_t \rangle - 2\eta_3 \langle g(x_t), \ell'_t \rangle^2 \big] - \sum_{t=1}^T \langle g(x_t), \ell_t \rangle \right] - B_1(\ell_{1:T}) \right]$$

$$= \left\langle\!\!\left\langle \sup_{x_t \in \mathcal{X}} \sup_{q_t \in \Delta(\mathcal{L})} \mathbb{E}_{\ell_t \sim q_t} \right\rangle\!\!\right\rangle_{t=1}^T \left[ \sup_{g \in \mathcal{G}} \left[ \sum_{t=1}^T \mathbb{E}_{\ell'_t \sim q_t} \big[ \langle g(x_t), \ell'_t \rangle \big] - \langle g(x_t), \ell_t \rangle - 2\eta_3 \sum_{t=1}^T \mathbb{E}_{\ell'_t \sim q_t} \big[ \langle g(x_t), \ell'_t \rangle^2 \big] \right] - B_1(\ell_{1:T}) \right].$$

$$\tag{7}$$

### C.2.3 Symmetrization

Introduce the notation $H(x) = x - \eta_3 x^2$. We now claim that the quantity appearing in (7) is bounded by

$$2 \cdot \sup_{\boldsymbol{x}} \sup_{\boldsymbol{\ell}} \mathbb{E}_\epsilon \left[ \sup_{g \in \mathcal{G}} \left[ \sum_{t=1}^T \epsilon_t H(\langle g(\boldsymbol{x}_t(\epsilon)), \boldsymbol{\ell}_t(\epsilon) \rangle) - \eta_3 \sum_{t=1}^T \langle g(\boldsymbol{x}_t(\epsilon)), \boldsymbol{\ell}_t(\epsilon) \rangle^2 \right] - B_1(\boldsymbol{\ell}_{1:T}(\epsilon)) \right], \quad (8)$$

where the supremum ranges over all $\mathcal{X}$-valued trees $\boldsymbol{x}$ and $\mathcal{L}$-valued trees $\boldsymbol{\ell}$, both of length $T$.

The value

$$\left\langle\!\!\left\langle \sup_{x_t \in \mathcal{X}} \sup_{q_t \in \Delta(\mathcal{L})} \mathbb{E}_{\ell_t \sim q_t} \right\rangle\!\!\right\rangle_{t=1}^T \left[ \sup_{g \in \mathcal{G}} \left[ \sum_{t=1}^T \mathbb{E}_{\ell'_t \sim q_t} \big[ \langle g(x_t), \ell'_t \rangle \big] - \langle g(x_t), \ell_t \rangle - 2\eta_3 \sum_{t=1}^T \mathbb{E}_{\ell'_t \sim q_t} \big[ \langle g(x_t), \ell'_t \rangle^2 \big] \right] - B_1(\ell_{1:T}) \right],$$

by adding and subtracting the same term, is equal to

$$\left\langle\!\!\left\langle \sup_{x_t \in \mathcal{X}} \sup_{q_t \in \Delta(\mathcal{L})} \mathbb{E}_{\ell_t \sim q_t} \right\rangle\!\!\right\rangle_{t=1}^T \left[ \sup_{g \in \mathcal{G}} \left[ \sum_{t=1}^T \mathbb{E}_{\ell'_t \sim q_t} \big[ \langle g(x_t), \ell'_t \rangle - \eta_3 \langle g(x_t), \ell'_t \rangle^2 \big] - \big( \langle g(x_t), \ell_t \rangle - \eta_3 \langle g(x_t), \ell_t \rangle^2 \big) \right. \right.$$

$$\left. \left. - \eta_3 \sum_{t=1}^T \big( \mathbb{E}_{\ell_t \sim q_t} \big[ \langle g(x_t), \ell_t \rangle^2 \big] + \langle g(x_t), \ell_t \rangle^2 \big) \right] - B_1(\ell_{1:T}) \right]$$

$$= \left\langle\!\!\left\langle \sup_{x_t \in \mathcal{X}} \sup_{q_t \in \Delta(\mathcal{L})} \mathbb{E}_{\ell_t \sim q_t} \right\rangle\!\!\right\rangle_{t=1}^T \left[ \sup_{g \in \mathcal{G}} \left[ \sum_{t=1}^T \mathbb{E}_{\ell'_t \sim q_t} \big[ H(\langle g(x_t), \ell'_t \rangle) \big] - H(\langle g(x_t), \ell_t \rangle) \right. \right.$$

$$\left. \left. - \eta_3 \sum_{t=1}^T \big( \mathbb{E}_{\ell_t \sim q_t} \big[ \langle g(x_t), \ell_t \rangle^2 \big] + \langle g(x_t), \ell_t \rangle^2 \big) \right] - B_1(\ell_{1:T}) \right].$$

Using Jensen's inequality, this is upper bounded by

$$\left\langle\!\!\left\langle \sup_{x_t \in \mathcal{X}} \sup_{q_t \in \Delta(\mathcal{L})} \mathbb{E}_{\ell_t, \ell'_t \sim q_t} \right\rangle\!\!\right\rangle_{t=1}^T \left[ \sup_{g \in \mathcal{G}} \left[ \sum_{t=1}^T H(\langle g(x_t), \ell'_t \rangle) - H(\langle g(x_t), \ell_t \rangle) \right. \right.$$

$$\left. \left. - \eta_3 \sum_{t=1}^T \big( \langle g(x_t), \ell'_t \rangle^2 + \langle g(x_t), \ell_t \rangle^2 \big) \right] - B_1(\ell_{1:n}) \right], \quad (9)$$

where $\ell'_{1:T}$ is a tangent sequence. We now claim that this is equal to

$$\left\langle\!\!\left\langle \sup_{x_t \in \mathcal{X}} \sup_{q_t \in \Delta(\mathcal{L})} \mathbb{E}_{\ell_t, \ell'_t \sim q_t} \right\rangle\!\!\right\rangle_{t=1}^{T} \left[ \sup_{g \in \mathcal{G}} \left[ \sum_{t=1}^{T} H(\langle g(x_t), \ell'_t \rangle) - H(\langle g(x_t), \ell_t \rangle) \right.\right.$$

$$\left.\left. - \eta_3 \sum_{t=1}^{T} \left( \langle g(x_t), \ell'_t \rangle^2 + \langle g(x_t), \ell_t \rangle^2 \right) \right] - \frac{1}{2} B_1(\ell_{1:T}) - \frac{1}{2} B_1(\ell'_{1:T}) \right].$$

This can be seen as follows: Let $Q$ be the joint distribution over $\ell_1, \ldots, \ell_T$ obtaining the supremum above, or if the supremum is not obtained let it be any point in a limit sequence approaching the supremum. Then the value of the $B_1$ term in (9) is equal to (respectively, $\varepsilon$-close to)

$$\mathbb{E}_Q B_1(\ell_{1:T}) = \eta_1 \sum_{t=1}^{T} \mathbb{E}_Q \|\ell_t\|_1 + \eta_2 \sum_{t=1}^{T} \mathbb{E}_Q \|\ell_t\|_1^2$$

$$= \eta_1 \sum_{t=1}^{T} \mathbb{E}_{\ell_{1:t-1}} \mathbb{E}\big[ \|\ell_t\|_1 \mid \ell_{1:t-1} \big] + \eta_2 \sum_{t=1}^{T} \mathbb{E}_{\ell_{1:t-1}} \mathbb{E}\Big[ \|\ell_t\|_1^2 \mid \ell_{1:t-1} \Big]$$

$$= \eta_1 \sum_{t=1}^{T} \mathbb{E}_{\ell_{1:t-1}} \mathbb{E}\big[ \|\ell'_t\|_1 \mid \ell_{1:t-1} \big] + \eta_2 \sum_{t=1}^{T} \mathbb{E}_{\ell_{1:t-1}} \mathbb{E}\Big[ \|\ell'_t\|_1^2 \mid \ell_{1:t-1} \Big]$$

$$= \mathbb{E}_{\ell_{1:T}} \mathbb{E}_{\ell'_{1:T}|\ell_{1:T}} B_1(\ell'_{1:T}).$$

Replacing $\ell_t$ with $\ell'_t$ follows from the definition of the tangent sequence, since $\ell'_t$ and $\ell_t$ are identically distributed, conditioned on $\ell_{1:t-1}$. This shows that we can replace $B_1(\ell_{1:T})$ with $B_1(\ell_{1:T})/2 + B_1(\ell'_{1:T})/2$ above, since we are working with the expectation.

We have now established that (9) is equal to

$$\left\langle\!\!\left\langle \sup_{x_t \in \mathcal{X}} \sup_{q_t \in \Delta(\mathcal{L})} \mathbb{E}_{\ell_t, \ell'_t \sim q_t} \right\rangle\!\!\right\rangle_{t=1}^{T} \left[ \sup_{g \in \mathcal{G}} \left[ \sum_{t=1}^{T} \underbrace{H(\langle g(x_t), \ell'_t \rangle)}_{A_1} - \underbrace{H(\langle g(x_t), \ell_t \rangle)}_{A_2} - \eta_3 \Big( \sum_{t=1}^{T} \underbrace{\langle g(x_t), \ell'_t \rangle^2 + \langle g(x_t), \ell_t \rangle^2}_{A_3} \Big) \right] \right.$$

$$\left. - \frac{\eta_1}{2} \Big( \sum_{t=1}^{T} \underbrace{\|\ell_t\|_1 + \|\ell'_t\|_1}_{A_4} \Big) - \frac{\eta_2}{2} \Big( \sum_{t=1}^{T} \underbrace{\|\ell_t\|_1^2 + \|\ell'_t\|_1^2}_{A_5} \Big) \right].$$

Fix a time $t$ and suppose the values of $\ell_t$ and $\ell'_t$ are exchanged. In this case the value of $A_1 - A_2$ is switched to $A_2 - A_1$, while the values of $A_3$, $A_4$, and $A_5$ are left unchanged. Appealing to Lemma 15, we can therefore introduce Rademacher random variables $\epsilon_1, \ldots, \epsilon_T$ with equality as follows:

$$\left\langle\!\!\left\langle \sup_{x_t \in \mathcal{X}} \sup_{q_t \in \Delta(\mathcal{L})} \mathbb{E}_{\ell_t, \ell'_t \sim q_t} \mathbb{E}_{\epsilon_t} \right\rangle\!\!\right\rangle_{t=1}^{T} \left[ \sup_{g \in \mathcal{G}} \left[ \sum_{t=1}^{T} \epsilon_t \big( H(\langle g(x_t), \ell'_t \rangle) - H(\langle g(x_t), \ell_t \rangle) \big) - \eta_3 \Big( \sum_{t=1}^{T} \langle g(x_t), \ell'_t \rangle^2 + \langle g(x_t), \ell_t \rangle^2 \Big) \right] \right.$$

$$\left. - \frac{\eta_1}{2} \Big( \sum_{t=1}^{T} \|\ell_t\|_1 + \|\ell'_t\|_1 \Big) - \frac{\eta_2}{2} \Big( \sum_{t=1}^{T} \|\ell_t\|_1^2 + \|\ell'_t\|_1^2 \Big) \right].$$

Splitting the supremum, this is upper bounded by

$$2 \cdot \left\langle\!\!\left\langle \sup_{x_t \in \mathcal{X}} \sup_{q_t \in \Delta(\mathcal{L})} \mathbb{E}_{\ell_t \sim q_t} \mathbb{E}_{\epsilon_t} \right\rangle\!\!\right\rangle_{t=1}^{T} \left[ \sup_{g \in \mathcal{G}} \left[ \sum_{t=1}^{T} \epsilon_t H(\langle g(x_t), \ell_t \rangle) - \eta_3 \sum_{t=1}^{T} \langle g(x_t), \ell_t \rangle^2 \right] - \frac{\eta_1}{2} \sum_{t=1}^{T} \|\ell_t\|_1 - \frac{\eta_2}{2} \sum_{t=1}^{T} \|\ell_t\|_1^2 \right]$$

$$= 2 \cdot \left\langle\!\!\left\langle \sup_{x_t \in \mathcal{X}} \sup_{\ell_t \in \mathcal{L}} \mathbb{E}_{\epsilon_t} \right\rangle\!\!\right\rangle_{t=1}^{T} \left[ \sup_{g \in \mathcal{G}} \left[ \sum_{t=1}^{T} \epsilon_t H(\langle g(x_t), \ell_t \rangle) - \eta_3 \sum_{t=1}^{T} \langle g(x_t), \ell_t \rangle^2 \right] - \frac{\eta_1}{2} \sum_{t=1}^{T} \|\ell_t\|_1 - \frac{\eta_2}{2} \sum_{t=1}^{T} \|\ell_t\|_1^2 \right]$$

$$= 2 \cdot \sup_{\boldsymbol{x}} \sup_{\boldsymbol{\ell}} \mathbb{E}_\epsilon \left[ \sup_{g \in \mathcal{G}} \left[ \sum_{t=1}^{T} \epsilon_t H(\langle g(\boldsymbol{x}_t(\epsilon)), \boldsymbol{\ell}_t(\epsilon) \rangle) - \eta_3 \sum_{t=1}^{T} \langle g(\boldsymbol{x}_t(\epsilon)), \boldsymbol{\ell}_t(\epsilon) \rangle^2 \right] - \frac{\eta_1}{2} \sum_{t=1}^{T} \|\boldsymbol{\ell}_t(\epsilon)\|_1 - \frac{\eta_2}{2} \sum_{t=1}^{T} \|\boldsymbol{\ell}_t(\epsilon)\|_1^2 \right].$$

The first equality is somewhat subtle, but holds because at time $T$, the expression is linear in $q_T$ so it is maximized at a point $\ell_T$, allowing us to work backwards to remove the $q_t$ distributions.

### C.2.4 Introducing a coarse cover

We now break the process appearing in (8) into multiple terms, each of which will be handled by covering. Consider any fixed pair of trees $\boldsymbol{x}, \boldsymbol{\ell}$. Note that with the trees fixed (7) is at most

$$2 \cdot \mathbb{E}_\epsilon \sup_{g \in \mathcal{G}} \left[ \sum_{t=1}^{T} \epsilon_t H(\langle g(\boldsymbol{x}_t(\epsilon)), \boldsymbol{\ell}_t(\epsilon) \rangle) - \eta_3 \sum_{t=1}^{T} \langle g(\boldsymbol{x}_t(\epsilon)), \boldsymbol{\ell}_t(\epsilon) \rangle^2 \right] - \mathbb{E}_\epsilon B_1(\boldsymbol{\ell}_{1:T}(\epsilon)).$$

We will focus on the supremum for now. We begin by adapting a trick from Rakhlin and Sridharan [37] to introduce a coarse sequential cover at scale $\beta$. Let $V'$ be a cover for $\mathcal{G}$ on the tree $\boldsymbol{x}$ with respect to $L_\infty/\ell_\infty$ at scale $\beta/2$. Then the size of $V'$ is $\mathcal{N}_{\infty,\infty}(\beta/2,\mathcal{G},\boldsymbol{x})$, and

$$\max_{g\in\mathcal{G}}\max_{\epsilon\in\{\pm 1\}^T}\min_{\boldsymbol{v}'\in V'}\max_{t\in[T]}\|g(\boldsymbol{x}_t(\epsilon))-\boldsymbol{v}'_t(\epsilon)\|_\infty \le \beta/2.$$

Recall that since $g(x)\in\mathbb{R}_+^K$ for all $g\in\mathcal{G}$, we may take each $\boldsymbol{v}'\in V'$ to have non-negative coordinates without loss of generality. Likewise, it follows that we may take each $\boldsymbol{v}'\in V'$ to have $\|\boldsymbol{v}'_t(\epsilon)\|_\infty \le \sup_{x\in\mathcal{X}}\sup_{g\in\mathcal{G}}\|g(x)\|_\infty$ without loss of generality.

We construct a new $\beta$-cover $V^1$ from $V'$ by defining for each tree $\boldsymbol{v}'\in V'$ a new tree $\boldsymbol{v}$ as follows:

$$\forall\epsilon\in\{\pm 1\}^T \ \forall t\in[T] \ \forall a\in[K]: \quad \boldsymbol{v}_t(\epsilon)_a = \max\{\boldsymbol{v}'_t(\epsilon)_a - \beta/2, 0\}.$$

It is easy to verify that for each time $t$ and path $\epsilon$ we have $\|\boldsymbol{v}_t(\epsilon)-\boldsymbol{v}'_t(\epsilon)\|_\infty \le \beta/2$, so $V^1$ is indeed a $\beta$-cover with respect to $L_\infty/\ell_\infty$. More importantly, for each $g\in\mathcal{G}$ and path $\epsilon$, there exists a tree $\boldsymbol{v}\in V'$ that is $\beta$-close in the $L_\infty/\ell_\infty$ sense and has $\boldsymbol{v}_t(\epsilon)_a \le g(\boldsymbol{x}_t(\epsilon))_a$ coordinate-wise. We will let $\boldsymbol{v}^1[\epsilon,g]$ denote this tree, and it is constructed by taking the $\beta/2$-close tree $\boldsymbol{v}'$ promised by the definition of $V'$, then performing the clipping operation above to get the corresponding $\beta$-close element of $V^1$. The clipping operation and $\beta/2$ closeness of $\boldsymbol{v}'$ imply that for each time $t\in[T]$ and coordinate $a\in[K]$,

$$\begin{aligned}
\boldsymbol{v}^1_t[\epsilon,g]_a - g(\boldsymbol{x}_t(\epsilon))_a &= \max\{\boldsymbol{v}'_t(\epsilon)_a - \beta/2, 0\} - g(\boldsymbol{x}_t(\epsilon))_a \\
&\le \max\{\|\boldsymbol{v}'_t(\epsilon) - g(\boldsymbol{x}_t(\epsilon))\|_\infty + g(\boldsymbol{x}_t(\epsilon))_a - \beta/2, 0\} - g(\boldsymbol{x}_t(\epsilon))_a \\
&\le \max\{g(\boldsymbol{x}_t(\epsilon))_a, 0\} - g(\boldsymbol{x}_t(\epsilon))_a = 0.
\end{aligned}$$

This establishes the desired ordering on coordinates. Returning to the process at hand, we have

$$\mathbb{E}_\epsilon\sup_{g\in\mathcal{G}}\left[\sum_{t=1}^T \epsilon_t H\langle g(\boldsymbol{x}_t(\epsilon)), \boldsymbol{\ell}_t(\epsilon)\rangle) - \eta_3\sum_{t=1}^T\langle g(\boldsymbol{x}_t(\epsilon)),\boldsymbol{\ell}_t(\epsilon)\rangle^2\right].$$

Now we add and subtract terms involving the covering element $\boldsymbol{v}^1(\epsilon,g)$:

$$\begin{aligned}
= \mathbb{E}_\epsilon\sup_{g\in\mathcal{G}}\Bigg[ &\sum_{t=1}^T \epsilon_t H(\langle \boldsymbol{v}^1_t[\epsilon,g], \boldsymbol{\ell}_t(\epsilon)\rangle) - \eta_3\sum_{t=1}^T\langle g(\boldsymbol{x}_t(\epsilon)),\boldsymbol{\ell}_t(\epsilon)\rangle^2 \\
&+ \sum_{t=1}^T \epsilon_t H(\langle g(\boldsymbol{x}_t(\epsilon)), \boldsymbol{\ell}_t(\epsilon)\rangle) - \epsilon_t H(\langle \boldsymbol{v}^1_t[\epsilon,g], \boldsymbol{\ell}_t(\epsilon)\rangle)\Bigg].
\end{aligned}$$

We now invoke the coordinate domination property of $\boldsymbol{v}^1[\epsilon,g]$ described above. Observe that since $g(\boldsymbol{x}_t(\epsilon))$, $\boldsymbol{v}^1_t[\epsilon,g]$, and $\boldsymbol{\ell}_t(\epsilon)$ are all nonnegative coordinate-wise, it holds that $\langle\boldsymbol{v}^1_t[\epsilon,g],\boldsymbol{\ell}_t(\epsilon)\rangle^2 \le \langle g(\boldsymbol{x}_t(\epsilon)),\boldsymbol{\ell}_t(\epsilon)\rangle^2$. Consequently, we can replace the offset term (not involving $\epsilon_t$) with a similar term involving $\boldsymbol{v}^1_t[\epsilon,g]$

$$\begin{aligned}
\le \mathbb{E}_\epsilon\sup_{g\in\mathcal{G}}\Bigg[ &\sum_{t=1}^T \epsilon_t H(\langle \boldsymbol{v}^1_t[\epsilon,f], \boldsymbol{\ell}_t(\epsilon)\rangle) - \eta_3\sum_{t=1}^T\langle \boldsymbol{v}^1_t[\epsilon,g],\boldsymbol{\ell}_t(\epsilon)\rangle^2 \\
&+ \sum_{t=1}^T \epsilon_t H(\langle f(\boldsymbol{x}_t(\epsilon)), \boldsymbol{\ell}_t(\epsilon)\rangle) - \epsilon_t H(\langle \boldsymbol{v}^1_t[\epsilon,f], \boldsymbol{\ell}_t(\epsilon)\rangle)\Bigg].
\end{aligned}$$

Splitting the supremum and gathering terms, this implies that $\mathcal{V}$ is upper bounded by

$$\underbrace{\mathbb{E}_\epsilon\sup_{\boldsymbol{v}^1\in V^1}\left[\sum_{t=1}^T \epsilon_t H(\langle \boldsymbol{v}^1_t(\epsilon), \boldsymbol{\ell}_t(\epsilon)\rangle) - \eta_3\sum_{t=1}^T\langle \boldsymbol{v}^1_t(\epsilon),\boldsymbol{\ell}_t(\epsilon)\rangle^2\right]}_{(\star)}$$

$$\underbrace{+ \mathbb{E}_\epsilon\sup_{g\in\mathcal{G}}\left[\sum_{t=1}^T \epsilon_t H(\langle g(\boldsymbol{x}_t(\epsilon)), \boldsymbol{\ell}_t(\epsilon)\rangle) - \epsilon_t H(\langle \boldsymbol{v}^1_t[\epsilon,g], \boldsymbol{\ell}_t(\epsilon)\rangle)\right] - \mathbb{E}_\epsilon B_1(\boldsymbol{\ell}_{1:T}(\epsilon))}_{(\star\star)}.$$

## C.2.5 Bounding $(\star)$

We appeal to Lemma 13 with a class of real-valued trees $U \triangleq \left\{ \epsilon \mapsto \left( \langle \boldsymbol{v}_t^1(\epsilon), \boldsymbol{\ell}_t(\epsilon) \rangle \right)_{t \leq T} \mid \boldsymbol{v}^1 \in V^1 \right\}$.
The class $U$ has range contained in $[-1, +1]$, since $\left| \langle \boldsymbol{v}_t^1(\epsilon), \boldsymbol{\ell}_t(\epsilon) \rangle \right| \leq \left\| \boldsymbol{v}_t^1(\epsilon) \right\|_\infty \left\| \boldsymbol{\ell}_t(\epsilon) \right\|_1 \leq 1$, where
these norm bounds are by assumption on $\mathcal{G}$ and $\mathcal{L}$. Recall that $H(x) = x - \eta_3 x^2$. We therefore
conclude that

$$
(\star) = \mathbb{E}_\epsilon \sup_{\boldsymbol{v}^1 \in V^1} \left[ \sum_{t=1}^T \epsilon_t H(\langle \boldsymbol{v}_t^1(\epsilon), \boldsymbol{\ell}_t(\epsilon) \rangle) - \eta_3 \sum_{t=1}^T \langle \boldsymbol{v}_t^1(\epsilon), \boldsymbol{\ell}_t(\epsilon) \rangle^2 \right]
$$

$$
\leq 2 \frac{1 + \eta_3^2}{\eta_3} \log |V^1| = 2 \frac{1 + \eta_3^2}{\eta_3} \log \mathcal{N}_{\infty, \infty}(\beta/2, \mathcal{G}, \boldsymbol{x}).
$$

## C.2.6 Bounding $(\star\star)$

Fix $\alpha > 0$ and let $N = \lfloor \log(\beta/\alpha) \rfloor - 1$. For each $i \geq 1$ define $\varepsilon_i = \beta e^{-(i-1)}$, and for each $i > 1$ let
$V^i$ be a sequential cover of $\mathcal{G}$ on $\boldsymbol{x}$ at scale $\varepsilon_i$ with respect to $L_\infty/\ell_\infty$ (keeping in mind that $V^1$ is
defined as in the preceding section). For a given path $\epsilon \in \{\pm 1\}^T$ and $g \in \mathcal{G}$, let $\boldsymbol{v}^i[\epsilon, g]$ denote the
$\varepsilon_i$-close element of $V^i$. Below, we will only evaluate $H(x) = x - \eta_3 x^2$ over the domain $[-1, +1]$; it
is $(1 + 2\eta_3)$-Lipschitz over this domain. Then the leading term of $(\star\star)$ is equal to

$$
\mathbb{E}_\epsilon \sup_{g \in \mathcal{G}} \left[ \sum_{t=1}^T \epsilon_t \big( H(\langle g(\boldsymbol{x}_t(\epsilon)), \boldsymbol{\ell}_t(\epsilon) \rangle) - H(\langle \boldsymbol{v}_t^1[\epsilon, g], \boldsymbol{\ell}_t(\epsilon) \rangle) \big) \right].
$$

Introducing the covering elements defined above to this expression, we have the equality

$$
= \mathbb{E}_\epsilon \sup_{g \in \mathcal{G}} \left[ \sum_{t=1}^T \epsilon_t \big( H(\langle g(\boldsymbol{x}_t(\epsilon)), \boldsymbol{\ell}_t(\epsilon) \rangle) - H(\langle \boldsymbol{v}_t^N[\epsilon, g], \boldsymbol{\ell}_t(\epsilon) \rangle) \big) \right.
$$

$$
\left. + \sum_{i=1}^{N-1} \sum_{t=1}^T \epsilon_t \big( H(\langle \boldsymbol{v}_t^{i+1}[\epsilon, g], \boldsymbol{\ell}_t(\epsilon) \rangle) - H(\langle \boldsymbol{v}_t^i[\epsilon, g], \boldsymbol{\ell}_t(\epsilon) \rangle) \big) \right]
$$

$$
\leq \underbrace{\mathbb{E}_\epsilon \sup_{g \in \mathcal{G}} \left[ \sum_{t=1}^T \epsilon_t \big( H(\langle g(\boldsymbol{x}_t(\epsilon)), \boldsymbol{\ell}_t(\epsilon) \rangle) - H(\langle \boldsymbol{v}_t^N[\epsilon, g], \boldsymbol{\ell}_t(\epsilon) \rangle) \big) \right]}_{\triangleq C_N}
$$

$$
+ \sum_{i=1}^{N-1} \underbrace{\mathbb{E}_\epsilon \sup_{g \in \mathcal{G}} \left[ \sum_{t=1}^T \epsilon_t \big( H(\langle \boldsymbol{v}_t^{i+1}[\epsilon, g], \boldsymbol{\ell}_t(\epsilon) \rangle) - H(\langle \boldsymbol{v}_t^i[\epsilon, g], \boldsymbol{\ell}_t(\epsilon) \rangle) \big) \right]}_{\triangleq C_i}.
$$

## C.2.7 Bounding $C_N$

We first bound $C_N$ in terms of one of the terms appearing in $B_1$.

$$
C_N = \mathbb{E}_\epsilon \sup_{g \in \mathcal{G}} \left[ \sum_{t=1}^T \epsilon_t \big( H(\langle g(\boldsymbol{x}_t(\epsilon)), \boldsymbol{\ell}_t(\epsilon) \rangle) - H(\langle \boldsymbol{v}_t^N[\epsilon, g], \boldsymbol{\ell}_t(\epsilon) \rangle) \big) \right]
$$

$$
\leq \mathbb{E}_\epsilon \left[ \sum_{t=1}^T \sup_{g \in \mathcal{G}} \left| H(\langle g(\boldsymbol{x}_t(\epsilon)), \boldsymbol{\ell}_t(\epsilon) \rangle) - H(\langle \boldsymbol{v}_t^N[\epsilon, g], \boldsymbol{\ell}_t(\epsilon) \rangle) \right| \right]
$$

$$
\leq (1 + 2\eta_3) \mathbb{E}_\epsilon \left[ \sum_{t=1}^T \sup_{g \in \mathcal{G}} \left| \langle g(\boldsymbol{x}_t(\epsilon)), \boldsymbol{\ell}_t(\epsilon) \rangle - \langle \boldsymbol{v}_t^N[\epsilon, g], \boldsymbol{\ell}_t(\epsilon) \rangle \right| \right]
$$

$$
\leq (1 + 2\eta_3) \mathbb{E}_\epsilon \left[ \sum_{t=1}^T \left\| \boldsymbol{\ell}_t(\epsilon) \right\|_1 \sup_{g \in \mathcal{G}} \left\| g(\boldsymbol{x}_t(\epsilon)) - \boldsymbol{v}_t^N[\epsilon, g] \right\|_\infty \right]
$$

$$
\leq (1 + 2\eta_3) \max_\epsilon \sup_{g \in \mathcal{G}} \max_{t \in [T]} \left\| g(\boldsymbol{x}_t(\epsilon)) - \boldsymbol{v}_t^N[\epsilon, g] \right\|_\infty \cdot \mathbb{E}_\epsilon \left[ \sum_{t=1}^T \left\| \boldsymbol{\ell}_t(\epsilon) \right\|_1 \right]
$$

$$
\leq (1 + 2\eta_3) e^2 \alpha \cdot \mathbb{E}_\epsilon \left[ \sum_{t=1}^T \left\| \boldsymbol{\ell}_t(\epsilon) \right\|_1 \right].
$$

The first inequality uses that $\epsilon_t \in \{\pm 1\}$, while the second uses the Lipschitzness of $H$ over $[-1, +1]$. The third and fourth are both applications of Hölder's inequality, first to the $\ell_1/\ell_\infty$ dual pairing, and then to for the distributions over $L_1/L_\infty$. Finally, the definition of the covering element $\boldsymbol{v}_t^N$—in particular, that it is an $L_\infty/\ell_\infty$-cover—implies that the supremum term is bounded by $\varepsilon_N \le e^2 \cdot \alpha$, which yields the final bound.

### C.2.8   Bounding $C_i$

Our goal is to bound

$$C_i = \mathbb{E}_\epsilon \sup_{g \in \mathcal{G}} \left[ \sum_{t=1}^T \epsilon_t \big( H\big(\langle \boldsymbol{v}_t^{i+1}[\epsilon, g], \boldsymbol{\ell}_t(\epsilon) \rangle \big) - H\big(\langle \boldsymbol{v}_t^i[\epsilon, g], \boldsymbol{\ell}_t(\epsilon) \rangle \big) \big) \right].$$

We define a class $W$ of real-valued trees as follows. Let $1 \le a \le |V^i|$ and $1 \le b \le |V^{i+1}|$, and fix an arbitrary ordering $\boldsymbol{v}^a \in V^i$ and $\boldsymbol{v}^b \in V^{i+1}$ of the elements of $V^i/V^{i+1}$. For each pair $(a, b)$ define a tree $\boldsymbol{w}^{(a,b)}$ via

$$\boldsymbol{w}_t^{(a,b)}(\epsilon) = \begin{cases} H\big(\langle \boldsymbol{v}_t^b(\epsilon), \boldsymbol{\ell}_t(\epsilon) \rangle\big) - H(\langle \boldsymbol{v}_t^a(\epsilon), \boldsymbol{\ell}_t(\epsilon) \rangle), & \exists g \in \mathcal{G} \text{ s.t. } \boldsymbol{v}^a = \boldsymbol{v}[\epsilon, g]^i, \boldsymbol{v}^b = \boldsymbol{v}[\epsilon, g]^{i+1}, \\ 0, & \text{otherwise.} \end{cases}$$

Then $C_i$ is bounded by

$$\mathbb{E}_\epsilon \sup_{\boldsymbol{w} \in W} \sum_{t=1}^T \epsilon_t \boldsymbol{w}_t(\epsilon).$$

Then [Lemma 14](#) implies that for any fixed $\eta > 0$,

$$\mathbb{E}_\epsilon \sup_{\boldsymbol{w} \in W} \left[ \sum_{t=1}^T \epsilon_t \boldsymbol{w}_t(\epsilon) - \eta \boldsymbol{w}_t^2(\epsilon) \right] \le \frac{\log|W|}{2\eta}.$$

Rearranging and applying subadditivity of the supremum, this implies

$$\mathbb{E}_\epsilon \sup_{\boldsymbol{w} \in W} \sum_{t=1}^T \epsilon_t \boldsymbol{w}_t(\epsilon) \le \eta \cdot \mathbb{E}_\epsilon \sup_{\boldsymbol{w} \in W} \sum_{t=1}^T \boldsymbol{w}_t^2(\epsilon) + \frac{\log|W|}{2\eta}.$$

Optimizing over $\eta$ (which is admissible because the statement above is a deterministic inequality) leads to a further bound of

$$\mathbb{E}_\epsilon \sup_{\boldsymbol{w} \in W} \sum_{t=1}^T \epsilon_t \boldsymbol{w}_t(\epsilon) \le \sqrt{2 \mathbb{E}_\epsilon \sup_{\boldsymbol{w} \in W} \sum_{t=1}^T \boldsymbol{w}_t^2(\epsilon) \cdot \log|W|}.$$

We proceed to bound each term in the square root. For the logarithmic term, by construction we have $|W| \le |V^i||V^{i+1}| \le |V^{i+1}|^2 = \mathcal{N}_{\infty,\infty}(\varepsilon_{i+1}, \mathcal{G}, \boldsymbol{x})^2$.

For the variance, let $\boldsymbol{w}^{(a,b)} \in W$ and the path $\epsilon$ be fixed. There are two cases: Either $\boldsymbol{w}(\epsilon) = \boldsymbol{0}$, or there exists $g \in \mathcal{G}$, such that $\boldsymbol{v}^a = \boldsymbol{v}[\epsilon, g]^i$ and $\boldsymbol{v}^b = \boldsymbol{v}[\epsilon, g]^{i+1}$. The former case is trivial while for the latter, in a similar way to the bound for $C_N$, we get

$$\sum_{t=1}^T \boldsymbol{w}_t^{(a,b)}(\epsilon)^2 = \sum_{t=1}^T \big( H\big(\langle \boldsymbol{v}_t^{i+1}[\epsilon, g], \boldsymbol{\ell}_t(\epsilon) \rangle\big) - H\big(\langle \boldsymbol{v}_t^i[\epsilon, g], \boldsymbol{\ell}_t(\epsilon) \rangle\big) \big)^2$$

$$\le (1 + 2\eta_3)^2 \sum_{t=1}^T \big( \langle \boldsymbol{v}_t^{i+1}[\epsilon, g], \boldsymbol{\ell}_t(\epsilon) \rangle - \langle \boldsymbol{v}_t^i[\epsilon, g], \boldsymbol{\ell}_t(\epsilon) \rangle \big)^2$$

$$\le (1 + 2\eta_3)^2 \sum_{t=1}^T \|\boldsymbol{\ell}_t(\epsilon)\|_1^2 \|\boldsymbol{v}_t^{i+1}[\epsilon, g] - \boldsymbol{v}_t^i[\epsilon, g]\|_\infty^2$$

$$\le (1 + 2\eta_3)^2 \max_{\epsilon'} \max_{t \in [T]} \|\boldsymbol{v}_t^{i+1}[\epsilon', g] - \boldsymbol{v}_t^i[\epsilon', g]\|_\infty^2 \cdot \sum_{t=1}^T \|\boldsymbol{\ell}_t(\epsilon)\|_1^2.$$

Where we have used Lipschitzness of $H$ in the first inequality and Hölder's inequality in the second and third.

Finally, using the $L_\infty/\ell_\infty$ cover property of $\boldsymbol{v}^i[\epsilon, g]$ and $\boldsymbol{v}^{i+1}[\epsilon, g]$ and the triangle inequality, we have

$$\max_\epsilon \max_{t\in[T]} \left\| \boldsymbol{v}^{i+1}_t[\epsilon, g] - \boldsymbol{v}^i_t[\epsilon, g] \right\|_\infty$$

$$\le \max_\epsilon \max_{t\in[T]} \left\| \boldsymbol{v}^{i+1}_t[\epsilon, g] - g(\boldsymbol{x}_t(\epsilon)) \right\|_\infty + \max_\epsilon \max_{t\in[T]} \left\| g(\boldsymbol{x}_t(\epsilon)) - \boldsymbol{v}^i_t[\epsilon, g] \right\|_\infty$$

$$\le \varepsilon_i + \varepsilon_{i+1} \le 2\varepsilon_i.$$

We have just shown that for every sequence $\epsilon$ and every $\boldsymbol{w}^{(a,b)} \in W$, $\sum_{t=1}^T \boldsymbol{w}^{(a,b)}_t(\epsilon)^2 \le 4(1 + 2\eta_3)^2 \varepsilon_i^2 \cdot \sum_{t=1}^T \|\boldsymbol{\ell}_t(\epsilon)\|_1^2$. It follows that

$$\mathbb{E}_\epsilon \sup_{\boldsymbol{w}\in W} \sum_{t=1}^T \boldsymbol{w}_t(\epsilon)^2 \le 4(1+2\eta_3)^2 \varepsilon_i^2 \cdot \mathbb{E}_\epsilon \sum_{t=1}^T \|\boldsymbol{\ell}_t(\epsilon)\|_1^2.$$

Plugging this bound back into the main inequality, we have shown

$$\mathbb{E}_\epsilon \sup_{\boldsymbol{w}\in W} \sum_{t=1}^T \epsilon_t \boldsymbol{w}_t(\epsilon) \le 4e(1+2\eta_3)\varepsilon_{i+1}\sqrt{\mathbb{E}_\epsilon \sum_{t=1}^T \|\boldsymbol{\ell}_t(\epsilon)\|_1^2 \cdot \log\mathcal{N}_{\infty,\infty}(\varepsilon_{i+1}, \mathcal{G}, \boldsymbol{x})}.$$

### C.2.9  Final bound on $(\star\star)$

Collecting terms, we have shown that

$(\star\star)$

$$\le (1+2\eta_3)e^2\alpha \cdot \mathbb{E}_\epsilon\left[\sum_{t=1}^T \|\boldsymbol{\ell}_t(\epsilon)\|_1\right] + 4e(1+2\eta_3)\sqrt{\mathbb{E}_\epsilon \sum_{t=1}^T \|\boldsymbol{\ell}_t(\epsilon)\|_1^2} \sum_{i=1}^{N-1}\varepsilon_{i+1}\sqrt{\log\mathcal{N}_{\infty,\infty}(\varepsilon_{i+1}, \mathcal{G}, \boldsymbol{x})} - \mathbb{E}_\epsilon B_1(\boldsymbol{\ell}_{1:T}(\epsilon)).$$

$$(10)$$

Following the standard Dudley chaining proof, we have

$$\sum_{i=1}^{N-1}\varepsilon_{i+1}\sqrt{\log\mathcal{N}_{\infty,\infty}(\varepsilon_{i+1}, \mathcal{G}, \boldsymbol{x})} \le \sum_{i=1}^N \varepsilon_i\sqrt{\log\mathcal{N}_{\infty,\infty}(\varepsilon_i, \mathcal{G}, \boldsymbol{x})} \le 2\sum_{i=1}^N (\varepsilon_i - \varepsilon_{i+1})\sqrt{\log\mathcal{N}_{\infty,\infty}(\varepsilon_i, \mathcal{G}, \boldsymbol{x})}$$

$$\le 2\int_{\varepsilon_{N+1}}^\beta \sqrt{\log\mathcal{N}_{\infty,\infty}(\varepsilon, \mathcal{G}, \boldsymbol{x})}d\varepsilon \le 2\int_\alpha^\beta \sqrt{\log\mathcal{N}_{\infty,\infty}(\varepsilon, \mathcal{G}, \boldsymbol{x})}d\varepsilon$$

$$\le 2\int_\alpha^\beta \sqrt{\log\mathcal{N}_{\infty,\infty}(\varepsilon, \mathcal{G}, T)}d\varepsilon.$$

Where we are using the definition of $N$, which implies that $\alpha \le \varepsilon_{N+1}$.

Now recall the definition of $B_1(\boldsymbol{\ell}_{1:T}(\epsilon))$:

$$B_1(\boldsymbol{\ell}_{1:T}(\epsilon)) = \eta_1 \sum_{t=1}^T \|\boldsymbol{\ell}_t(\epsilon)\|_1 + \eta_2 \sum_{t=1}^T \|\boldsymbol{\ell}_t(\epsilon)\|_1^2$$

Taking $\eta_1 \ge (1+2\eta_3)e^2\alpha$, the first term in $B_1$ cancels out the first term in (10), leaving us with

$$(\star\star) \le 8e(1+2\eta_3)\sqrt{\mathbb{E}_\epsilon \sum_{t=1}^T \|\boldsymbol{\ell}_t(\epsilon)\|_1^2} \int_\alpha^\beta \sqrt{\log\mathcal{N}_{\infty,\infty}(\varepsilon, \mathcal{G}, T)}d\varepsilon - \eta_2 \mathbb{E}_\epsilon \sum_{t=1}^T \|\boldsymbol{\ell}_t(\epsilon)\|_1^2$$

$$\le 8e(1+2\eta_3)\left(\frac{\eta_4}{4}\mathbb{E}_\epsilon \sum_{t=1}^T \|\boldsymbol{\ell}_t(\epsilon)\|_1^2 + \frac{1}{\eta_4}\right)\int_\alpha^\beta \sqrt{\log\mathcal{N}_{\infty,\infty}(\varepsilon, \mathcal{G}, T)}d\varepsilon - \eta_2 \mathbb{E}_\epsilon \sum_{t=1}^T \|\boldsymbol{\ell}_t(\epsilon)\|_1^2.$$

Where the last step applies for any $\eta_4 > 0$ by the AM-GM inequality. For any $\eta_2 \ge 8e(1+2\eta_3)\eta_4 \cdot \int_\alpha^\beta \sqrt{\log\mathcal{N}_{\infty,\infty}(\varepsilon, \mathcal{G}, T)}d\varepsilon$, the first and third terms cancel, leaving us with an upper bound of

$$(\star\star) \le \frac{8e(1+2\eta_3)}{\eta_4}\int_\alpha^\beta \sqrt{\log\mathcal{N}_{\infty,\infty}(\varepsilon, \mathcal{G}, T)}d\varepsilon.$$

This term does not depend on the trees $\boldsymbol{x}$ or $\boldsymbol{\ell}$, so we are done with $(\star\star)$.

### C.2.10 Final bound

Under the assumptions on $\eta_1, \eta_2, \eta_3, \eta_4, \alpha$, and $\beta$, the bounds on $(\star)$ and $(\star\star)$ we have established imply

$$\mathcal{V} \le 2\frac{1+\eta_3^2}{\eta_3}\log\mathcal{N}_{\infty,\infty}(\beta/2,\mathcal{G},T) + \frac{8e(1+2\eta_3)}{\eta_4}\int_\alpha^\beta\sqrt{\log\mathcal{N}_{\infty,\infty}(\varepsilon,\mathcal{G},T)}d\varepsilon.$$

The definition of $\mathcal{V}$ implies that there exists an algorithm with regret bounded by $\mathcal{V} + B(p_{1:T}, \ell_{1:T})$ on every sequence. The final regret inequality is

$$\sum_{t=1}^{T}\mathbb{E}_{s\sim p_t}\langle s, \ell_t\rangle - \inf_{g\in\mathcal{G}}\sum_{t=1}^{T}\langle g(x_t), \ell_t\rangle$$

$$\le 2\eta_3\sum_{t=1}^{T}\mathbb{E}_{s\sim p_t}\langle s,\ell_t\rangle^2 + 2\frac{1+\eta_3^2}{\eta_3}\log\mathcal{N}_{\infty,\infty}(\beta/2,\mathcal{G},T)$$

$$+ 8e(1+2\eta_3)\left(\frac{\eta_4}{4}\sum_{t=1}^{T}\|\ell_t\|_1^2 + \frac{1}{\eta_4}\right)\int_\alpha^\beta\sqrt{\log\mathcal{N}_{\infty,\infty}(\varepsilon,\mathcal{G},T)}d\varepsilon + (1+2\eta_3)e^2\alpha\sum_{t=1}^{T}\|\ell_t\|_1.$$

To obtain the bound in the theorem statement, we rebind $\eta = \eta_3, \lambda = \eta_4$ and use the assumption $\eta \le 1$.

### C.3 Proofs for remaining results

Our bandit results require a generalization of Theorem 12 to the case where losses and the class $\mathcal{G}$ may not be bounded by 1.

**Corollary 16.** *Suppose we are in the setting of Theorem 12, but with the bounds $\|\ell\|_1 \le R$ for all $\ell \in \mathcal{L}$ and $\|s\|_\infty \le B$ for all $s \in \mathcal{S}$. For any constants $\eta \in (0,1]$, $\lambda > 0$, and $\beta > \alpha > 0$, there exists an algorithm making predictions in $\mathcal{S}$ that attains a regret guarantee of*

$$\sum_{t=1}^{T}\mathbb{E}_{s_t\sim p_t}\langle s_t, \ell_t\rangle - \inf_{g\in\mathcal{G}}\sum_{t=1}^{T}\langle g(x_t), \ell_t\rangle \le \frac{2\eta}{RB}\sum_{t=1}^{T}\mathbb{E}_{s_t\sim p_t}\langle s_t, \ell_t\rangle^2 + \frac{4RB}{\eta}\log\mathcal{N}_{\infty,\infty}(\beta/2,\mathcal{G},T) + 3e^2\alpha\sum_{t=1}^{T}\|\ell_t\|_1$$

$$+ 24e\left(\frac{\lambda}{4R}\sum_{t=1}^{T}\|\ell_t\|_1^2 + \frac{R}{\lambda}\right)\int_\alpha^\beta\sqrt{\log\mathcal{N}_{\infty,\infty}(\varepsilon,\mathcal{G},T)}d\varepsilon.$$

*Furthermore, if upper bounds $\sum_{t=1}^{T}\|\ell_t\|_1^2 \le V$ and $\sum_{t=1}^{T}\mathbb{E}_{s_t\sim p_t}\langle s_t, \ell_t\rangle^2 \le \widetilde{V}$ are known in advance, $\eta$ and $\lambda$ can be selected to guarantee regret*

$$\sum_{t=1}^{T}\mathbb{E}_{s_t\sim p_t}\langle s_t, \ell_t\rangle - \inf_{g\in\mathcal{G}}\sum_{t=1}^{T}\langle g(x_t), \ell_t\rangle$$

$$\le 8\sqrt{\widetilde{V}\cdot\log\mathcal{N}_{\infty,\infty}(\beta/2,\mathcal{T},T)} + 8RB\log\mathcal{N}_{\infty,\infty}(\beta/2,\mathcal{G},T)$$

$$+ 24e\sqrt{V}\int_\alpha^\beta\sqrt{\log\mathcal{N}_{\infty,\infty}(\varepsilon,\mathcal{G},T)}d\varepsilon + 3e\alpha\sum_{t=1}^{T}\|\ell_t\|_1.$$

**Proof of Corollary 16.** Apply Theorem 12 with losses $\ell_t/R$ and class $\mathcal{G}/B$. The preconditions of the theorem are satisified, so it implies existence of an algorithm making predictions in $\mathcal{S}/B$ with regret bound

$$\frac{1}{R}\sum_{t=1}^{T}\mathbb{E}_{s_t\sim p_t}\langle s_t, \ell_t\rangle - \frac{1}{R}\inf_{g'\in\mathcal{G}/B}\sum_{t=1}^{T}\langle g'(x_t), \ell_t\rangle \le \frac{2\eta}{R^2}\sum_{t=1}^{T}\mathbb{E}_{s_t\sim p_t}\langle s_t, \ell_t\rangle^2 + \frac{4}{\eta}\log\mathcal{N}_{\infty,\infty}(\beta/2,\mathcal{G}/B,T) + \frac{3e^2\alpha}{R}\sum_{t=1}^{T}\|\ell_t\|_1$$

$$+ 24e\left(\frac{\lambda}{4R^2}\sum_{t=1}^{T}\|\ell_t\|_1^2 + \frac{1}{\lambda}\right)\int_\alpha^\beta\sqrt{\log\mathcal{N}_{\infty,\infty}(\varepsilon,\mathcal{G}/B,T)}d\varepsilon.$$

Rescaling both sides by $BR$ and letting $\hat{s}_t = s_t \cdot B$ (so $\hat{s}_t \in \mathcal{S}$), this implies

$$\sum_{t=1}^{T} \mathbb{E}_{\hat{s}_t \sim p_t} \langle \hat{s}_t, \ell_t \rangle - \inf_{g \in \mathcal{G}} \sum_{t=1}^{T} \langle g(x_t), \ell_t \rangle \leq \frac{2\eta}{RB} \sum_{t=1}^{T} \mathbb{E}_{\hat{s}_t \sim p_t} \langle \hat{s}_t, \ell_t \rangle^2 + \frac{4RB}{\eta} \log \mathcal{N}_{\infty,\infty}(\beta/2, \mathcal{G}/B, T) + 3e^2 \alpha B \sum_{t=1}^{T} \|\ell_t\|_1$$

$$+ 24e \left( \frac{\lambda B}{4R} \sum_{t=1}^{T} \|\ell_t\|_1^2 + \frac{RB}{\lambda} \right) \int_{\alpha}^{\beta} \sqrt{\log \mathcal{N}_{\infty,\infty}(\varepsilon, \mathcal{G}/B, T)} d\varepsilon.$$

$$\leq \frac{2\eta}{RB} \sum_{t=1}^{T} \mathbb{E}_{\hat{s}_t \sim p_t} \langle \hat{s}_t, \ell_t \rangle^2 + \frac{4RB}{\eta} \log \mathcal{N}_{\infty,\infty}(\beta B/2, \mathcal{G}, T) + 3e^2 \alpha B \sum_{t=1}^{T} \|\ell_t\|_1$$

$$+ 24e \left( \frac{\lambda B}{4R} \sum_{t=1}^{T} \|\ell_t\|_1^2 + \frac{RB}{\lambda} \right) \int_{\alpha}^{\beta} \sqrt{\log \mathcal{N}_{\infty,\infty}(\varepsilon B, \mathcal{G}, T)} d\varepsilon.$$

Using a change of variables in the Dudley integral, we get

$$\leq \frac{2\eta}{RB} \sum_{t=1}^{T} \mathbb{E}_{\hat{s}_t \sim p_t} \langle \hat{s}_t, \ell_t \rangle^2 + \frac{4RB}{\eta} \log \mathcal{N}_{\infty,\infty}(\beta B/2, \mathcal{G}, T) + 3e^2 \alpha B \sum_{t=1}^{T} \|\ell_t\|_1$$

$$+ 24e \left( \frac{\lambda}{4R} \sum_{t=1}^{T} \|\ell_t\|_1^2 + \frac{R}{\lambda} \right) \int_{\alpha B}^{\beta B} \sqrt{\log \mathcal{N}_{\infty,\infty}(\varepsilon, \mathcal{G}, T)} d\varepsilon.$$

The final result follows by rebinding $\alpha' = \alpha B$ and $\beta' = \beta B$.

For the second claim, apply the upper bounds to obtain

$$\frac{2\eta}{RB} \tilde{V} + \frac{4RB}{\eta} \log \mathcal{N}_{\infty,\infty}(\beta/2, \mathcal{G}, T) + 3e^2 \alpha B \sum_{t=1}^{T} \|\ell_t\|_1$$

$$+ 24e \left( \frac{\lambda}{4R} V + \frac{R}{\lambda} \right) \int_{\alpha}^{\beta} \sqrt{\log \mathcal{N}_{\infty,\infty}(\epsilon, \mathcal{G}, T)} d\epsilon.$$

Now set $\lambda = 2R/\sqrt{V}$ and $\eta = \sqrt{2}RB\sqrt{\log \mathcal{N}_{\infty,\infty}(\beta/2, \mathcal{G}, T)/\tilde{V}} \wedge 1$ to obtain the claimed bound. Note that the range term arises from the constraint that $\eta \in (0, 1]$. $\qquad\square$

**Proof of Theorem 4.** Recall that we use the reduction:

- Initialize full information algorithm whose existence is guaranteed by Theorem 12 with $\mathcal{G} = \phi^\gamma \circ \mathcal{F}$:

- For time $t = 1, \dots, T$:

    - Receive $x_t$ and define $P_t(a) \triangleq \mathbb{E}_{s_t \sim p_t} \frac{s_t(a)}{\sum_{a' \in [K]} s_t(a')}$, where $p_t$ is the output of the full information algorithm at time $t$.
    - Sample action $a_t \sim P_t^\mu$ and feed importance-weighted loss $\hat{\ell}_t(a) = \mathbf{1}\{a_t = a\}\ell_t(a)/P_t^\mu(a)$ into the full information algorithm.

With this setup, Corollary 16 guarantees that the following deterministic regret inequality holds for every sequence of outcomes (i.e. for every sequence $a_1, \dots, a_T$ sampled by the algorithm):

$$\sum_{t=1}^{T} \mathbb{E}_{s_t \sim p_t} \langle s_t, \hat{\ell}_t \rangle - \inf_{f \in \mathcal{F}} \sum_{t=1}^{T} \langle \phi^\gamma(f(x_t)), \hat{\ell}_t \rangle$$

$$\leq \frac{2\eta}{RB} \sum_{t=1}^{T} \mathbb{E}_{s_t \sim p_t} \langle s_t, \hat{\ell}_t \rangle^2 + \frac{4RB}{\eta} \log \mathcal{N}_{\infty,\infty}(\beta/2, \phi^\gamma \circ \mathcal{F}, T) + 3e^2 \alpha \sum_{t=1}^{T} \|\hat{\ell}_t\|_1$$

$$+ 24e \left( \frac{\lambda}{4R} \sum_{t=1}^{T} \|\hat{\ell}_t\|_1^2 + \frac{R}{\lambda} \right) \int_{\alpha}^{\beta} \sqrt{\log \mathcal{N}_{\infty,\infty}(\varepsilon, \phi^\gamma \circ \mathcal{F}, T)} d\varepsilon,$$

where the boundedness of the ramp loss implies $B \leq 1$ and the smoothing factor $\mu$ in $P_t^\mu$ guarantees $R \leq 1/\mu$. Taking expectation over the draw of $a_1, \dots, a_T$, for any fixed $f \in \mathcal{F}$ we obtain the inequality

$$\mathbb{E}\left[ \sum_{t=1}^{T} \mathbb{E}_{s_t \sim p_t} \langle s_t, \hat{\ell}_t \rangle - \sum_{t=1}^{T} \langle \phi^\gamma(f(x_t)), \hat{\ell}_t \rangle \right]$$

$$\leq \mathbb{E}\left[ \frac{2\eta}{1/\mu} \sum_{t=1}^{T} \mathbb{E}\left[ \mathbb{E}_{s_t \sim p_t} \langle s_t, \hat{\ell}_t \rangle^2 \mid \mathcal{J}_t \right] + \frac{4}{\eta\mu} \log \mathcal{N}_{\infty,\infty}(\beta/2, \phi^\gamma \circ \mathcal{F}, T) + 3e^2 \alpha \sum_{t=1}^{T} \mathbb{E}\left[ \|\hat{\ell}_t\|_1 \mid \mathcal{J}_t \right] \right.$$

$$\left. + 24e \left( \frac{\lambda}{4/\mu} \sum_{t=1}^{T} \mathbb{E}\left[ \|\hat{\ell}_t\|_1^2 \mid \mathcal{J}_t \right] + \frac{1}{\lambda\mu} \right) \int_{\alpha}^{\beta} \sqrt{\log \mathcal{N}_{\infty,\infty}(\varepsilon, \phi^\gamma \circ \mathcal{F}, T)} d\varepsilon \right],$$

where the filtration $\mathcal{J}_t$ is defined as in Lemma 3. Using that the importance weighted losses are unbiased, we have that the left-hand side is equal to

$$\mathbb{E}\left[\sum_{t=1}^{T}\mathbb{E}_{s_t \sim p_t}\langle s_t, \ell_t\rangle - \sum_{t=1}^{T}\langle \phi^\gamma(f(x_t)), \ell_t\rangle\right].$$

We also have the following three properties, where the first two use that $\hat{\ell}_t$ is 1-sparse, and the last follows from Lemma 3:

1. $\mathbb{E}\left[\|\hat{\ell}_t\|_1 \mid \mathcal{J}_t\right] = \sum_{a\in[K]} P_t^\mu(a)\hat{\ell}_t(a) = \sum_{a\in[K]} \ell_t(a) \le K.$

2. $\mathbb{E}\left[\|\hat{\ell}_t\|_1^2 \mid \mathcal{J}_t\right] = \sum_{a\in[K]} P_t^\mu(a)\hat{\ell}_t^2(a) = \sum_{a\in[K]} \frac{\ell_t(a)}{P_t^\mu(a)} \le \frac{K}{\mu}.$

3. $\mathbb{E}\left[\mathbb{E}_{s_t\sim p_t}\langle s_t, \hat{\ell}_t\rangle^2 \mid \mathcal{J}_t\right] \le K^2.$

Together, these facts yield the bound

$$\mathbb{E}\left[\sum_{t=1}^{T}\mathbb{E}_{s_t \sim p_t}\langle s_t, \ell_t\rangle - \sum_{t=1}^{T}\langle \phi^\gamma(f(x_t)), \ell_t\rangle\right] \le \frac{2\eta}{1/\mu}K^2 T + \frac{4}{\eta\mu}\log\mathcal{N}_{\infty,\infty}(\beta/2, \phi^\gamma\circ\mathcal{F}, T) + 3e^2\alpha KT$$
$$+ 24e\left(\frac{\lambda KT}{4} + \frac{1}{\lambda\mu}\right)\int_\alpha^\beta \sqrt{\log\mathcal{N}_{\infty,\infty}(\varepsilon, \phi^\gamma\circ\mathcal{F}, T)}d\varepsilon.$$

Optimizing $\eta$ and $\lambda$ (as in the proof of the second claim of Corollary 16) leads to a bound of

$$\mathbb{E}\left[\sum_{t=1}^{T}\mathbb{E}_{s_t \sim p_t}\langle s_t, \ell_t\rangle - \sum_{t=1}^{T}\langle \phi^\gamma(f(x_t)), \ell_t\rangle\right]$$
$$\le 4\sqrt{2K^2 T\log\mathcal{N}_{\infty,\infty}(\beta/2, \phi^\gamma\circ\mathcal{F}, T)} + \frac{8}{\mu}\log\mathcal{N}_{\infty,\infty}(\beta/2, \phi^\gamma\circ\mathcal{F}, T)$$
$$+ 3e^2\alpha KT + 24e\sqrt{\frac{KT}{\mu}}\int_\alpha^\beta \sqrt{\log\mathcal{N}_{\infty,\infty}(\varepsilon, \phi^\gamma\circ\mathcal{F}, T)}d\varepsilon.$$

Since $\phi^\gamma$ is $\frac{1}{\gamma}$-Lipschitz with respect to the $\ell_\infty$ norm (as a coordinate-wise mapping from $\mathbb{R}^K$ to $\mathbb{R}^K$), we can upper bound in terms of the covering numbers for the original class:

$$\mathbb{E}\left[\sum_{t=1}^{T}\mathbb{E}_{s_t \sim p_t}\langle s_t, \ell_t\rangle - \sum_{t=1}^{T}\langle \phi^\gamma(f(x_t)), \ell_t\rangle\right] \le 4\sqrt{2K^2 T\log\mathcal{N}_{\infty,\infty}(\gamma\beta/2, \mathcal{F}, T)} + \frac{8}{\mu}\log\mathcal{N}_{\infty,\infty}(\gamma\beta/2, \mathcal{F}, T)$$
$$+ 3e^2\alpha KT + 24e\sqrt{\frac{KT}{\mu}}\int_\alpha^\beta \sqrt{\log\mathcal{N}_{\infty,\infty}(\gamma\varepsilon, \mathcal{F}, T)}d\varepsilon.$$

Using a change of variables and the reparameterization $\alpha' = \alpha\gamma$, $\beta' = \beta\gamma$, the right hand side equals

$$4\sqrt{2K^2 T\log\mathcal{N}_{\infty,\infty}(\beta'/2, \mathcal{F}, T)} + \frac{8}{\mu}\log\mathcal{N}_{\infty,\infty}(\beta'/2, \mathcal{F}, T)$$
$$+ \frac{1}{\gamma}\left(3e^2\alpha KT + 24e\sqrt{\frac{KT}{\mu}}\int_{\alpha'}^{\beta'} \sqrt{\log\mathcal{N}_{\infty,\infty}(\varepsilon, \mathcal{F}, T)}d\varepsilon\right).$$

Lastly, via Lemma 1, we have

$$\sum_{t=1}^{T}\mathbb{E}_{s_t \sim p_t}\langle s_t, \ell_t\rangle \ge \sum_{t=1}^{T}\mathbb{E}_{s_t \sim p_t}\frac{\sum_{a\in[K]} s_t(a)\ell_t(a)}{\sum_{a\in[K]} s_t(a)} = \sum_{t=1}^{T}\mathbb{E}_{a_t \sim P_t}\ell_t(a_t).$$

Finally, the definition of the smoothed distribution $P_t^\mu$ and boundedness of $\ell$ immediately implies

$$\sum_{t=1}^{T}\mathbb{E}_{a_t \sim P_t}\ell_t(a_t) \ge \sum_{t=1}^{T}\mathbb{E}_{a_t \sim P_t^\mu}\ell_t(a_t) - \mu KT. \qquad \square$$

**Proof of Proposition 5.** Suppose $\log\mathcal{N}_{\infty,\infty}(\varepsilon, \mathcal{F}, T) \propto \varepsilon^{-p}$.

- When $p \geq 2$, it suffices to set $\beta = \operatorname{rad}_{\infty,\infty}(\mathcal{F}, T)$, $\mu = (KT)^{-1/(p+1)}\gamma^{-p/(p+1)}$, and $\alpha = 1/(KT\mu)^{1/p}$ in Theorem 4 to obtain $\widetilde{O}\big((KT/\gamma)^{p/(p+1)}\big)$.

- When $p \in (0, 2]$, it suffices to set $\alpha = 1/(KT)$, $\mu = (KT)^{-2/(p+4)}\gamma^{-2p/(4+p)}$, and $\beta = \gamma^{2/(2+p)}/(KT\mu)^{1/(2+p)}$ in Theorem 4 to obtain $\widetilde{O}\big((KT)^{(p+2)/(p+4)}\gamma^{-2p/(p+4)}\big)$.

For the parametric case, set $\alpha = \beta = \gamma/KT$ and $\mu = \sqrt{d\log(KT/\gamma)/KT}$ to conclude the bound.

Similarly, in the finite class case, set $\alpha = \beta = 0$ and $\mu = \sqrt{\log|\Pi|/KT}$. $\qquad\square$

**Proof of Example 3.** Let $\mathcal{F}|_a = \{x \mapsto f(x)_a \mid f \in \mathcal{F}\}$. Then clearly it holds that
$$\log \mathcal{N}_{\infty,\infty}(\varepsilon, \mathcal{F}, T) \leq \sum_{a \in [K]} \log \mathcal{N}_{\infty}(\varepsilon, \mathcal{F}|_a, T) \leq K \max_{a \in [K]} \log \mathcal{N}_{\infty}(\varepsilon, \mathcal{F}|_a, T),$$
where have dropped the second "$\infty$" subscript on the right-hand side to denote that this is the covering number for a scalar-valued class. Let $a^\star$ be the action that obtains the maximum in this expression. Returning to the integral expression in Theorem 4, we have just shown an upper bound of
$$3e^2\alpha KT + 24eK\sqrt{\frac{T}{\mu}}\int_\alpha^\beta \sqrt{\log \mathcal{N}_{\infty}(\varepsilon, \mathcal{F}|_{a^\star}, T)}d\varepsilon.$$
For any scalar-value function class $\mathcal{G} \subseteq (\mathcal{X} \to [0,1])$, define
$$\mathfrak{R}(\mathcal{G}, T) = \sup_{\boldsymbol{x}} \mathbb{E}_\epsilon \sup_{g \in \mathcal{G}} \sum_{t=1}^T \epsilon_t g(\boldsymbol{x}_t(\epsilon)).$$
Following the proof of Lemma 9 in Rakhlin et al. [42], by choosing $\beta = 1$ and $\alpha = 2\mathfrak{R}(\mathcal{F}|_{a^\star}, T)/T$, we may upper bound the $L_\infty$ covering number by the sequential Rademacher complexity (via fat-shattering), to obtain
$$6eK\mathfrak{R}(\mathcal{F}|_{a^\star}, T) + 96\sqrt{2}eK\sqrt{\frac{1}{\mu}\mathfrak{R}(\mathcal{F}|_{a^\star}, T)}\int_{2\mathfrak{R}(\mathcal{F}|_{a^\star}, T)/T}^1 \frac{1}{\varepsilon}\sqrt{\log(2eT/\varepsilon)}d\varepsilon.$$
Using straightforward calculation from the proof of Lemma 9 in Rakhlin et al. [42], this is upper bounded by
$$O\bigg(\frac{K}{\sqrt{\mu}}\mathfrak{R}(\mathcal{F}|_{a^\star}, T)\log^{3/2}(T/\mathfrak{R}(\mathcal{F}|_{a^\star}, T))\bigg).$$
Returning to the regret bound in Theorem 4, we have shown an upper bound of
$$O\bigg(\frac{K}{\gamma\sqrt{\mu}}\mathfrak{R}(\mathcal{F}|_{a^\star}, T)\log^{3/2}(T/\mathfrak{R}(\mathcal{F}|_{a^\star}, T)) + \mu KT\bigg),$$
where we have used that $\log \mathcal{N}_{\infty}(1, \mathcal{F}|_{a^\star}, T) = 0$ under the boundedness assumption on $\mathcal{F}$. Setting $\mu \propto (\mathfrak{R}(\mathcal{F}|_{a^\star}, T)/(T\gamma))^{2/3}$ yields the result. $\qquad\square$

**Proof of Example 4.** This is an immediate consequence of Example 3 and that Banach spaces for which the martingale type property holds with constant $\beta$ have sequential Rademacher complexity $O(\sqrt{\beta T})$ [45]. $\qquad\square$

## C.4 Additional results

Here we briefly state an analogue of Theorem 4 for the hinge loss. Note that this bound leads to the same exponents for $T$ as Theorem 4, but has worse dependence on the margin $\gamma$ and depends on the scale parameter $B$ explicitly.

**Theorem 17** (Contextual bandit chaining bound for hinge loss)**.** *For any fixed constants $\beta > \alpha > 0$, hinge loss parameter $\gamma > 0$, and smoothing parameter $\mu \in (0, 1/K]$ there exists an adversarial contextual bandit strategy $(P_t)_{t \leq T}$ with expected regret bounded as*

$$\mathbb{E}\bigg[\sum_{t=1}^T \ell_t(a_t)\bigg] \leq \frac{1}{K}\bigg\{\inf_{f \in \mathcal{F}} \mathbb{E}\bigg[\sum_{t=1}^T \langle \psi^\gamma(f(x_t)), \ell_t\rangle\bigg] + \frac{1}{\gamma}\sqrt{2K^2B^2T\log\mathcal{N}_{\infty,\infty}(\beta/2, \mathcal{F}, T)} + \mu BK^2T$$

$$+ \frac{8B}{\gamma\mu}\log\mathcal{N}_{\infty,\infty}(\beta/2, \mathcal{F}, T) + \frac{1}{\gamma}\bigg(3e\alpha KT + 24e\sqrt{\frac{KT}{\mu}}\int_\alpha^\beta \sqrt{\log\mathcal{N}_{\infty,\infty}(\varepsilon, \mathcal{F}, T)}d\varepsilon\bigg)\bigg\},$$

*where we recall $B = \sup_{f \in \mathcal{F}} \sup_{f \in \mathcal{X}} \|f(x)\|_\infty$.*

## D    Analysis of HINGE-LMC

This appendix contains the proofs of Theorem 6 and the corresponding corollaries. The proof has many ingredients which we compartmentalize into subsections. First, in Subsection D.1, we analyze the sampling routine, showing that Langevin Monte Carlo can be used to generate a sample from an approximation of the exponential weights distribution. Then, in Subsection D.2, we derive the regret bound for the continuous version of exponential weights. Finally, we put the components together together, instantiate all parameters, and compute the final regret and running time in Subsection D.3. The corollaries are straightforward and proved in Subsection D.4

To begin, we restate the main theorem, with all the assumptions and the precise parameter settings.

**Theorem 18.** *Let $\mathcal{F}$ be a set of functions parameterized by a compact convex set $\Theta \subset \mathbb{R}^d$ that contains the origin-centered Euclidean ball of radius $1$ and is contained within a Euclidean ball of radius $R$. Assume that $f(x; \theta)$ is convex in $\theta$ for each $x \in \mathcal{X}$, and that $\sup_{x,\theta} \|f(x;\theta)\|_\infty \le B$, that $f(x, a; \theta)$ is L-Lipschitz as a function of $\theta$ with respect to the $\ell_2$ norm for each $x, a$. For any $\gamma$, if we set*

$$\eta = \sqrt{\frac{d\gamma^2 \log(RLTK/\gamma)}{5K^2B^2T}}, \qquad \mu = \sqrt{\frac{1}{K^2T}}, \qquad M = \sqrt{T},$$

*in HINGE-LMC, and further set*

$$u = \frac{1}{T^{3/2}LB_\ell R\eta\sqrt{d}}, \qquad \lambda = \frac{1}{8T^{1/2}R^3}, \qquad \alpha = \frac{R^2}{N},$$

$$N = \tilde{\mathcal{O}}\left(R^{18}L^{12}T^6d^{12} + \frac{R^{24}L^{48}d^{12}}{K^{24}}\right), \qquad m = \tilde{\mathcal{O}}\left(T^3dR^4L^2B_\ell^2(K\gamma)^{-2}\right),$$

*in each call to Projected LMC, then HINGE-LMC guarantees*

$$\sum_{t=1}^T \mathbb{E}\ell_t(a_t) \le \min_{\theta \in \Theta} \sum_{t=1}^T \mathbb{E}\langle \ell_t, \psi^\gamma(f(x_t; \theta))\rangle + \frac{\sqrt{T}}{\gamma} + \frac{2d}{K\eta}\log(RLTK/\gamma) + \frac{10\eta}{\gamma^2}B^2KT$$

$$\le \min_{\theta \in \Theta} \sum_{t=1}^T \mathbb{E}\langle \ell_t, \psi^\gamma(f(x_t; \theta))\rangle + \tilde{\mathcal{O}}\left(\frac{B}{\gamma}\sqrt{dT}\right).$$

*Moreover, the running time of HINGE-LMC is $\tilde{\mathcal{O}}\left(\frac{R^{22}L^{14}d^{14}B_\ell^2T^{10}}{K^2\gamma^2} + \frac{R^{28}L^{50}d^{14}B_\ell^2T^4}{K^{26}\gamma^2}\right)$.*

### D.1    Analysis of the sampling routine

In this section, we show how Projected LMC can be used to generate a sample from a distribution that is close to the exponential weights distribution. Define

$$F(\theta) = \eta \sum_{\tau=1}^t \langle \tilde{\ell}_\tau, \psi^\gamma(f(x_\tau; \theta))\rangle, \quad P(\theta) \propto \exp(-F(\theta)). \tag{11}$$

We are interested in sampling from $P(\theta)$.

Let us define the Wasserstein distance. For random variables $X, Y$ with density $\mu, \nu$ respectively

$$\mathcal{W}_1(\mu, \nu) \triangleq \inf_{\pi \in \Gamma(\mu,\nu)} \int \|X - Y\|_2 d\pi(X, Y) = \sup_{f:\text{Lip}(f)\le 1} \left|\int f(d\mu(X) - d\nu(Y))\right|.$$

Here $\Gamma(\mu, \nu)$ is the set of couplings between the two densities, that is the set of joint distributions with marginals equal to $\mu, \nu$. $\text{Lip}(f)$ is the set of all functions that are 1-Lipschitz with respect to $\ell_2$.

**Theorem 19.** *Let $\Theta \subset \mathbb{R}^d$ be a convex set containing a Euclidean ball of radius $r = 1$ with center $0$, and contained within a Euclidean ball of radius $R$. Let $f : \mathcal{X} \times \Theta \to \mathbb{R}_{\ge 0}^K$ be convex in $\theta$ with $f_a(x; \cdot)$ being L-Lipschitz w.r.t. $\ell_2$ norm for each $a \in \mathcal{A}$. Assume $\|\tilde{\ell}_\tau\|_1 \le B_\ell$ and define*

---

**Algorithm 3** Smoothed Projected Langevin Monte Carlo for (11)

---

Input: Parameters $m, u, \lambda, N, \alpha$.

Set $\tilde{\theta}_0 \leftarrow 0 \in \mathbb{R}^d$

**for** $k = 1, \dots, N$ **do**

Sample $z_1, \dots, z_m \overset{iid}{\sim} \mathcal{N}(0, u^2 I_d)$ and form the function

$$\tilde{F}_k(\theta) = \frac{1}{m} \sum_{i=1}^m F(\theta + z_i) + \frac{\lambda}{2} \|\theta\|_2^2.$$

Sample $\xi_k \sim \mathcal{N}(0, I_d)$ and update

$$\tilde{\theta}_k \leftarrow \mathcal{P}_\Theta \left( \tilde{\theta}_{k-1} - \frac{\alpha}{2} \nabla \tilde{F}_k(\tilde{\theta}_{k-1}) + \sqrt{\alpha} \xi_k \right).$$

**end for**

Return $\tilde{\theta}_N$.

---

*$F$ and $P$ as in (11). Let a target accuracy $\tau > 0$ be fixed. Then Algorithm 3 with parameters $m, N, \lambda, u, \alpha \in poly(1/\tau, d, R, \eta, B_\ell, L)$ generates a sample from a distribution $\tilde{P}$ satisfying*

$$\mathcal{W}_1(\tilde{P}, P) \le \tau.$$

*Therefore, the algorithm runs in polynomial time.*

The precise values for each of the parameters $m, N, u, \lambda, \alpha$ can be found at the end of the proof, which will lead to a setting of $\tau$ in application of the theorem.

Towards the proof, we will introduce the intermediate function $\hat{F}(\theta) = \mathbb{E}_Z F(\theta + Z) + \frac{\lambda}{2} \|\theta\|_2^2$, where $Z$ is a random variable with distribution $\mathcal{N}(0, u^2 I_d)$. This is the randomized smoothing technique studied by Duchi, Bartlett and Wainwright [19]. The critical properties of this function are

**Proposition 20** (Properties of $\hat{F}$). *Under the assumptions of Theorem 19, The function $\hat{F}$ satisfies*

1. $F(\theta) \le \hat{F}(\theta) \le F(\theta) + \eta T B_\ell L u \sqrt{d}/\gamma + \frac{\lambda}{2} R^2$.

2. $\hat{F}(\theta)$ is $\eta T B_\ell L / \gamma + \lambda R$-Lipschitz with respect to the $\ell_2$ norm.

3. $\hat{F}(\theta)$ is continuously differentiable and its gradient is $\frac{\eta T B_\ell L}{u\gamma} + \lambda$-Lipschitz continuous with respect to the $\ell_2$ norm.

4. $\hat{F}(\theta)$ is $\lambda$-strongly convex with respect to the $\ell_2$ norm.

5. $\mathbb{E} \nabla F(\theta + Z) = \nabla \hat{F}(\theta)$.

**Proof.** See Duchi et al. [19, Lemma E.3] for the proof of all claims, except for claim 4, which is an immediate consequence of the $\ell_2$ regularization term. □

Using property 1 in Proposition 20 and setting $\varepsilon_1 \triangleq \eta T B_\ell L u \sqrt{d}/\gamma + \lambda R^2$, we know that

$$e^{-\varepsilon_1} \exp(-F(\theta)) \le \exp(-\hat{F}(\theta)) \le \exp(-F(\theta)),$$

pointwise. Therefore, defining $\hat{P}$ to be the distribution with density $\hat{p}(\theta) = \exp(-\hat{F}(\theta))/\hat{Z}$, where $\hat{Z} = \int \exp(-\hat{F}(\theta))d\theta$, we have

$$TV(P \| \hat{P}) = \int \frac{e^{-F(\theta)}}{Z} \left| \frac{e^{-\hat{F}(\theta) + F(\theta)}}{\hat{Z}/Z} - 1 \right| d\theta \le e^{\varepsilon_1} - 1 \le 2\varepsilon_1,$$

for $\varepsilon_1 \le 1$. This shows that $\hat{P}$ approximates $P$ well when $u$ and $\lambda$ are sufficiently small. The next lemma further shows that the $\tilde{F}_k$ functions themselves approximate $\hat{F}$ well.

**Lemma 21** (Properties of $\tilde{F}_k$)**.** *For any fixed $\theta$, $k \in [N]$, and constant $\varepsilon_2 > 0$,*

$$\mathbb{P}\left[\left\|\nabla\hat{F}(\theta) - \nabla\tilde{F}_k(\theta)\right\|_2 \geq \varepsilon_2 + \frac{2}{\sqrt{m}} \cdot \frac{\eta T B_\ell L}{\gamma}\right] \leq \exp\left(\frac{-4\varepsilon_2^2\gamma^2 m}{(\eta TLB_\ell)^2}\right).$$

**Proof of Lemma 21.** Let $k$ be fixed. Since $\tilde{F}_k$ are identically distributed for all $k$ we will henceforth abbreviate to $\tilde{F}$.

We proceed using a crude concentration argument. Observe that by Proposition 20, $\mathbb{E}\nabla\tilde{F}(\theta) = \nabla\hat{F}(\theta)$ and moreover $\nabla\tilde{F}(\theta)$ is a sum of $m$ i.i.d., vector-valued random variables (plus the deterministic regularization term).

Via the Chernoff method, for any fixed $\theta$, we have

$$\mathbb{P}\left[\|\nabla\tilde{F}(\theta) - \nabla\hat{F}(\theta)\|_2 \geq t\right] \leq \inf_{\beta>0}\exp(-t\beta)\mathbb{E}\exp(\beta\|\nabla\tilde{F}(\theta) - \nabla\hat{F}(\theta)\|_2)$$

Using the sum structure and symmetrizing:

$$\leq \inf_{\beta>0}\exp(-t\beta)\mathbb{E}_{z_{1:m}}\mathbb{E}_\epsilon\exp\left(2\beta\left\|\frac{1}{m}\sum_{i=1}^m \epsilon_t\nabla G(\theta + z_i)\right\|_2\right),$$

where $G(\theta) = \eta\sum_{\tau=1}^t\langle\tilde{\ell}_\tau, \psi^\gamma(f(x_\tau;\theta))\rangle$. Condition on $z_{1:m}$ and let $W(\epsilon) = \left\|\frac{1}{m}\sum_{i=1}^m \epsilon_i\nabla G(\theta + z_i)\right\|_2$. Then for any $i$,

$$|W(\epsilon_1, \ldots, \epsilon_i, \ldots, \epsilon_m) - W(\epsilon_1, \ldots, -\epsilon_i, \ldots, \epsilon_m)| \leq \frac{1}{m}\|\nabla G(\theta + z_i)\|_2$$

$$\leq \frac{\eta}{m}\sum_{\tau=1}^t\|\tilde{\ell}_\tau\|_1\|\nabla\psi^\gamma(f(x_\tau;\theta + z_i))_a\|_2$$

$$\leq \frac{\eta T B_\ell L}{m\gamma}.$$

By the standard bounded differences argument (e.g. [11]), this implies that $W - \mathbb{E}W$ is subgaussian with variance proxy $\sigma^2 = \frac{1}{4m}\left(\frac{\eta T B_\ell L}{\gamma}\right)^2$. Furthermore, the standard application of Jensen's inequality implies that $\mathbb{E}W \leq 2\sigma$.

Returning to the upper bound, these facts together imply

$$\mathbb{E}_\epsilon\exp\left(2\beta\left\|\frac{1}{m}\sum_{i=1}^m \epsilon_t\nabla G(\theta + z_i)\right\|_2\right) \leq \exp(2\beta^2\sigma^2 + 4\beta\sigma).$$

The final bound is therefore,

$$\mathbb{P}\left[\|\nabla\tilde{F}(\theta) - \nabla\hat{F}(\theta)\|_2 \geq t\right] \leq \inf_{\beta>0}\exp(-t\beta + 2\beta^2\sigma^2 + 4\beta\sigma).$$

Rebinding $t = t' + 4\sigma$ for $t' \geq 0$, we have

$$\mathbb{P}\left[\|\nabla\tilde{F}(\theta) - \nabla\hat{F}(\theta)\|_2 \geq t' + 4\sigma\right] \leq \inf_{\beta>0}\exp(-t'\beta + 2\beta^2\sigma^2) = \exp(-(t')^2/8\sigma^2).$$

$\square$

Now, for the purposes of the proof, suppose we run the Projected LMC algorithm on the function $\hat{F}$, which generates the iterate sequence $\hat{\theta}_0 = 0$

$$\hat{\theta}_k \leftarrow \mathcal{P}_\Theta\left(\hat{\theta}_{k-1} - \frac{\alpha}{2}\nabla\hat{F}(\hat{\theta}_{k-1}) + \sqrt{\alpha}\xi_k\right).$$

Owing to the smoothness of $\hat{F}$, we may apply the analysis of Projected LMC due to Bubeck, Eldan, and Lehec [12] to bound the total variation distance between the random variable $\hat{\theta}_N$ and the distribution with density proportional to $\exp(-\hat{F}(\theta))$.

**Theorem 22** ([12]). *Let $\hat{P}$ be the distribution on $\Theta$ with density proportional to $\exp(-\hat{F}(\theta))$. For any $\varepsilon > 0$ and with $\alpha = \tilde{\Theta}(R^2/N)$, we have $TV(\hat{\theta}_N, \hat{P}) \leq \varepsilon$ with*

$$N \geq \tilde{\Omega}\left(\frac{R^6 \max\{d, R\eta TB_\ell L/\gamma + R^2\lambda, R(\eta TB_\ell L/(u\gamma) + \lambda)\}^{12}}{\varepsilon^{12}}\right).$$

This specializes the result of Bubeck et al. [12] to our setting, using the Lipschitz and smoothness constants from Proposition 20.

Unfortunately, since we do not have access to $\hat{F}$ in closed form, we cannot run the Projected LMC algorithm on it exactly. Instead, Algorithm 3 runs LMC on the sequence of approximations $\tilde{F}_k$ and generates the iterate sequence $\tilde{\theta}_k$. The last step in the proof is to relate our iterate sequence $\tilde{\theta}_k$ to a hypothetical iterate sequence $\hat{\theta}_k$ formed by running Projected LMC on the function $\hat{F}$.

**Lemma 23.** *Let $\varepsilon_2$ be fixed. Assume the conditions of Theorem 19—in particular that*

$$m \geq 16(\eta TLB_\ell/\gamma)^2 \log(4R/\alpha\varepsilon_2)/\varepsilon_2^2, \qquad \alpha \leq 2(\eta TB_\ell L/(u\gamma) + \lambda)^{-1}.$$

*Then for any $k \in [N]$ we have*

$$\mathcal{W}_1(\hat{\theta}_k, \tilde{\theta}_k) \leq k\alpha\varepsilon_2.$$

**Proof of Lemma 23.** The proof is by induction, where the base case is obvious, since $\hat{\theta}_0 = \tilde{\theta}_0$. Now, let $\pi_{k-1}^\star$ denote the optimal coupling for $\tilde{\theta}_{k-1}, \hat{\theta}_{k-1}$ and extend this coupling in the obvious way by sampling $z_1, \ldots, z_m$ i.i.d. and by using the same gaussian random variable $\xi_k$ in both LMC updates. Let $\mathcal{E}_k = \{z_{1:m} : \|\nabla\tilde{F}(\tilde{\theta}_{k-1}) - \nabla\hat{F}(\tilde{\theta}_{k-1})\| \leq \varepsilon_2 + \varepsilon'\}$, where $\varepsilon' \triangleq \frac{2}{\sqrt{m}} \cdot \frac{\eta TB_\ell L}{\gamma}$; this is the "good" event in which the samples provide a high-quality approximation to the gradient at $\tilde{\theta}_{k-1}$. We then have

$$\mathcal{W}_1(\hat{\theta}_k, \tilde{\theta}_k)$$

$$= \inf_{\pi \in \Gamma(\hat{\theta}_k, \tilde{\theta}_k)} \int \|\hat{\theta}_k - \tilde{\theta}_k\|_2 d\pi$$

$$\leq \int \mathbb{E}_{z_{1:m}}(\mathbf{1}\{\mathcal{E}_k\} + \mathbf{1}\{\mathcal{E}_k^C\})\|\mathcal{P}_\Theta(\hat{\theta}_{k-1} - \frac{\alpha}{2}\nabla\hat{F}(\hat{\theta}_{k-1}) - \sqrt{\alpha}\xi_k) - \mathcal{P}_\Theta(\tilde{\theta}_{k-1} - \frac{\alpha}{2}\nabla\tilde{F}(\tilde{\theta}_{k-1}) - \sqrt{\alpha}\xi_k)\|_2 d\pi_{k-1}^\star$$

$$\leq \int \mathbb{E}_{z_{1:m}}\mathbf{1}\{\mathcal{E}_k\}\|\hat{\theta}_{k-1} - \frac{\alpha}{2}\nabla\hat{F}(\hat{\theta}_{k-1}) - (\tilde{\theta}_{k-1} - \frac{\alpha}{2}\nabla\tilde{F}(\tilde{\theta}_{k-1}))\|_2 d\pi_{k-1}^\star + 2R\int \mathbb{P}[\mathcal{E}_{k-1}^C]d\pi_{k-1}^\star$$

$$\leq \int \mathbb{E}_{z_{1:m}}\mathbf{1}\{\mathcal{E}_k\}\|\hat{\theta}_{k-1} - \frac{\alpha}{2}\nabla\hat{F}(\hat{\theta}_{k-1}) - (\tilde{\theta}_{k-1} - \frac{\alpha}{2}\nabla\tilde{F}(\tilde{\theta}_{k-1}))\|_2 d\pi_{k-1}^\star + 2R\exp\left(\frac{-4\varepsilon_2^2\gamma^2 m}{(\eta TLB_\ell)^2}\right).$$

The first inequality introduces the potentially suboptimal coupling $\pi_{k-1}^\star$. In the second inequality we first use that the projection operator is contractive, and we also use that the domain is contained in a Euclidean ball of radius $R$, providing a coarse upper bound on the second term. For the third inequality, we apply the concentration argument in Lemma 21. Working just with the first term, using the event in the indicator, we have

$$\int \mathbb{E}_{z_{1:m}}\mathbf{1}\{\mathcal{E}_k\}\|\hat{\theta}_{k-1} - \frac{\alpha}{2}\nabla\hat{F}(\hat{\theta}_{k-1}) - (\tilde{\theta}_{k-1} - \frac{\alpha}{2}\nabla\tilde{F}(\tilde{\theta}_{k-1}))\|_2 d\pi_{k-1}^\star$$

$$\leq \int \|\hat{\theta}_{k-1} - \frac{\alpha}{2}\nabla\hat{F}(\hat{\theta}_{k-1}) - (\tilde{\theta}_{k-1} - \frac{\alpha}{2}\nabla\hat{F}(\tilde{\theta}_{k-1}))\|_2 d\pi_{k-1}^\star + \frac{\alpha(\varepsilon_2 + \varepsilon')}{2}.$$

Now, observe that we are performing one step of gradient descent on $\hat{F}$ from two different starting points, $\hat{\theta}_{k-1}$ and $\tilde{\theta}_{k-1}$. Moreover, we know that $\hat{F}$ is smooth and strongly convex, which implies that the gradient descent update is *contractive*. Thus we will be able to upper bound the first term by $\mathcal{W}_1(\hat{\theta}_{k-1}, \tilde{\theta}_{k-1})$, which will lead to the result.

Here is the argument. Consider two arbitrary points $\theta, \theta' \in \Theta$. Let $G : \theta \to \theta - \alpha/2\nabla\hat{F}(\theta)$ be a vector valued function, and observe that the Jacobian is $I - \alpha/2\nabla^2\hat{F}(\theta)$. By the mean value theorem, there exists $\theta''$ such that

$$\|\theta - \frac{\alpha}{2}\nabla\hat{F}(\theta) - (\theta' - \frac{\alpha}{2}\nabla\hat{F}(\theta'))\|_2 \leq \|(I - \alpha/2\nabla^2\hat{F}(\theta''))(\theta - \theta')\|_2$$

$$\leq \|I - \alpha/2\nabla^2\hat{F}(\theta'')\|_\sigma\|\theta - \theta'\|_2.$$

Now, since $\hat{F}$ is $\lambda$-strongly convex and $\eta T B_\ell L/u + \lambda$ smooth, we know that all eigenvalues of $\nabla^2 \hat{F}(\theta'')$ are in the interval $[\lambda, \eta T B_\ell L/(u\gamma) + \lambda]$. Therefore, if $\alpha \le 2(\eta T B_\ell L/(u\gamma) + \lambda)^{-1} \le 1/\lambda$, the spectral norm term here is at most 1, implying that gradient descent is contractive. Thus, we get

$$\mathcal{W}_1(\hat{\theta}_k, \tilde{\theta}_k) \le \int \|\hat{\theta}_{k-1} - \tilde{\theta}_{k-1}\|_2 d\pi_{k-1}^\star + \frac{\alpha(\varepsilon_2 + \varepsilon')}{2} + 2R \exp\left(\frac{-4\varepsilon_2^2 \gamma^2 m}{(\eta TLB_\ell)^2}\right)$$

$$\le \mathcal{W}_1(\hat{\theta}_{k-1}, \tilde{\theta}_{k-1}) + \frac{\alpha}{2}\varepsilon_2 + \frac{\alpha}{\sqrt{m}} \cdot \frac{\eta T B_\ell L}{\gamma} + 2R \exp\left(\frac{-4\varepsilon_2^2 \gamma^2 m}{(\eta TLB_\ell)^2}\right).$$

The choice of $m$ ensures that the second and third term together are at most $\alpha\varepsilon_2$, from which the result follows. $\qquad\square$

**Fact 24.** *For any two distributions $\mu, \nu$ on $\Theta$, we have*
$$\mathcal{W}_1(\mu, \nu) \le R \cdot TV(\mu, \nu).$$

**Proof.** We use the coupling characterization of the total variation distance:
$$\mathcal{W}_1(\mu, \nu) = \inf_\pi \int \|\theta - \theta'\|_2 d\pi \le \text{diam}(\Theta) \inf_\pi \mathbb{P}_\pi[\theta \ne \theta'] \le R \cdot TV(\mu, \nu). \qquad\square$$

**Proof of Theorem 19.** By the triangle inequality and Fact 24 we have
$$\mathcal{W}_1(\tilde{\theta}_N, P) \le \mathcal{W}_1(\tilde{\theta}_N, \hat{\theta}_N) + R \cdot \left(TV(\hat{\theta}_N, \hat{P}) + TV(\hat{P}, P)\right).$$

The first term here is the Wasserstein distance between our true iterates $\tilde{\theta}_N$ and the idealized iterates from running LMC on $\hat{F}$, which is controlled by Lemma 23. The second is the total variation distance between the idealized iterates and the smoothed density $\hat{P}$, which is controlled in Theorem 22. Finally, the third term is the approximation error between the smoothed density $\hat{P}$ and the true, non-smooth one $P$. Together, for any choice of $\varepsilon > 0$ and $\varepsilon_2 > 0$ we obtain the bound

$$\mathcal{W}_1(\tilde{\theta}_N, P) \le N\alpha\varepsilon_2 + R\varepsilon + 2R(\eta T B_\ell L u \sqrt{d}/\gamma + \lambda R^2), \tag{12}$$

under the requirements

$$N \ge \frac{c_0 R^6 \max\{d, R\eta T B_\ell L/\gamma + R^2\lambda, R(\eta T B_\ell L/(u\gamma) + \lambda)\}^{12}}{\varepsilon^{12}}, \tag{13}$$

$$m \ge \frac{16(\eta TLB_\ell/\gamma)^2 \log(4R/\alpha\varepsilon_2)}{\varepsilon_2^2}.$$

There are also two requirements on $\alpha$, one arising from Theorem 22 and the other from Lemma 23. These are:
$$\alpha \le 2(\eta T B_\ell L/(u\gamma) + \lambda)^{-1}, \quad \text{and} \quad \alpha = c_1 R^2/N, \tag{14}$$

for any constant $c_1$.

Returning to the error bound, if we set
$$u = \frac{\tau}{8R\eta T B_\ell L\sqrt{d}}, \quad \text{and} \quad \lambda = \frac{\tau}{8R^3},$$

the last term in (12) is at most $\tau/2$.

We will make the choice $\alpha = c_1 R^2/N$. In this case, the values for $u$ and $\lambda$ above, combined with the inequality (14) give the constraint

$$N \ge 2c_1 R^2 \cdot \left(\frac{8(\eta TLB_\ell)^2 R\sqrt{d}}{\gamma\tau} + \frac{\tau}{8R^2}\right). \tag{15}$$

Now for the first term in (12), plug in the choice $\alpha = c_1 R^2/N$ and set $\varepsilon_2 = \tau/(4c_1 R^2)$ so that this term is at most $\tau/4$. For the second term, set $\varepsilon = \tau/(4R)$ so that this term is also at most $\tau/4$. With these choices, the requirements on $m$ and $N$ become:

$$m \ge \frac{64c_1^2 R^4(\eta T B_\ell/\gamma)^2 \log(\tau/(16RN))}{\tau^2}, \quad \text{and} \quad N \ge c_0' R^{18} \max\{d, (R\eta T B_\ell L/\gamma)^2 \sqrt{d}/\tau\}^{12}/\tau^{12},$$

where we have noted that the first constraint (13) clearly implies the second constraint (15), and this proves the theorem. $\qquad\square$

## D.2 Continuous exponential weights.

The focus of this section of the appendix is Lemma 25, which analyzes a continuous version of the Hedge/exponential weights algorithms in the full information setting. This lemma appears in various forms in several places, e.g. [13]. For the setup, consider an online learning problem with a parametric benchmark class $\mathcal{F} = \{f(\cdot\,;\theta) \mid \theta \in \Theta\}$ where $f(\cdot\,;\theta) \in (\mathcal{X} \to \mathbb{R}^K_{=0})$ and further assume that $\Theta \in \mathbb{R}^d$ contains the centered Euclidean ball of radius $r = 1$ and is contained in the Euclidean ball of radius $R$. Finally, assume that $f(x;\cdot)_a$ is $L$-Lipschitz with respect to $\ell_2$ norm in $\theta$ for all $x \in \mathcal{X}$. On each round $t$ an adversary chooses a context $x_t \in \mathcal{X}$ and a loss vector $\ell_t \in \mathbb{R}^K_+$, the learner then choose a distribution $p_t \in \Delta(\mathcal{F})$ and suffers loss:

$$\mathbb{E}_{f \sim p_t} \langle \ell_t, \psi^\gamma(f(x_t)) \rangle.$$

The entire loss vector $\ell_t$ is then revealed to the learner. Here, performance is measured via regret:

$$\mathrm{Regret}(T, \mathcal{F}) \triangleq \sum_{t=1}^T \mathbb{E}_{f \sim p_t} \langle \ell_t, \psi^\gamma(f(x_t)) \rangle - \inf_{f \in \mathcal{F}} \sum_{t=1}^T \langle \ell_t, \psi^\gamma(f(x_t)) \rangle.$$

Our algorithm is a continuous version of exponential weights. Starting with $w_0(f) \triangleq 0$, we perform the updates:

$$p_t(f) = \frac{\exp(-\eta w_t(f))}{\int_{\mathcal{F}} \exp(-\eta w_t(f)) d\lambda(f)}, \quad \text{and} \quad w_{t+1}(f) = w_t(f) + \langle \ell_t, \psi^\gamma(f(x_t)) \rangle.$$

Here $\eta$ is the learning rate and $\lambda$ is the Lebesgue measure on $\mathcal{F}$ (identifying elements $f \in \mathcal{F}$ with their representatives $\theta \in \mathbb{R}^d$).

With these definitions, the continuous Hedge algorithm enjoys the following guarantee.

**Lemma 25.** *Assume that the losses $\ell_t$ satisfy $\|\ell_t\|_\infty \le B_\ell$, $\Theta \subset \mathbb{R}^d$ is contained within the Euclidean ball of radius $R$, and $f(x;\cdot)_a$ is $L$-Lipschitz continuous in the third argument with respect to $\ell_2$. Let the margin parameter $\gamma$ be fixed. Then the continuous Hedge algorithm with learning rate $\eta > 0$ enjoys the following regret guarantee:*

$$\mathrm{Regret}(T, \mathcal{F}) \le \inf_{\varepsilon > 0} \left\{ \frac{TKB_\ell\varepsilon}{\gamma} + \frac{d}{\eta} \log(RL/\varepsilon) + \frac{\eta}{2} \sum_{t=1}^T \mathbb{E}_{f \sim p_t} \langle \ell_t, \psi^\gamma(f(x_t)) \rangle^2 \right\}.$$

**Proof.** Following the standard analysis for continuous Hedge (e.g. Lemma 10 in [32]), we know that the regret to some benchmark distribution $Q \in \Delta(\mathcal{F})$ is

$$\sum_{t=1}^T (\mathbb{E}_{f \sim p_t} - \mathbb{E}_{f \sim Q})(\langle \ell_t, \psi^\gamma(f(x_t)) \rangle) = \frac{\mathrm{KL}(Q \,\|\, p_0) - \mathrm{KL}(Q \,\|\, p_T)}{\eta} + \frac{1}{\eta} \sum_{t=1}^T \mathrm{KL}(p_{t-1} \,\|\, p_t).$$

For the KL terms, using the standard variational representation, we have

$$\mathrm{KL}(p_{t-1} \,\|\, p_t) = \log \mathbb{E}_{f \sim p_{t-1}} \exp\left( -\eta \Big\langle \ell_t, \psi^\gamma(f(x_t)) - \mathbb{E}_{f \sim p_{t-1}} \psi^\gamma(f(x_t)) \Big\rangle \right)$$

$$\le \log\left( 1 + \frac{\eta^2}{2} \mathbb{E}_{f \sim p_{t-1}} \Big\langle \ell_t, \psi^\gamma(f(x_t)) - \mathbb{E}_{f \sim p_{t-1}} \psi^\gamma(f(x_t)) \Big\rangle^2 \right)$$

$$\le \frac{\eta^2}{2} \mathbb{E}_{f \sim p_{t-1}} \langle \ell_t, \psi^\gamma(f(x_t)) \rangle^2.$$

Here the first inequality is $e^{-x} \le 1 - x + x^2/2$, using that the term inside the exponential is centered. The second inequality is $\log(1 + x) \le x$.

Using non-negativity of KL, we only have to worry about the $\mathrm{KL}(Q \,\|\, p_0)$ term. Let $f^\star$ be the minimizer of the cumulative hinge loss. Let $\theta^\star \in \Theta$ be a representative for $f^\star$ and let $Q$ be the uniform distribution on $\mathcal{F}_\varepsilon(\theta^\star, x_{1:T}) \triangleq \{\theta : \max_{t \in [T]} \|f(x_t;\theta) - f(x_t;\theta^\star)\|_\infty \le \varepsilon\}$, then we have

$$\mathrm{KL}(Q \,\|\, p_0) = \int_f q(f) \log(q(f)/p_0(f)) d\lambda(f) = \int dQ(f) \cdot \log \frac{\int_{\mathcal{F}} d\lambda(f)}{\int_{\mathcal{F}_\varepsilon} d\lambda(f)} = \log \frac{\mathrm{Vol}(\mathcal{F})}{\mathrm{Vol}(\mathcal{F}_\varepsilon(\theta^\star, x_{1:T}))},$$

where $\mathrm{Vol}(S)$ denotes the Lebesgue integral. We know that $\mathrm{Vol}(\Theta) \le c_d R^d$ where $c_d$ is the Lebesgue volume of the unit Euclidean ball and $R$ is the radius of the ball containing $\Theta$, and so we must lower bound the volume of $\mathcal{F}_\varepsilon(f^\star, x_{1:T})$. For this step, observe that by the Lipschitz-property of $f$,

$$\sup_{x \in \mathcal{X}} \|f(x;\theta) - f(x;\theta^\star)\|_\infty \le L\|\theta - \theta^\star\|_2,$$

and hence $\mathcal{F}_\varepsilon(\theta^\star, x_{1:T}) \supset B_2(\theta^\star, \varepsilon/L)$. Thus the volume ratio is

$$\frac{\mathrm{Vol}(\mathcal{F})}{\mathrm{Vol}(\mathcal{F}_\varepsilon(\theta^\star, x_{1:T}))} \le \frac{c_d R^d}{c_d(\varepsilon/L)^d} = (RL/\varepsilon)^d.$$

Finally, using the fact that the hinge surrogate is $1/\gamma$-Lipschitz, we know that

$$\sum_{t=1}^T \mathbb{E}_{f \sim Q} \langle \ell_T, \psi^\gamma(f(x_t)) - \psi^\gamma(f^\star(x_t)) \rangle \le TB_\ell \sup_{t \in [T], f \in \mathrm{supp}(Q)} \|\psi^\gamma(f(x_t)) - \psi^\gamma(f^\star(x_t))\|_1$$
$$\le \frac{TKB_\ell \varepsilon}{\gamma}. \qquad \square$$

### D.3  From full information to bandits.

We now combine the results of Subsection D.1 and Subsection D.2 to give the final guarantee for HINGE-LMC.

We begin by translating the regret bound in Lemma 25, followed by many steps of approximation. At round $t$, let $P_t$ denote the Hedge distribution on $\Theta$ using the losses $\tilde{\ell}_{1:t-1}$. Let $\tilde{P}_t$ denote the distribution from which $\theta_t \in \Theta$ is sampled in Algorithm 3.

Let $p_t \in \Delta(\mathcal{A})$ denote the induced distributions on actions induced by $P_t$, i.e. the distribution induced by the process $\theta \sim P_t$, $p_t(a) \propto \psi^\gamma(f(x_t;\theta))$. Likewise, let $\sim p_t \in \Delta(\mathcal{A})$ be the distribution induced by $\theta \sim \tilde{P}_t$, $\tilde{p}_t(a) \propto \psi^\gamma(f(x_t;\theta))$); in this notation $\tilde{p}_t^\mu$ is precisely the distribution from which actions are sampled in Algorithm 1.

Recall that we use $\mu$ in the superscript to denote smoothing (e.g. $p_t^\mu$). Let $m_t$ denote the random variable sampled at round $t$ to approximate the importance weight.

We also let $\hat{\ell}_t(a) = \frac{\ell_t(a)}{\tilde{p}_t^\mu(a)} \mathbf{1}\{a_t = a\}$ denote estimated losses under the true importance weights, which are not explicitly used by Algorithm 1 but are used in the analysis.

Let $\mathbf{1}_a \in \mathbb{R}^K$ be the vector with 1 at coordinate $a$ and 0 at all other coordinates.

**Proof of Theorem 18.** The thrust of this proof is to show that the full information bound in Lemma 25 does not degrade significantly under importance weighting and under the approximate LMC implementation of continuous exponential weights.

**Variance control**  Controlling the variance term in Lemma 25 requires an application of Lemma 3. After taking conditional expectations, the variance term is

$$\sum_{t=1}^T \mathbb{E}_{\theta \sim P_t} \mathbb{E}_{a_t \sim \tilde{p}_t^\mu} \mathbb{E}_{m_t} \langle \tilde{\ell}_t, \psi^\gamma(f(x_t;\theta)) \rangle^2 = \sum_{t=1}^T \mathbb{E}_{s \sim P_t} \mathbb{E}_{a_t \sim \tilde{p}_t^\mu} \mathbb{E}_{m_t} m_t^2 \langle \ell_t(a_t) \mathbf{1}_{a_t}, s \rangle^2.$$

Here we are identifying $s$ with $\psi^\gamma(f(x_t;\theta))$ and marginalizing out $\theta$ in the outermost expectation. Note that this is the same definition of $s$ as in Lemma 3.

First let us handle the $m_t$ random variable. Note that conditional on everything up to round $t$ and $a_t$, $m_t$ is distributed according to a geometric distribution with mean $\tilde{p}_t^\mu(a_t)$, truncated at $M$. It is straightforward (cf. [33]) to show that $m_t$ is stochastically dominated by a geometric random variable with mean $\frac{1}{\tilde{p}_t^\mu(a_t)}$ and hence the second moment of this random variable is at most $\frac{2}{\tilde{p}_t^\mu(a_t)^2}$. Thus, we

are left with

$$\le 2 \sum_{t=1}^{T} \mathbb{E}_{s \sim P_t} \mathbb{E}_{a_t \sim \tilde{p}_t^{\mu}} \frac{1}{\tilde{p}_t^{\mu}(a_t)^2} \langle \ell_t(a_t) \mathbf{1}_{a_t}, s \rangle^2$$

$$= 2 \sum_{t=1}^{T} \mathbb{E}_{s \sim P_t} \mathbb{E}_{a_t \sim \tilde{p}_t^{\mu}} \langle \hat{\ell}_t, s \rangle^2$$

$$\le 2 \sum_{t=1}^{T} (\mathbb{E}_{s \sim \tilde{P}_t} - \mathbb{E}_{s \sim P_t}) \mathbb{E}_{a_t \sim \tilde{p}_t^{\mu}} \langle \hat{\ell}_t, s \rangle^2 + \mathbb{E}_{s \sim \tilde{P}_t} \mathbb{E}_{a_t \sim \tilde{p}_t^{\mu}} \langle \hat{\ell}_t, s \rangle^2.$$

We can apply Lemma 3 on the second term, since the only condition for the lemma is that the action distribution is induced from the distribution in the outer expectation. It follows that this term is bounded as

$$\sum_{t=1}^{T} \mathbb{E}_{s \sim \tilde{P}_t} \mathbb{E}_{a_t \sim \tilde{p}_t^{\mu}} \langle \hat{\ell}_t, s \rangle^2 \le T K^2 (1 + B/\gamma)^2.$$

For the first term, evaluating the inner expectation, using the fact that $\tilde{p}_t^{\mu}(a) \ge \mu$ and applying the Lipschitz properties of $\psi^{\gamma}(\cdot)$, $f(x; \cdot)$ (in particular that $f(x; \cdot)$ is $L$-Lipschitz with respect to $\ell_2$ and that the Wasserstein distance we work with is defined relative to $\ell_2$) we have

$$(\mathbb{E}_{s \sim \tilde{P}_t} - \mathbb{E}_{s \sim P_t}) \mathbb{E}_{a_t \sim \tilde{p}_t^{\mu}} \langle \hat{\ell}_t, s \rangle^2 = \sum_a (\mathbb{E}_{\theta \sim \tilde{P}_t} - \mathbb{E}_{\theta \sim P_t}) \frac{\ell_t^2(a)}{\tilde{p}_t^{\mu}(a)} \psi^{\gamma}(f(x_t; \theta)_a)^2$$

$$\le 2 \frac{(1 + B/\gamma) K L}{\gamma \mu} \sup_{g, \|g\|_{\mathrm{Lip}} \le 1} \left| \int g(dP_t - d\tilde{P}_t) \right| = 2 \frac{(1 + B/\gamma) K L}{\gamma \mu} \mathcal{W}_1(P_t, \tilde{P}_t).$$

Finally, using the Wasserstein guarantee $\mathcal{W}_1(P_t, \tilde{P}_t) \le \tau$ from Theorem 19, we conclude that the cumulative variance term is upper bounded as

$$\sum_{t=1}^{T} \mathbb{E} \langle \tilde{\ell}_t, \psi^{\gamma}(f(x_t; \theta)) \rangle^2 \le \frac{4(1 + B/\gamma) K T L \tau}{\gamma \mu} + 2(1 + B/\gamma)^2 K^2 T.$$

**Bounding regret**   We first relate the cumulative loss under Algorithm 1 to the cumulative loss of continuous exponential weights. Observe that

$$\sum_{t=1}^{T} \langle \ell_t, \tilde{p}_t^{\mu} \rangle \le \mu K T + \sum_{t=1}^{T} \langle \ell_t, \tilde{p}_t \rangle$$

$$\le \mu K T + \frac{1}{K} \sum_{t=1}^{T} \mathbb{E}_{\theta \sim \tilde{P}_t} \langle \ell_t, \psi^{\gamma}(f(x_t; \theta)) \rangle$$

$$\le \mu K T + \frac{T L \tau}{\gamma} + \frac{1}{K} \sum_{t=1}^{T} \mathbb{E}_{\theta \sim P_t} \langle \ell_t, \psi^{\gamma}(f(x_t; \theta)) \rangle.$$

This first inequality is a straightforward consequence of smoothing, while the second is a direct application of Lemma 1.

The third inequality is based on the fact that $\langle \ell_t, \psi^{\gamma}(f(x_t; \theta)) \rangle$ is $K L/\gamma$-Lipschitz in $\theta$ with respect to $\ell_2$ norm under our assumptions. This step also uses the Wasserstein guarantee in Theorem 19 which produces the approximation factor $\tau$.

Following the analysis in [33] and using the boundedness of $\psi^{\gamma}$, the bias introduced due to using geometric resampling with truncation at $M$ instead of exact inverse propensity scores is

$$\sum_{t=1}^{T} \mathbb{E}_{\theta \sim P_t} \langle \ell_t, \psi^{\gamma}(f(x_t; \theta)) \rangle \le \mathbb{E}_{a_{1:T}, m_{1:T}} \sum_{t=1}^{T} \mathbb{E}_{\theta \sim P_t} \langle \tilde{\ell}_t, \psi^{\gamma}(f(x_t; \theta)) \rangle + \frac{T(1 + B/\gamma)}{eM}.$$

For the remaining term, we apply Lemma 25 with $\varepsilon = \gamma/(TKM)$, since $M$ is an upper bound on the norm $\|\tilde{\ell}_t\|_1$ of the losses to the full information algorithm.

$$\mathbb{E}_{a_{1:T}, m_{1:T}} \sum_{t=1}^{T} \mathbb{E}_{\theta \sim P_t} \langle \tilde{\ell}_t, \psi^{\gamma}(f(x_t; \theta)) \rangle$$

$$\le \mathbb{E} \inf_{\theta \in \Theta} \sum_{t=1}^{T} \langle \tilde{\ell}_t, \psi^{\gamma}(f(x_t; \theta)) \rangle + 1 + \frac{d}{\eta} \log(RLTKM/\gamma) + \eta \left( \frac{(1 + B/\gamma) K T L \tau}{\gamma \mu} + (1 + B/\gamma)^2 K^2 T \right).$$

The first term here is the benchmark we want to compare to, since $\mathbb{E}\inf(\cdot) \le \inf\mathbb{E}[\cdot]$ and so the regret contains several terms:

$$\mu KT + \frac{TL\tau}{\gamma} + \frac{T(1+B/\gamma)}{eMK} + \frac{1}{K} + \frac{d}{K\eta}\log(RLTKM/\gamma) + \frac{\eta}{2K}\left(\frac{2(1+B/\gamma)KTL\tau}{\gamma\mu} + 4(1+B/\gamma)^2 K^2 T\right)$$

$$\le \mu KT + \frac{TL\tau}{\gamma} + \frac{T(1+B/\gamma)}{eMK} + \frac{1}{K} + \frac{d}{K\eta}\log(RLTKM/\gamma) + \frac{2\eta}{K\gamma^2}\left(\frac{BKTL\tau}{\mu} + 4B^2 K^2 T\right).$$

Here we use the assumption $B/\gamma \ge 1$. We will simplify the expression to obtain an $\tilde{\mathcal{O}}(\sqrt{dKT})$-type bound, first set $\mu = 1/(K\sqrt{T})$, $M = \sqrt{T}$ and $\tau = \sqrt{1/(TL^2)}$. This gives

$$2\sqrt{T} + \frac{2B}{\gamma}\sqrt{T} + \frac{2d}{K\eta}\log(RLTK/\gamma) + \frac{2\eta}{K\gamma^2}\left(BK^2 T + 4B^2 K^2 T\right)$$

$$\le O(B\sqrt{T}/\gamma) + \frac{2d}{K\eta}\log(RLTK/\gamma) + \frac{10\eta}{\gamma^2}B^2 KT.$$

Finally set $\eta = \sqrt{\frac{d\gamma^2\log(RLTK/\gamma)}{5K^2 B^2 T}}$ to get

$$O(\sqrt{T}/\gamma) + O\left(\frac{B}{\gamma}\sqrt{dT\log(RLTK/\gamma)}\right) = \tilde{\mathcal{O}}(\frac{B}{\gamma}\sqrt{dT}).$$

This concludes the proof of the regret bound.

**Running time calculation.** At each round make $M+1$ calls to the LMC sampling routine for a total of $O(T^{3/2})$ calls across all rounds. We now bound the running time for a single call.

We always use parameter $\tau = \sqrt{1/(TL^2)}$ and we know $\|\tilde{\ell}\|_1 \le 1/\mu = K\sqrt{T}$ and $\eta = \tilde{\mathcal{O}}(\sqrt{\frac{d}{K^2 T}})$. Plugging into the parameter choices at the end of the proof of Theorem 19, we must sample

$$m = \tilde{\mathcal{O}}(T^3 dR^4 L^2 B_\ell^2/(K\gamma)^2)$$

samples from a gaussian distribution on each iteration, and the number of iterations to generate a single sample is:

$$N = \tilde{\mathcal{O}}\left(R^{18}L^{12}T^6 d^{12} + \frac{R^{24}L^{48}d^{12}}{K^{24}}\right).$$

Therefore, the total running time across all rounds is

$$\tilde{\mathcal{O}}\left(\frac{R^{22}L^{14}d^{14}B_\ell^2 T^{10}}{K^2\gamma^2} + \frac{R^{28}L^{50}d^{14}B_\ell^2 T^4}{K^{26}\gamma^2}\right).$$

$\square$

### D.4 Proofs for corollaries

Corollary 7 is an immediate consequence of Theorem 6. For Corollary 8, we apply Lemma 10, since $\theta^\star \in \Theta$ satisfies the conditions of the lemma pointwise. Thus

$$K^{-1}\mathbb{E}\langle\ell_t, \psi^\gamma(f(x_t; \theta^\star))\rangle = K^{-1}\mathbb{E}[\langle\bar{\ell}_t, \psi^\gamma(f(x_t; \theta^\star))\rangle \mid x_t] = \mathbb{E}[\min_a \bar{\ell}_t(a)|x_t].$$

Therefore, letting $a_t^\star$ denote the optimal action minimizing $\bar{\ell}_t$, we obtain the expected regret bound

$$\sum_{t=1}^{T}\mathbb{E}[\langle\bar{\ell}_t, a_t - a_t^\star\rangle] \le \tilde{\mathcal{O}}((B/\gamma)\sqrt{dT}).$$

## E Analysis of SMOOTHFTL

Recall we are in the stochastic setting. Let $\mathcal{D}$ denote the distribution over $(\mathcal{X}, \mathbb{R}_+^K)$.

The bulk of the analysis is the following uniform convergence lemma, which is based on chaining for the function class $\mathcal{F}$. Recall that $\mathcal{N}_{\infty,\infty}(\varepsilon, \mathcal{F})$ is the $L_\infty/\ell_\infty$ covering number from Definition 1.

**Lemma 26.** *Fix a predictor $\hat{f}$ and let $\{x_i, a_i, \ell_i(a_i)\}_{i=1}^n$ be a dataset of $n$ samples, Suppose that $(x_i, \ell_i)$ are drawn i.i.d. from some distribution $\mathcal{D}$ and $a_i$ is sampled from $p_i \triangleq (1 - K\mu)\pi_{hinge}\hat{f}(x_i) + \mu$. Define $\hat{R}_n^\psi(f) \triangleq \frac{1}{n}\sum_{i=1}^n \langle \hat{\ell}_i, \psi^\gamma(f(x_i))\rangle$, where $\hat{\ell}_i$ is the importance-weighted loss. Then:*

$$\mathbb{E}\sup_{f\in\mathcal{F}}|R^\psi(f) - \hat{R}_n^\psi(f)\rangle|$$

$$\leq \frac{1}{\gamma}\inf_{\beta\geq 0}\left\{2K\beta + 12\int_\beta^2\left(\sqrt{\frac{2K}{n\mu}\log(n\mathcal{N}_{\infty,\infty}(\varepsilon,\mathcal{F},n))} + \frac{3\log(n\mathcal{N}_{\infty,\infty}(\varepsilon,\mathcal{F},n))}{n\mu}\right)d\varepsilon\right\}.$$

**Proof of Lemma 26.** Note that since the data-collection policy $\hat{f}$ is fixed, and since we are in the stochastic setting with $(x_i, \ell_i) \sim \mathcal{D}$, the samples $\{x_i, a_i, \ell_i(a_i)\}_{i=1}^n$ are i.i.d. Consequently, we can apply the standard symmetrization upper bound for uniform convergence. Beginning with

$$\mathbb{E}_{x_{1:n}, a_{1:n}, \ell_{1:n}}\sup_{f\in\mathcal{F}}\left[R^\psi(f) - \hat{R}_n^\psi(f)\right],$$

we introduce a second "ghost" dataset of samples $\tau = n+1, \ldots, 2n$ via Jensen's inequality.

$$\leq \mathbb{E}_{x_{1:2n}, a_{1:2n}, \ell_{1:2n}}\sup_{f\in\mathcal{F}}\frac{1}{n}\sum_{\tau=n+1}^{2n}\langle \hat{\ell}_\tau, \psi^\gamma(f(x_\tau))\rangle - \frac{1}{n}\sum_{\tau=1}^n\langle \hat{\ell}_\tau, \psi^\gamma(f(x_\tau))\rangle.$$

Introducing Rademacher random variables and splitting the supremum:

$$\leq 2\mathbb{E}_{x_{1:n}, a_{1:n}, \ell_{1:n}, \epsilon_{1:n}}\sup_{f\in\mathcal{F}}\frac{1}{n}\sum_{\tau=1}^n\epsilon_\tau\langle \hat{\ell}_\tau, \psi^\gamma(f(x_\tau))\rangle.$$

Now condition on $x_{1:n}$ and define a sequence $\beta_i = 2^{1-i}$ for $i \in \{0, 1, 2, \ldots, N\}$, where $N$ is such that $\beta_{N+1} \geq \beta \geq \beta_{N+2}$ for the value of $\beta$ in the lemma statement. For each $\beta_i$ let $V_i$ be a (classical) $L_\infty/\ell_\infty$ cover for $f$ at scale $\beta_i$ on $x_{1:n}$, that is

$$\forall f \in \mathcal{F}, \forall i, \exists v \in V_i \text{ s.t. } \max_{t\in[n]}\|f(x_t) - v_t\|_\infty \leq \beta_i.$$

We can always ensure $|V_i| \leq \mathcal{N}_{\infty,\infty}(\beta_i, \mathcal{F}, n)$ and since $\|f(x)\|_\infty \leq 1$, we know that $\mathcal{N}_{\infty,\infty}(\beta_0, \mathcal{F}, n) \leq 1$. Now, letting $v^{(i)}(f)$ denote the covering element for $f$ at scale $\beta_i$, we have

$$\mathbb{E}_{a_{1:n}, \ell_{1:n}, \epsilon_{1:n}}\sup_{f\in\mathcal{F}}\frac{1}{n}\sum_{\tau=1}^n\epsilon_\tau\langle \hat{\ell}_\tau, \psi^\gamma(f(x_\tau))\rangle$$

$$\leq \mathbb{E}_{a_{1:n}, \ell_{1:n}, \epsilon_{1:n}}\sup_{f\in\mathcal{F}}\frac{1}{n}\sum_{\tau=1}^n\epsilon_\tau\langle \hat{\ell}_\tau, \psi^\gamma(f(x_\tau)) - \psi^\gamma(v_\tau^{(N)}(f))\rangle$$

$$+ \sum_{i=1}^N\sup_{f\in\mathcal{F}}\frac{1}{n}\sum_{\tau=1}^n\epsilon_\tau\langle \hat{\ell}_\tau, \psi^\gamma(v_\tau^{(i)}(f)) - \psi^\gamma(v_\tau^{(i-1)}(f))\rangle$$

$$+ \sup_{f\in\mathcal{F}}\frac{1}{n}\sum_{\tau=1}^n\epsilon_\tau\langle \hat{\ell}_\tau, \psi^\gamma(v_\tau^{(0)}(f))\rangle.$$

Since $|V_0| \leq 1$, the expected value of the third term is zero. The remaining work is to bound the first and second terms.

For the first term note that by Hölder's inequality, for any $f \in \mathcal{F}$,

$$\frac{1}{n}\sum_{\tau=1}^n\epsilon_\tau\langle \hat{\ell}_\tau, \psi^\gamma(f(x_\tau)) - \psi^\gamma(v_\tau^{(N)}(f))\rangle \leq \frac{1}{n}\sum_{\tau=1}^n\|\hat{\ell}_\tau\|_1\|\psi^\gamma(f(x_\tau)) - \psi^\gamma(v_\tau^{(N)}(f))\|_\infty$$

$$\leq \frac{\beta_N}{\gamma}\frac{1}{n}\sum_{\tau=1}^n\|\hat{\ell}_\tau\|_1,$$

since $\psi^\gamma$ is $1/\gamma$-Lipschitz. Thus for the first term, we have

$$\mathbb{E}_{a_{1:n}, \ell_{1:n}, \epsilon_{1:n}}\sup_{f\in\mathcal{F}}\frac{1}{n}\sum_{\tau=1}^n\epsilon_\tau\langle \hat{\ell}_\tau, \psi^\gamma(f(x_\tau)) - \psi^\gamma(v_\tau^{(N)}(x_\tau))\rangle \leq \frac{\beta_N}{\gamma}\mathbb{E}_{a_{1:n}, \ell_{1:n}}\frac{1}{n}\sum_{\tau=1}^n\|\hat{\ell}_\tau\|_1 \leq \frac{\beta_N K}{\gamma}.$$

Note that there is no dependence on the smoothing parameter $\mu$ here.

For the second term, let us denote the $i$th term in the summation by

$$\underbrace{\mathbb{E}_{a_{1:n},\ell_{1:n},\epsilon_{1:n}}\sup_{f\in\mathcal{F}}\frac{1}{n}\sum_{\tau=1}^{n}\epsilon_{\tau}\langle\hat{\ell}_{\tau},\psi^{\gamma}(v_{\tau}^{(i)}(f))-\psi^{\gamma}(v_{\tau}^{(i-1)}(f))\rangle}_{\triangleq\,\mathcal{E}_i}.$$

We control $\mathcal{E}_i$ using Bernstein's inequality and a union bound. First, note that the individual elements in the sum satisfy the deterministic bound

$$|\epsilon_{\tau}\langle\hat{\ell}_{\tau},\psi^{\gamma}(v_{\tau}^{(i)}(f))-\psi^{\gamma}(v_{\tau}^{(i-1)}(f))\rangle|\le\frac{3\beta_i}{\mu\gamma}, \tag{16}$$

and the variance bound,

$$\mathbb{E}\langle\hat{\ell}_{\tau},\psi^{\gamma}(v_{\tau}^{(i)}(f))-\psi^{\gamma}(v_{\tau}^{(i-1)}(f))\rangle^2\le\sum_a\mathbb{E}_{a_{\tau}}\frac{\mathbf{1}\{a_{\tau}=a\}}{p_{\tau}(a)^2}(\psi^{\gamma}(v_{\tau}^{(i)}(f)_a)-\psi^{\gamma}(v_{\tau}^{(i-1)}(f)_a))^2$$

$$\le\sum_a\frac{1}{\mu}(3\beta_i/\gamma)^2=\frac{9\beta_i^2 K}{\mu\gamma^2}. \tag{17}$$

Here we are using that $v^{(i)}(f)$ and $v^{(i-1)}(f)$ are the covering elements for $f$, Lipschitzness of $\psi^{\gamma}$, and the definition of the importance weighted loss $\hat{\ell}_{\tau}$.

Using (16) and (17), Bernstein's inequality (e.g. [11], Theorem 2.9) implies that for any $\delta\in(0,1)$,

$$\frac{1}{n}\sum_{\tau=1}^{n}\epsilon_{\tau}\langle\hat{\ell}_{\tau},\psi^{\gamma}(v_{\tau}^{(i)}(f))-\psi^{\gamma}(v_{\tau}^{(i-1)}(f))\rangle\le 6\sqrt{\frac{\beta_i^2 K}{n\mu\gamma^2}\log(1/\delta)}+\frac{6\beta_i}{n\mu\gamma}\log(1/\delta),$$

with probability at least $1-\delta$. The important point here is that $1/(n\mu)$ appears in the square root, as opposed to $1/(n\mu^2)$. Via a union bound, for any $\delta\in(0,1)$, with probability at least $1-\delta$,

$$\sup_f\frac{1}{n}\sum_{\tau=1}^{n}\epsilon_{\tau}\langle\hat{\ell}_{\tau},\psi^{\gamma}(v_{\tau}^{(i)}(f))-\psi^{\gamma}(v_{\tau}^{(i-1)}(f))\rangle$$

$$\le 6\sqrt{\frac{\beta_i^2 K}{n\mu\gamma^2}\log(|V_i||V_{i-1}|/\delta)}+\frac{6\beta_i}{n\mu\gamma}\log(|V_i||V_{i-1}|/\delta)$$

$$\le\frac{6\beta_i}{\gamma}\left(\sqrt{\frac{2K}{n\mu}\log(|V_i|/\delta)}+\frac{2\log(|V_i|/\delta)}{n\mu}\right),$$

since $|V_{i-1}|\le|V_i|$. Now, recalling the shorthand definition $\mathcal{E}_i$

$$\mathbb{E}_{a_{1:n},\ell_{1:n},\epsilon_{1:n}}\mathcal{E}_i\le\inf_{\zeta}\mathbb{E}\mathbf{1}\{\mathcal{E}_i\le\zeta\}\cdot\zeta+\mathbb{E}\mathbf{1}\{\mathcal{E}_i>\zeta\}\cdot\frac{3\beta_i}{\mu\gamma}$$

$$\le\inf_{\delta\in(0,1)}\frac{6\beta_i}{\gamma}\left(\sqrt{\frac{2K}{n\mu}\log(|V_i|/\delta)}+\frac{2\log(|V_i|/\delta)}{n\mu}\right)+\frac{3\beta_i\delta}{\mu\gamma}.$$

Choosing $\delta=1/n$:

$$\le\frac{6\beta_i}{\gamma}\left(\sqrt{\frac{2K}{n\mu}\log(n|V_i|)}+\frac{3\log(n|V_i|)}{n\mu}\right).$$

Thus, the second term in the chaining decomposition is

$$\frac{6}{\gamma}\sum_{i=1}^{N}\beta_i\left(\sqrt{\frac{2K}{n\mu}\log(n|V_i|)}+\frac{3\log(n|V_i|)}{n\mu}\right)$$

$$=\frac{12}{\gamma}\sum_{i=1}^{N}(\beta_i-\beta_{i+1})\left(\sqrt{\frac{2K}{n\mu}\log(n|V_i|)}+\frac{3\log(n|V_i|)}{n\mu}\right)$$

$$\le\frac{12}{\gamma}\int_{\beta_{N+1}}^{\beta_0}\left(\sqrt{\frac{2K}{n\mu}\log(n\mathcal{N}_{\infty,\infty}(\beta,\mathcal{F}))}+\frac{3\log(n\mathcal{N}_{\infty,\infty}(\beta,\mathcal{F}))}{n\mu}\right)d\beta.$$

This concludes the uniform deviation statement. Exactly the same argument applies to the other tail, so the bound holds on the absolute value. $\qquad\square$

**Proof of Theorem 9.** Let us denote the right hand side of Lemma 26, when the dataset is size $n$, as $\Delta_n$. Define,

$$f^\star = \operatorname*{argmin}_{f \in \mathcal{F}} \mathbb{E}\langle \ell, \psi^\gamma(f(x))\rangle,$$

Since the $m^{\text{th}}$ epoch proceeds for $n_m \triangleq 2^m$ rounds, and the predictor that we use in the $m^{\text{th}}$ epoch is the ERM on all of the data from the $(m-1)^{\text{st}}$ epoch, the expected cumulative hinge regret for the $m^{\text{th}}$ epoch is

$$2^m \cdot \left(\mathbb{E}R^\psi(\hat{f}_{m-1}) - R^\psi(f^\star)\right).$$

Using the optimality guarantee for ERM:

$$\leq 2^m \cdot \left(\mathbb{E}R^\psi(\hat{f}_{m-1}) - \frac{1}{n_{m-1}}\sum_{\tau=n_{m-1}}^{n_m-1}\langle\hat{\ell}_\tau, \psi^\gamma(\hat{f}_{m-1}(x_\tau))\rangle + \frac{1}{n_{m-1}}\sum_{\tau=n_{m-1}}^{n_m-1}\langle\hat{\ell}_\tau, \psi^\gamma(f^\star(x_\tau))\rangle - R^\psi(f^\star)\right)$$

$$\leq 2^{m+1}\mathbb{E}\sup_f\left|R^\psi(f) - \hat{R}^\psi_{n_{m-1}}(f)\right|.$$

Using the guarantee from Lemma 26:

$$\leq 2^{m+1}\Delta_{n_{m-1}}. \tag{18}$$

Summing this bound over all rounds, the cumulative expected regret after the zero-th epoch is $\sum_{m=1}^{\log_2(T)} 2^{m+1}\Delta_{n_{m-1}}$. The zero-th epoch contributes $1/\gamma$ to the regret, which will be lower order. This gives the following upper bound on the cumulative expected hinge loss regret.

$$\text{Regret}(T, \mathcal{F}) \leq \sum_{m=1}^{\log_2(T)} 2^{m+1}\Delta_{n_{m-1}}$$

$$\leq \frac{4}{\gamma}\sum_{m=1}^{\log_2(T)}\inf_{\beta>0}\left\{n_m K\beta + 12\cdot 2^{m-1}\cdot\int_\beta^{2B}\left(\sqrt{\frac{2K}{n_{m-1}\mu}\log(n_{m-1}\mathcal{N}_{\infty,\infty}(\varepsilon,\mathcal{F}))} + \frac{3\log(n_{m-1}\mathcal{N}_{\infty,\infty}(\varepsilon,\mathcal{F}))}{n_{m-1}\mu}\right)d\varepsilon\right\}$$

$$\leq \frac{4}{\gamma}\inf_{\beta>0}\left\{KT\beta + 12\log_2(T)\cdot\int_\beta^{2B}\left(\sqrt{\frac{2KT}{\mu}\log(T\mathcal{N}_{\infty,\infty}(\varepsilon,\mathcal{F}))} + \frac{3\log(T\mathcal{N}_{\infty,\infty}(\varepsilon,\mathcal{F}))}{\mu}\right)d\varepsilon\right\}.$$

$$\triangleq C.$$

Let $z_t = \hat{f}_{m-1}(x_t)$ for each time $t$ in epoch $m$. We have just shown

$$\sum_{t=1}^T \mathbb{E}\langle\ell_t, \psi^\gamma(z_t)\rangle \leq T\cdot\mathbb{E}\langle\ell, \psi^\gamma(f^\star(x))\rangle + C.$$

Using Lemma 1, this implies

$$\sum_{t=1}^T \mathbb{E}\big\langle\ell_t, \pi_{\text{hinge}}(z_t)\big\rangle \leq \frac{T}{K}\cdot\mathbb{E}\langle\ell, \psi^\gamma(f^\star(x))\rangle + \frac{C}{K}.$$

Finally since $p_t = (1-K\mu)\pi_{\text{hinge}}(z_t) + \mu$ and $\|\ell_t\|_\infty \leq 1$, this implies the bound

$$\sum_{t=1}^T \mathbb{E}\,\ell_t(a_t) \leq \frac{T}{K}\cdot\mathbb{E}\langle\ell, \psi^\gamma(f^\star(x))\rangle + \underbrace{\frac{C}{K} + \mu KT}_{\triangleq C'}.$$

We proceed to bound the final regret $C'$ under the specific covering number behavior assumed in the theorem statement. Assume that $\log(\mathcal{N}_{\infty,\infty}(\varepsilon,\mathcal{F})) \leq \varepsilon^{-p}$ for some $p > 2$. Omitting the $\log(T)$ additive terms, which will contribute $O(B\gamma^{-1}\sqrt{KT\log(T)/\mu} + B\gamma^{-1}\log(T)/\mu)$ to the overall regret, the bound is now

$$\mu KT + \frac{1}{\gamma K}\left(\inf_{\beta>0} 4KT\beta + 12\log_2(T)\cdot\int_\beta^2\sqrt{\frac{2KT}{\mu\varepsilon^p}}d\varepsilon + 36\log_2(T)\cdot\int_\beta^2\frac{1}{\mu\varepsilon^p}d\varepsilon\right).$$

Choosing $\beta = (KT\mu)^{-1/p}$, this bound becomes

$$O\left(\mu KT + \frac{1}{\gamma K}\log(T)(KT)^{1-1/p}\mu^{-1/p}\right).$$

Finally, we choose $\mu = \gamma^{-\frac{p}{p+1}}T^{-\frac{1}{p+1}}K^{-1}$, leading to a final bound of $O\left((T/\gamma)^{\frac{p}{p+1}}\right)$. $\qquad\square$

# F SMOOTHFTL for Lipschitz CB

Here we analyze SMOOTHFTL in a stochastic Lipschitz contextual bandit setting. To describe the setting, let $\mathcal{X}$ be a metric space endowed with metric $\rho$ and with covering dimension $p$. This latter fact means that for each $0 < \varepsilon \le 1$, $\mathcal{X}$ can be covered using at most $C_\mathcal{X} \varepsilon^{-p}$ balls of radius $\varepsilon$. Let $\mathcal{A}$ be a finite set of $K$ actions. In this section, we define the benchmark class $\mathcal{G} \subset (\mathcal{X} \to \Delta(\mathcal{A}))$ to be the set of 1-Lipschitz functions, meaning that $\|g(x) - g(x')\|_1 \le \rho(x, x')$ for all $g, x, x'$ (The choice of $\ell_1$ norm is natural since we are operating over the simplex).

We focus on the stochastic setting where there is a distribution $\mathcal{D}$ over $\mathcal{X} \times [0,1]^K$. At each round $(x_t, \ell_t) \sim \mathcal{D}$ is drawn and $x_t$ is presented to the learner. The learner chooses a distribution $p_t \in \Delta(\mathcal{A})$, samples an action $a_t \in \mathcal{A}$ from $p_t$, and observes the loss $\ell_t(a_t)$. We measure regret via

$$\text{Regret}(T, \mathcal{G}) = \sum_{t=1}^{T} \mathbb{E}\ell_t(a_t) - \inf_{g \in \mathcal{G}} T\mathbb{E}\langle g(x), \ell \rangle.$$

In this setting, SMOOTHFTL takes the following form. Before the $m^{\text{th}}$ epoch, we choose a function $\hat{g}_{m-1}$ by solving the empirical risk minimization (ERM) problem

$$\hat{g}_{m-1} = \operatorname*{argmin}_{g \in \mathcal{G}} \sum_{\tau=n_{m-1}}^{n_m - 1} \langle \hat{\ell}_\tau, g(x_\tau) \rangle,$$

where $\hat{\ell}_\tau$ is the importance weighted loss. Then, we use $\hat{g}_{m-1}$ for all the rounds in the $m^{\text{th}}$ epoch, which means that after observing $x_t$, we set $p_t(a) = (1 - K\mu)\hat{g}_m(x_t, a) + \mu$. We sample $a_t \sim p_t$, observe $\ell_t(a_t)$ and use the standard importance weighting scheme:

$$\hat{\ell}_t(a) = \frac{\ell_t(a_t)\mathbf{1}\{a = a_t\}}{p_t(a)}.$$

For this algorithm, we have the following guarantee.

**Theorem 27.** SMOOTHFTL *in the Lipschitz CB setting enjoys a regret of* $\tilde{\mathcal{O}}((KT)^{\frac{p}{p+1}})$.

This theorem improves upon the recent result of Cesa-Bianchi et al. [14], who obtain $\tilde{\mathcal{O}}(T^{\frac{p+1}{p+2}})$ in this setting.

**Proof.** We are in a position to apply Lemma 26. The main difference is that there is no margin parameter, since our functions are 1-Lipschitz, instead of $1/\gamma$-Lipschitz after applying the surrogate loss. The $\ell_\infty$-metric entropy at scale $\varepsilon$ is $C_\mathcal{X} \varepsilon^{-p}$ up to polynomial factors in $K$ and logarithmic factors, and so in the $m^{\text{th}}$ epoch the ERM has sub-optimality (see (18)) at most

$$\tilde{\mathcal{O}}\left( \inf_\beta K\beta + \int_\beta^1 \sqrt{\frac{K\beta^{-p}}{n_{m-1}\mu}} + \frac{\beta^{-p}}{n_{m-1}\mu} \right),$$

where $\tilde{O}$ hides dependence on $C_\mathcal{X}$. Following the argument in the proof of Theorem 9, the overall regret is then

$$\text{Regret}(T, \mathcal{G}) = \tilde{\mathcal{O}}\left( \mu KT + \inf_\beta TK\beta + \int_\beta^1 \sqrt{\frac{TK\beta^{-p}}{\mu}} + \frac{\beta^{-p}}{\mu} \right).$$

Set $\beta = (TK\mu)^{-1/p}$ and then $\mu = (TK)^{\frac{-1}{p+1}}$ now to obtain the result. $\qquad\square$

In principle our technique can be further extended to the setting where the action space is also a general metric space, and the losses are Lipschitz, which is the more general setting addressed by Cesa-Bianchi et al. [14]. If the action space has covering dimension $p_\mathcal{A}$ then we discretize the action space to resolution $\epsilon$, set $K = \epsilon^{-p_\mathcal{A}}$ in the above argument, and balance $\epsilon$ with an additional $T\epsilon$ factor that we pay for discretization. This is the approach used in Cesa-Bianchi et al. [14] to obtain $T^{\frac{p+p_\mathcal{A}+1}{p+p_\mathcal{A}+2}}$. Unfortunately, our argument above obtains a somewhat poor dependence on $K$ ($K^{\frac{p}{p+1}}$ as opposed to $K^{\frac{1}{p+1}}$, which is more natural). Consequently, the argument produces a bound of $\tilde{\mathcal{O}}(T^{\frac{p+pp_\mathcal{A}}{p+pp_\mathcal{A}+1}})$ which only improves on Cesa-Bianchi et al. [14] when $p_\mathcal{A} \le 1/(p-1)$.

## Footnotes

[10]Measuring loss in $\ell_1$ may seem restrictive, but this is natural when working with importance-weighted losses since these are 1-sparse, and by duality this enables us to cover in $\ell_{\infty}$ norm on the output space.