[Reviews · NeurIPS 2018]

Reviewer 1



[Summary] This paper studies a contextual bandit problem in which regret is defined against a given set of randomized policies that map from a context space to an arm space. As sets of randomized policies, the authors consider classes of composite functions of a K-dimensional 0-component-sum real vector-valued function over the context space and a component-wise ramp or hinge loss function. For ramp and hinge losses, they propose algorithms and prove their regret upper bounds. [Originality] Problem setting, proposed algorithms and obtained their regret upper bounds look new. [Strengths] * Nice theoretical results * technically sound [Weakness] * No empirical demonstration * Bad readability [Recommendation] I recommend this paper to be evaluated as "Marginally above the acceptance threshold". The theoretical results look nice, but it is very difficult to understand the role of surrogate losses. Benchmark sets of policies themselves are defined using surrogate loss functions, then are those losses what loss's surrogates are? In Lemma 1, make clear what loss is translated to what loss for what purpose. [Detailed Comments] p.2: Specify \gamma's range. p.4 Th 3 R^3: Don't use a footnote mark as a superscript of a mathematical formula. p.4 l.138: Don't refer a theorem whose statement is in Appendix. [Comments to Authors' Feedback] In my understanding, a surrogate loss function means a surrogate of some loss function such as a 0-1 loss function. However, the authors' surrogate losses do not replace some other losses. In that meaning, it might be better to be called not a surrogate loss but a somewhat other name.

Reviewer 2



The paper was too technical for a conference review (with 34 pages of supplement!), that too within a short period of time. I glanced over the high level contributions and if the proofs are correct, would definitely fit into the theoretical niche areas of NIPS. I would defer to the opinion of other reviewers who might have been able to provide more time to the paper for a thorough review.

Reviewer 3



This submission may contain some important results, but I did not see clear motivation to consider these losses (only the second paragraph in the Introduction, which is not very convincing). And, it is not clear why new algorithms are needed. I understand that this submission is theoretical, but it would be good to mention some potential applications or when and where we should use these new losses and the new algorithms.